# ARTICLES

## OPEN

# The metabolic enzyme hexokinase 2 localizes to the nucleus in AML and normal haematopoietic stem and progenitor cells to maintain stemness

Geethu Emily Thomas[1,6], Grace Egan[1,2,6], Laura García-Prat[1], Aaron Botham[1], Veronique Voisin[3], Parasvi S. Patel[1], Fieke W. Hoff[4], Jordan Chin[1], Boaz Nachmias [1], Kerstin B. Kaufmann[1], Dilshad H. Khan[1], Rose Hurren[1], Xiaoming Wang[1], Marcela Gronda[1], Neil MacLean[1], Cristiana O'Brien[1], Rashim P. Singh[1], Courtney L. Jones[1], Shane M. Harding [1], Brian Raught[1], Andrea Arruda[1], Mark D. Minden[1], Gary D. Bader [3], Razq Hakem[1], Steve Kornblau[5], John E. Dick[1] and Aaron D. Schimmer [1✉]

**Mitochondrial metabolites regulate leukaemic and normal stem cells by affecting epigenetic marks. How mitochondrial enzymes localize to the nucleus to control stem cell function is less understood. We discovered that the mitochondrial metabolic enzyme hexokinase 2 (HK2) localizes to the nucleus in leukaemic and normal haematopoietic stem cells. Overexpression of nuclear HK2 increases leukaemic stem cell properties and decreases differentiation, whereas selective nuclear *HK2* knockdown promotes differentiation and decreases stem cell function. Nuclear HK2 localization is phosphorylation-dependent, requires active import and export, and regulates differentiation independently of its enzymatic activity. HK2 interacts with nuclear proteins regulating chromatin openness, increasing chromatin accessibilities at leukaemic stem cell-positive signature and DNA-repair sites. Nuclear HK2 overexpression decreases double-strand breaks and confers chemoresistance, which may contribute to the mechanism by which leukaemic stem cells resist DNA-damaging agents. Thus, we describe a non-canonical mechanism by which mitochondrial enzymes influence stem cell function independently of their metabolic function.**

Acute myeloid leukaemia (AML) is characterized by the clonal proliferation of immature myeloid precursors paired with arrested differentiation. Similar to normal haematopoiesis, AML is organized in a hierarchy, with leukaemic stem cells (LSCs) responsible for replenishing the bulk population of AML cells and driving long-term clonal growth[1,2]. Despite the importance of stem cells in normal and malignant haematopoiesis, the factors that regulate the growth and differentiation of leukaemic and normal stem cells are not yet fully elucidated.

Metabolic intermediates produced in the mitochondria—including acetyl-CoA, α-ketoglutarate, *S*-adenosylmethionine and nicotinamide adenine dinucleotide—are known to regulate stem cell function and differentiation by serving as cofactors for the epigenetic modification of nuclear genes[3,4]. Similarly, mutations in metabolic enzymes such as isocitrate dehydrogenase 1 and 2 (IDH1 and IDH2) generate oncometabolites, which lead to increased histone and DNA methylation, promoting leukemogenesis and inhibiting differentiation[5,6]. Although metabolites that regulate stem cell function have been described, much less is known about the mitochondrial metabolic enzymes that 'moonlight' in the nucleus to directly influence gene expression, cell differentiation and stem cell function.

In this study we discovered that the initial and rate-limiting enzyme in the glycolytic pathway hexokinase 2 (HK2) can localize to the nucleus in AML and normal haematopoietic stem and progenitor cells. Nuclear HK2 modifies stem/progenitor cell function and differentiation independently of its kinase and metabolic function. Thus, we describe a non-canonical mechanism by which a mitochondrial enzyme regulates gene expression and stem cell function.

## Results

**HK2 localizes to the nucleus in AML.** Metabolic intermediates produced in the mitochondria regulate epigenetic marks to influence the function and differentiation of stem cells[7-11]. However, it is largely unknown whether metabolic enzymes that traditionally localize to the mitochondria moonlight in the nucleus to directly impact stem cell function and differentiation. To identify such moonlighting proteins, we searched for mitochondrial glycolytic and tricarboxylic acid-cycle enzymes that were present in the nucleus of 8227 cells, a low-passage primary AML model[12]. We detected HK2 in the nucleus of the 8227 cells. HK2 is the first enzyme in the glycolytic pathway and it converts glucose to glucose-6-phosphate (Extended Data Fig. 1a). We also detected the glycolytic enzyme aldolase in the nucleus but its nuclear expression was much lower compared with the cytoplasmic fraction. In contrast, other metabolic enzymes, including phosphofructokinase, fumarase, pyruvate

[1]Princess Margaret Cancer Centre, University Health Network, Toronto, Ontario, Canada. [2]Division of Hematology/Oncology, The Hospital for Sick Children, Toronto, Ontario, Canada. [3]Terrence Donnelly Centre for Cellular and Biomedical Research, University of Toronto, Toronto, Ontario, Canada. [4]Department of Pediatric Hematology/Oncology, University Medical Center Groningen, Groningen, The Netherlands. [5]Section of Molecular Hematology and Therapy, Department of Leukemia, The University of Texas MD Anderson Cancer Center, Houston, TX, USA. [6]These authors contributed equally: Geethu Emily Thomas, Grace Egan. ✉e-mail: aaron.schimmer@uhn.ca

kinase 2, glucose phosphate isomerase, enolase 1, citrate synthase, aconitase 2 and succinate dehydrogenase, were not detected in the nuclear lysates by immunoblotting (Fig. 1a).

We also detected HK2, but not other metabolic enzymes, in the nucleus of NB4, OCI-AML2, U937 and TEX leukaemia cells by immunoblotting and confocal microscopy (Extended Data Fig. 1b–d). Finally, nuclear HK2 was detected in seven of nine primary AML samples by immunoblotting (Extended Data Fig. 1e,f) and confirmed by reverse-phase protein array (RPPA) analysis (Fig. 1d) and confocal microscopy (Fig. 1c and Extended Data Fig. 1g). Therefore, we focused our study on HK2.

8227 cells are arranged in a hierarchy with functionally defined stem cells in the CD34+CD38− fraction. We separated 8227 cells into CD34+CD38− stem cells and CD34−CD38+-committed populations by fluorescence-activated cell sorting (FACS), and prepared nuclear and cytoplasmic lysates. The nuclear and total levels of HK2 were higher in AML stem cells compared with bulk cells, as determined by immunoblotting and confocal microscopy (Fig. 1a,b and Extended Data Fig. 1h).

Finally, we measured the levels of nuclear HK2 in primary AML samples separated into functionally defined leukaemic stem and bulk populations based on low and high expression of reactive oxygen species (ROS), respectively[13,14]. Confocal microscopy revealed enhanced nuclear HK2 in stem cells versus bulk cells (Fig. 1c and Extended Data Figs. 1h, 10h).

**Nuclear HK2 is important for stem and progenitor function in AML.** To test whether nuclear HK2 is required for AML stem and progenitor cell function, we selectively increased HK2 in the nucleus by expressing HK2 tagged with a c-Myc (NLS1, PAAKRVKLD) or SV40 (NLS2, PKKKRKV) nuclear localizing signal[15] (NLS; Extended Data Fig. 2a). Through confocal microscopy (Extended Data Fig. 2b,c,e) and immunoblotting (Extended Data Fig. 2d), we confirmed increased levels of nuclear HK2, whereas the levels of cytoplasmic and mitochondrial HK2 were unchanged (Extended Data Fig. 2d). Overexpression of nuclear HK2 increased the clonogenic efficiency of NB4 cells (Extended Data Fig. 2g) without altering their proliferation rate in culture (Extended Data Fig. 2f). We also measured the effects of nuclear HK2 on α-retinoic acid (ATRA)-mediated differentiation in NB4 cells (Extended Data Fig. 2i–k). The levels of HK2 in the nucleus decreased after differentiation with ATRA, whereas the levels of mitochondrial HK2 remained unchanged (Extended Data Fig. 2i–k). In addition, as measured by expression of the differentiation marker CD11b and clonogenic growth, overexpression of nuclear HK2 protected the cells from ATRA-induced differentiation (Extended Data Fig. 2g,h).

We also examined the functional importance of nuclear HK2 overexpression in 8227 and 130578 cells. Similar to 8227 cells, 130578 cells are a low-passage primary AML model arranged in a functional hierarchy with the stem cells located in the CD34+CD38− compartment[16]. Both 8227 and 130578 cells are sensitive to ATRA.

In both 8227 and 130578 cells, overexpression of nuclear HK2 protected the CD34+CD38− leukaemic stem cell fraction from ATRA without changing their basal growth rate (Extended Data Fig. 2l–o).

Finally, we tested whether nuclear HK2 influenced the engraftment of AML cells into mouse marrow. We transduced 8227 cells with NLS1–HK2 and injected them into the femur of sublethally irradiated NOD/SCID-GF mice. Eight weeks later the percentage of leukaemic cells in the uninjected mouse femur was quantified by flow cytometry. Overexpression of nuclear HK2 increased the engraftment of 8227 cells into the mouse marrow (Fig. 1e,f). Using serial transplants of 8227 cells harvested from the marrow of the primary recipients into secondary recipients, overexpression of nuclear HK2 continued to increase the engraftment efficiency, further demonstrating a functional effect on the stem cell population (Fig. 1g). Moreover, mice engrafted with NLS1–HK2-transduced TEX cells showed decreased survival compared with mice engrafted with empty vector (EV)-transduced control TEX cells (Extended Data Fig. 2p,q).

**Depletion of nuclear HK2 decreases AML stem cell function.** As an additional approach to understand the importance of nuclear HK2 in maintaining AML stem cell function, we knocked down HK2 in the nucleus while sparing mitochondrial HK2. NB4 and 8227 cells were transduced with HK2 tagged with an outer mitochondrial membrane-localizing signal (OMMLS) from the carboxy (C)-terminal region of OPA25 (Extended Data Fig. 3a,b) to ensure mitochondrial tethering of HK2 (ref. [17]). Selective localization of OMMLS–HK2 to the mitochondria was confirmed by confocal microscopy (Fig. 2a) and immunoblotting (Extended Data Fig. 3c,d) and OMMLS–HK2-transfected cells were found to be more resistant to 2-deoxy-D-glucose (2-DG) compared with the EV-transfected control cells, demonstrating that the mitochondrial tagged protein is metabolically active (Fig. 2b and Extended Data Fig. 3h). Overexpression of OMMLS–HK2 also increased cell proliferation, consistent with previous reports in which HK2 was overexpressed in cells (Extended Data Fig. 3g)[18].

Cells overexpressing OMMLS–HK2 were then transduced with short hairpin RNA (shRNA) targeting endogenous *HK2* to deplete nuclear HK2 while preserving HK2 in the mitochondria in NB4 and 8227 cells (Fig. 2c and Extended Data Fig. 3c–f,l,m). Selective knockdown of nuclear *HK2* reduced the clonogenic growth of NB4 cells (Fig. 2d) and the number of 8227 LSCs (CD34+CD38−; Fig. 2f). In addition, selective knockdown of nuclear *HK2* reduced the growth and engraftment of AML cells in vivo (Fig. 2g–i and Extended Data Fig. 3k).

The gene expression profile of 8227 cells with nuclear *HK2* knockdown was compared with the genetic signatures of primary AML stem cells (LSC+) and bulk cells (LSC−)[19]. Knockdown of nuclear *HK2* in 8227 cells decreased the expression of genes associated with primitive/stem-like AML fractions (LSC+; Fig. 2j,k). The nuclear *HK2*-knockdown gene expression profile was compared

**Fig. 1 | HK2 localizes to the nuclei of leukaemic stem and progenitor cells. a**, Immunoblot of glycolytic and tricarboxylic acid-cycle enzymes in the nucleus, cytoplasm and whole-cell lysate of FACS-sorted stem and bulk 8227 cells. Representative immunoblot from $n = 3$ biological repeats. **b**, Confocal microscopy images of HK2 and the mitochondrial protein Tom20 in FACS-sorted stem and bulk 8227 cells. Representative images from $n = 3$ biological repeats. **c**, Confocal microscopy images of HK2 in ROS-low LSCs and ROS-high bulk primary cells from patients with AML. Images are representative of three biologically independent samples. **d**, Nuclear HK2 expression in samples from patients with AML ($n = 25$) and AML cell lines ($n = 15$), determined using RPPA. Patient samples: minimum, −2.696; maximum, −1.200; and median −1.679; AML cell lines: minimum, −3.1878; maximum, −0.5461; and median, −1.997. In the box-and-whisker plots, the horizontal lines mark the median, the box limits indicate the 25th and 75th percentiles, and the whiskers extend to 1.5× the interquartile range from the 25th and 75th percentiles. **e**, 8227 cells were transduced with NLS1–HK2 or control, using a blue fluorescent protein (BFP)-expressing vector. BFP-sorted cells were imaged using confocal microscopy. Representative images of HK2 in control-vector and NLS1–HK2 8227 cells from $n = 3$ biological repeats are shown. **f**, The right femur of NOD/SCID-GF mice ($n = 7$ EV and 8 NLS1–HK2 mice) was injected with 8227 cells transduced with NLS–HK2 or control vector. Eight weeks post injection, engraftment of 8227 cells into the uninjected left femur was measured by flow cytometry. **g**, Cells from **f** were injected into secondary mice and the engraftment efficiency was measured 8 weeks later by flow cytometry ($n = 7$ mice per group). **b,c,e**, Scale bars, 10 μm. **f,g**, Statistical analyses were performed using a two-tailed unpaired Student's *t*-test. Data represent the mean ± s.e.m.

with 20 different subfractions of normal haematopoietic cells from 38 human samples[20]. Genes associated with haematopoietic stem cells (HSC-like) were observed to be significantly reduced in nuclear *HK2*-knockdown conditions (Fig. 2l and Extended Data Fig. 3n).

Knockdown of nuclear *HK2* also increased the sensitivity of NB4 cells to ATRA, as demonstrated by increased levels of

CD11b⁺ cells (Fig. 2e and Extended Data Fig. 3i) and decreased clonogenic growth (Fig. 2d and Extended Data Fig. 3j). Finally, knockdown of nuclear *HK2* in 8227 cells reduced the percentage of CD34⁺CD38⁻ stem cells after ATRA treatment (Fig. 2f). Thus, nuclear HK2 is essential for stem cell function and differentiation in AML.

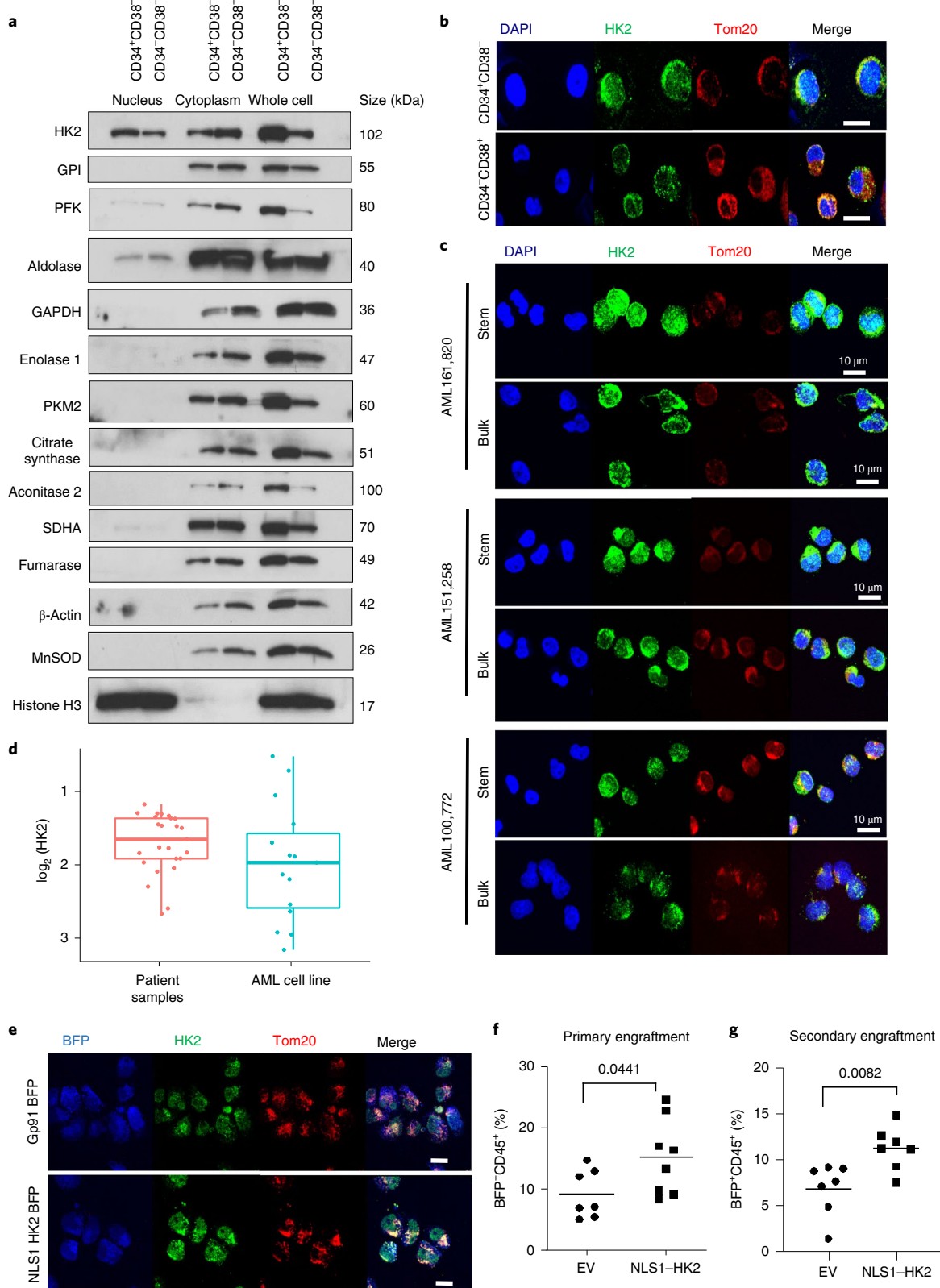

**HK2 localizes to haematopoietic stem and progenitor cell nuclei.**
We also examined the nuclear expression and functional importance of HK2 in normal haematopoiesis. Subpopulations of normal haematopoietic cells were isolated from cord blood by high-resolution sorting and the levels of HK2 were measured by confocal microscopy. Both the nuclear and total levels of HK2 were higher in haematopoietic stem cells (HSCs) and multipotent progenitor fractions, and declined as the cells matured, with minimal amounts of nuclear HK2 in differentiated cells (Fig. 3a and Extended Data Fig. 4a,b).

Overexpression of nuclear HK2 in normal cord blood increased the levels of primary and secondary engraftment of these cells into mice (Fig. 3b,c), indicating nuclear HK2 was functionally important for normal HSCs.

To further explore the effects of nuclear HK2 on normal haematopoiesis, we created a transgenic mouse overexpressing nuclear HK2 driven by a Vav promoter to selectively increase HK2 expression in the haematopoietic system (Fig. 3d and Extended Data Fig. 4c). The size and weight of the Vav-NLS–HK2 transgenic mice was similar to their wild-type littermates (Extended Data Fig. 4d,e). However, the Vav-NLS–HK2 mice had increased abundance of HSCs in the marrow and decreased levels of monocytes and lymphocytes in the peripheral blood (Fig. 3e,f and Extended Data Figs. 4f, 10i).

To functionally evaluate HSCs in Vav-NLS–HK2 transgenic mice, we performed competitive repopulation assays using the CD45.1 and CD45.2 congenic system. The efficacy of bone marrow reconstitution was determined in chimaera mice by mixing (1:1 ratio) donor cells (CD45.2+; C57B6 background; Vav-NLS–HK2 or wild type) and clonogenic competitor cells (CD45.1+; B6.SJL). Compared with the wild-type chimaera, the Vav-NLS–HK2 chimaera mice demonstrated increased repopulation in the peripheral blood at weeks 4, 6, 8 and 10, and in the bone marrow at week 12 (Fig. 3g,h and Extended Data Fig. 10j).

**Nuclear HK2 maintains stemness independently of its kinase activity.** We next examined the mechanisms that control the nuclear localization of HK2. AKT-dependent phosphorylation of HK2 at Thr 473 positively controls mitochondrial HK2 localization, whereas dephosphorylation by protein phosphatase (PHLPP) decreases its mitochondrial localization[21]. Consistent with the role of phosphorylation in HK2 mitochondrial localization, inhibition of AKT increased the nuclear levels of HK2 (Extended Data Fig. 5a,b). Conversely, inhibition of PHLPP1 decreased the nuclear levels of HK2 (Extended Data Fig. 5c). In contrast to findings in

yeast[22,23], the amount of glucose in the media did not impact HK2 localization (Extended Data Fig. 5d).

HK2 is a transferase kinase that phosphorylates glucose at the outer mitochondrial membrane to glucose-6-phosphate, initiating glycolysis[24]. We next investigated whether the kinase activity of HK2 is necessary for its nuclear function to maintain stem cell properties. Asp 209 and Asp 657 are residues in the HK2 protein that are necessary for its catalytic function. Double mutation of these residues (D209A/D657A) in HK2 results in a complete loss of its catalytic activity[25]. Therefore, we created a kinase-dead double mutant of nuclear HK2 tagged with the c-Myc NLS (NLS1–HK2 D209A/D567A; Extended Data Fig. 5e). We overexpressed NLS1–HK2 D209A/D567A in NB4 cells and confirmed its selective localization to the nucleus by confocal microscopy (Fig. 4a). Overexpression of the nuclear-localized kinase-dead HK2 produced a phenotype similar to overexpression of nuclear HK2 with a wild-type kinase domain (NLS1–HK2 and NLS2–HK2; Fig. 4b). Specifically, kinase-dead nuclear HK2 enhanced clonogenic growth and blocked cell differentiation after ATRA treatment, similar to NLS1–HK2 and NLS2–HK2 (Fig. 4c,d). Notably, HK2 also has a PAR-binding motif, but modification of this motif did not change the nuclear function of HK2 (Extended Data Fig. 5f–l). Thus, nuclear HK2 maintains stem and progenitor functions independently of its kinase and metabolic activity.

Nuclear import of proteins is frequently regulated by the importin family of proteins that bind the NLS sequences of cargo proteins. Amino-acid sequence alignment analysis demonstrated multiple mono and bipartite NLS sequences at the C and amino (N) termini of HK2 (Extended Data Fig. 6a,b). Through the analysis of a dataset of high-throughput protein–protein interactors in HeLa cells, we found cofractionation of HK2 with the importin IPO5 (BioGRID: https://thebiogrid.org). To test whether IPO5 is important for HK2 nuclear localization, we knocked down IPO5 and measured the nuclear and total levels of HK2 in NB4 cells. Knockdown of IPO5 decreased the nuclear levels of HK2 (Fig. 4e,f). Demonstrating specificity for IPO5, knockdown of IPO11, a related importin family member that was not identified as an interactor with HK2 in this database, did not alter HK2 localization (Fig. 4f).

Proteins are exported from the nucleus via the exportin complex that binds the nuclear-export signal on cargo protein. Analysis of the genetic and amino-acid sequences also led to the identification of three distinct nuclear-export-signal sequences at the N terminus of HK2 (Extended Data Fig. 6c). A previous study demonstrated that leptomycin inhibited the export of nuclear HK2 in HeLa cells[26].

**Fig. 2 | Knockdown of nuclear *HK2* reduces stem and progenitor cell function. a**, Confocal microscopy images of HK2 and Tom20 staining in NB4 cells 5 d after transduction with OMMLS–HK2 or EV. **b**, Half maximal inhibitory concentration (IC$_{50}$) of 2-DG in OMMLS–HK2 and control NB4 cells 48 h after treatment. **c**, Confocal images of HK2 staining after transduction of OMMLS–HK2 NB4 cells with two shRNA (SH1 and SH2) targeting the untranslated region (UTR) of *HK2*. **a,c**, Scale bars,10 μm. Images are representative of *n* = 3 biological repeats. **d**, Clonogenic growth of OMMLS–HK2 NB4 cells transduced with shRNAs targeting the *HK2* UTR and treated with ATRA (100 nM). **e**, Expression levels of CD11b in OMMLS–HK2 NB4 cells transduced with shRNA to the *HK2*-UTR and treated with ATRA (100 nM). **f**, Percentage of CD34+CD38− cells in 8227 cells transduced with OMMLS–HK2 and *HK2*-UTR shRNAs, in incubated with or without 100 nM ATRA for 24 h. **b,d–f**, *n* = 3 biological repeats. **g**, OCI-AML2 cells with selective nuclear *HK2* knockdown or EV were subcutaneously injected into the flanks of SCID mice (*n* = 10 mice per group). Tumour volume was measured every alternate day for 18 d, starting 6 d after injection. **h**, OMMLS-HK2 AML2 cells were transduced with shRNAs targeting the *HK2* UTR and subcutaneously injected into the flanks of SCID mice. The weights of subcutaneous tumours were measured at the end of the experiment (*n* = 10 mice per group). **i**, TEX cells with nuclear *HK2* knockdown or EV were injected into the right femur of NOD/SCID-GF mice (*n* = 5 per group). Engraftment of TEX cells into the uninjected left femur was measured by flow cytometry 5 weeks post injection. **j**, GSEA of 8227 cells transduced with OMMLS–HK2 and shRNAs targeting the *HK2* UTR. The NES and *P* values were analysed using a modified Kolmogorov–Smirnov test. **k**, Gene-set variation analysis score for LSC-positive gene signatures in 8227 cells transduced with OMMLS–HK2 and shRNAs targeting the *HK2* UTR. Control: minimum, 0.8600; maximum, 1.271; and median, 1.066. SH1: minimum, −0.9867; maximum, −0.4826; and median, −0.6619. **l**, Gene-set variation analysis (GSVA) score for HSC gene signatures in 8227 cells transduced with OMMLS–HK2 and shRNAs targeting the *HK2*-UTR. Control: minimum, 0.09000; maximum, 0.2500; and median, 0.1700. SH1: minimum, −0.2500; maximum, −0.09000; and median, −0.1500. **k,l**, In the box-and-whisker plots, the horizontal lines mark the median, the box limits indicate the 25th and 75th percentiles, and the whiskers extend to 1.5× the interquartile range from the 25th and 75th percentiles. **b,d–i,k,l**, Statistical analyses were performed using a two-tailed unpaired Student's *t*-test (**b,h,i,k,l**) and ordinary one-way (**d–f**) or two-way (**g**) analysis of variance (ANOVA) with Sidak's multiple comparison test; *P* values for comparisons to the control are provided. Data represent the mean ± s.e.m. Nuc-HK2 KD, nuclear *HK2* knockdown.

We confirmed these findings in leukaemia cells; treatment with both leptomycin and selinexor increased the levels of nuclear HK2, demonstrating that the nuclear export of HK2 requires exportin 1 (XPO1,

also known as CRM1; Fig. 4g,h). Reverse-phase protein array analysis performed on 20 leukaemic cell lines showed increased nuclear and decreased cytoplasmic HK2 levels after 24 h of XPO1-inhibitor

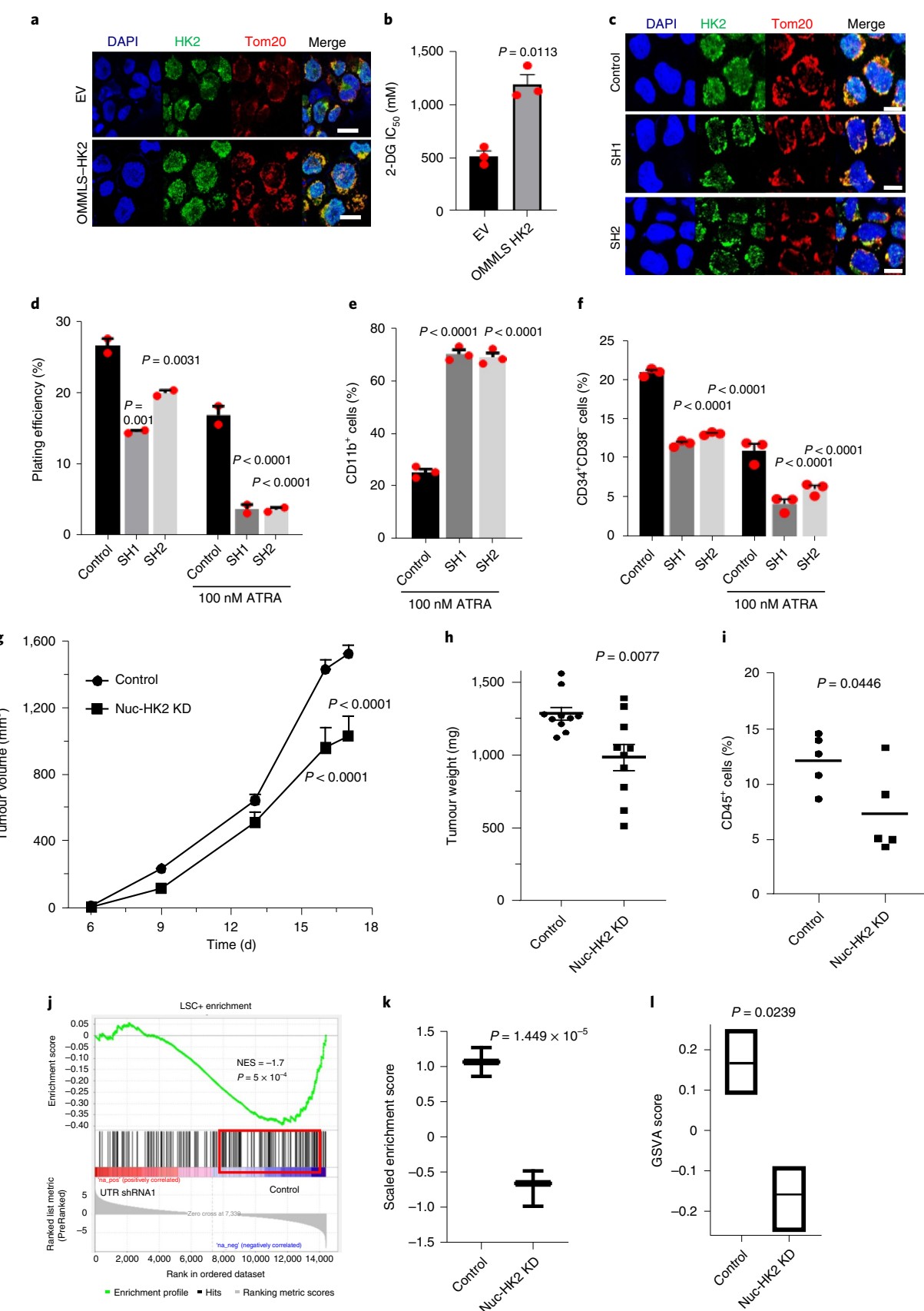

treatment (Fig. 4i). Thus, together, the nuclear localization of HK2 is dependent on its phosphorylation and requires active import and export through IPO5 and XPO1, respectively.

**Nuclear HK2 modulates chromatin accessibility.** To understand the mechanism by which HK2 regulates stem cell function, we sought to identify nuclear proteins that interact with HK2 using proximity-dependent biotin labelling (BioID) coupled with mass spectrometry[27]. We overexpressed NLS1–HK2 tagged with a mutant *Escherichia coli* biotin-conjugating enzyme, BirA R118G (BirA*), in Flp-In T-REx human embryonic kidney (HEK) 293 cells to biotinylate proteins interacting with HK2. From our BioID screen, we identified 12 proteins that preferentially interacted with nuclear HK2 over the control; these included proteins involved in chromatin organization and regulation (CTR9, MAX, PHF8, PHF10 and SPIN1), transcriptional regulation (AASDH, CCNL2, IWS1 and ZNF136) and DNA-damage response (SIRT1, TDP2 and UBR5; Fig. 5a). To further characterize endogenous nuclear HK2 interactions in leukaemic cells, we performed in situ protein ligation assay (PLA) to visualize protein–protein interactions in intact cells using fluorescence microscopy[28]. Using this assay, we confirmed interactions between endogenous HK2 and MAX, SIRT1, IWS1, CTR9 and SPIN1 (Fig. 5b and Extended Data Fig. 6d).

Given that nuclear HK2 interacted with proteins that regulate chromatin organization, we examined how nuclear HK2 impacted global chromatin accessibility using transposase-accessible chromatin using sequencing (ATAC-seq). Overexpression of nuclear HK2 increased chromatin accessibility (Fig. 5c and Extended Data Fig. 6e,g,i), whereas selective knockdown of nuclear *HK2* decreased chromatin accessibility (Extended Data Fig. 6f,h,j). These observations are in line with previous studies demonstrating that stem cells have more accessible chromatin[29–31]. Genes that were more accessible in nuclear HK2 overexpression were enriched in genes associated with an LSC+ signature (Fig. 5d,e). Differentially accessible regions enriched after overexpression of nuclear HK2 were associated with Gene Ontology processes including chromatin assembly, mitotic checkpoints, regulation of stem cell differentiation and DNA-damage response, and genes in these pathways overlapped with HSC signature genes from published datasets (Fig. 5f).

To identify regions of DNA bound by HK2 or HK2 complexes, we performed chromatin immunoprecipitation with sequencing (ChIP–seq) in NB4 cells. A total of 6,350 unique peaks common to three biological replicates were identified. Analysis of these peaks indicated enrichment for pathways associated with the regulation of stem cell differentiation, embryonic development, cell-cycle transition/checkpoint regulation and DNA-damage response (Fig. 5g and Extended Data Fig. 7a). We also performed ChIP–seq following overexpression of nuclear HK2 (tagged with either NLS1 or NLS2). Overexpression of nuclear NLS1–HK2 and NLS2–HK2 increased

the number of peaks, with 17,530 and 8,655 unique peaks common to three biological replicates, respectively. Genes identified by ChIP–seq with HK2 overlapped with NLS1–HK2 and NLS2–HK2, and were highly enriched compared with the control (Fig. 5g and Extended Data Fig. 7a,g–i).

Motif enrichment analysis of the ChIP–seq peaks led to the identification of a consensus sequence, CACGTG, which corresponds to the basic helix loop helix (bHLH) E-box motif (Fig. 5h and Extended Data Fig. 7b–e). Estimation of the CACGTG motif enrichment using TFmotifView revealed significant enrichment of this motif in peaks identified in the previously mentioned pathways (Fig. 5i and Extended Data Fig. 7f). Furthermore, the genomic position of the motif was enriched at the centre of the peak representing the peak summit, which would model the binding of HK2 or HK2 complexes to this region (Fig. 5i). Interestingly, our BioID results demonstrated that HK2 interacts with MYC-associated factor X (MAX), which is a known class III bHLH E-Box-binding protein implicated in cell proliferation, stem cell maintenance and differentiation as well as the DNA-damage response[32]. To further explore the interaction between HK2 and MAX, peaks that overlapped between the NLS–HK2 ChIP–seq and MAX ChIP–seq data in NB4 cells from the ENCODE ENCSR000EHS dataset were retrieved. Similar binding peaks were identified in pathways related to the DNA-damage response, regulation of stem cell differentiation and mitotic cell cycle, with a false-discovery rate (FDR) for a gene set enrichment analysis (GSEA) pathway enrichment of 0.000001 (Extended Data Fig. 8a–d).

**Nuclear HK2 enhances the DNA-damage response in AML.** Nuclear HK2 interacted with genes associated with the DNA-damage response and increased accessibility of genes in the DNA-damage-response pathways. Therefore, we examined the DNA-damage response in stem and bulk AML populations and investigated whether nuclear HK2 could influence DNA-damage repair. We overexpressed nuclear HK2 in 8227 and NB4 cells, and treated the cells with daunorubicin, an intercalating chemotherapeutic agent that causes double-strand DNA breaks[33]. We then measured the double-strand DNA breaks and expression of DNA-repair proteins at multiple time points following daunorubicin treatment by quantifying the levels of γH2AX, 53BP1 and RAD51 in the nucleus. RAD51 is essential for homologous recombination, γH2AX is a surrogate marker of double-strand breaks and 53BP1 mediates non-homologous end-joining repair[34,35]. Overexpression of nuclear HK2 increased the levels of nuclear 53BP1 and RAD51 expression, and decreased the number of double-strand breaks, as measured by γH2AX foci in 8227 and NB4 cells (Fig. 6a,c,e and Extended Data Fig. 9a,d–j). As measured using the comet assay, 8227 cells overexpressing nuclear HK2 had decreased levels of double-strand DNA breaks before and after treatment with daunorubicin (Fig. 6h and

**Fig. 3 | HK2 localizes to the nuclei of haematopoietic stem and progenitor cells. a**, Fluorescence intensity of nuclear HK2 in haematopoietic cell populations from cord blood. A.u., arbitrary units; n = 183 cells examined from three biological samples. **b**, CD34+-enriched normal haematopoietic cells were transduced with NLS1–HK2 or control vector and injected into the right femur of NOD/SCID-GF mice (n = 4 (EV) and 5 (NLS1–HK2) mice). Engraftment of transduced cord blood cells in the left femur of mice was measured using flow cytometry 8 weeks after the injection. **c**, Cells from **b** were injected into secondary mice and the engraftment efficiency was measured 8 weeks later using flow cytometry (n = 5 mice per group). **d**, Confocal microscopy images of HK2 and Tom20 staining in bone marrow cells from Vav-NLS–HK2 mice and control littermate wild-type mice. Scale bar, 10 μm. Images are representative of n = 20 biologically independent samples. **e**, Percentage of lin−ckit+Sca+CD48−CD150+ cells in the bone marrow of the Vav-NLS–HK2 mice and their littermate controls (n = 12 (wild-type control) and 10 (Vav-NLS–HK2) mice). **f**, Percentage of granulocyte-monocyte and common myeloid progenitor (left), and megakaryocyte–erythroid progenitor (right) cells in the bone marrow of the Vav-NLS–HK2 mice and their littermate controls (n = 9 (wild-type control) and 6 (Vav-NLS–HK2) mice). **g**, Bone marrow cells from Vav-NLS–HK2 mice or their littermates (CD45.2+; donor) were co-transplanted with CD45.1+ bone marrow cells as competitors (1:1 ratio) into B6.SJL recipient mice (CD45.1+). Reconstitution units (CD45.2/CD45.1) were analysed in the peripheral blood of the chimaera mice over the specified period using flow cytometry (n = 10 mice per group). **h**, Reconstitution efficacy in the bone marrow from **g** was analysed at week 12 (n = 10 mice per group). **b,c,e–h**, Statistical analyses were performed using a two-tailed unpaired Student's t-test. Data represent the mean ± s.e.m. MPP, multipotent progenitors; MLP, multilymphoid progenitors; CMP, common myeloid progenitors; GMP, granulocyte-monocyte progenitors; and MEP, megakaryocyte–erythroid progenitors.

Extended Data Fig. 9b), and overexpression of nuclear HK2 conferred resistance to daunorubicin (Fig. 6g) and the PARP inhibitor olaparib (Extended Data Fig. 9c).

As nuclear HK2 is increased in AML stem cells under basal conditions, we compared the DNA-damage response in AML stem versus bulk cells. We FACS-sorted 8227 cells into stem and bulk fractions and measured the expression levels of DNA-repair genes.

Leukaemic stem cells were primed to respond to DNA damage, with increased expression of genes associated with homologous recombination (*XRCC2* and *XRCC3*) and non-homologous end joining (*XRCC4*, *XRCC5* and *PRKDC*; Fig. 6i). Next, we treated the stem and bulk cell fractions with daunorubicin. Compared with the bulk cells, the stem cell fraction demonstrated an enhanced DNA-damage response, with increased levels of 53BP1 and RAD51, and decreased

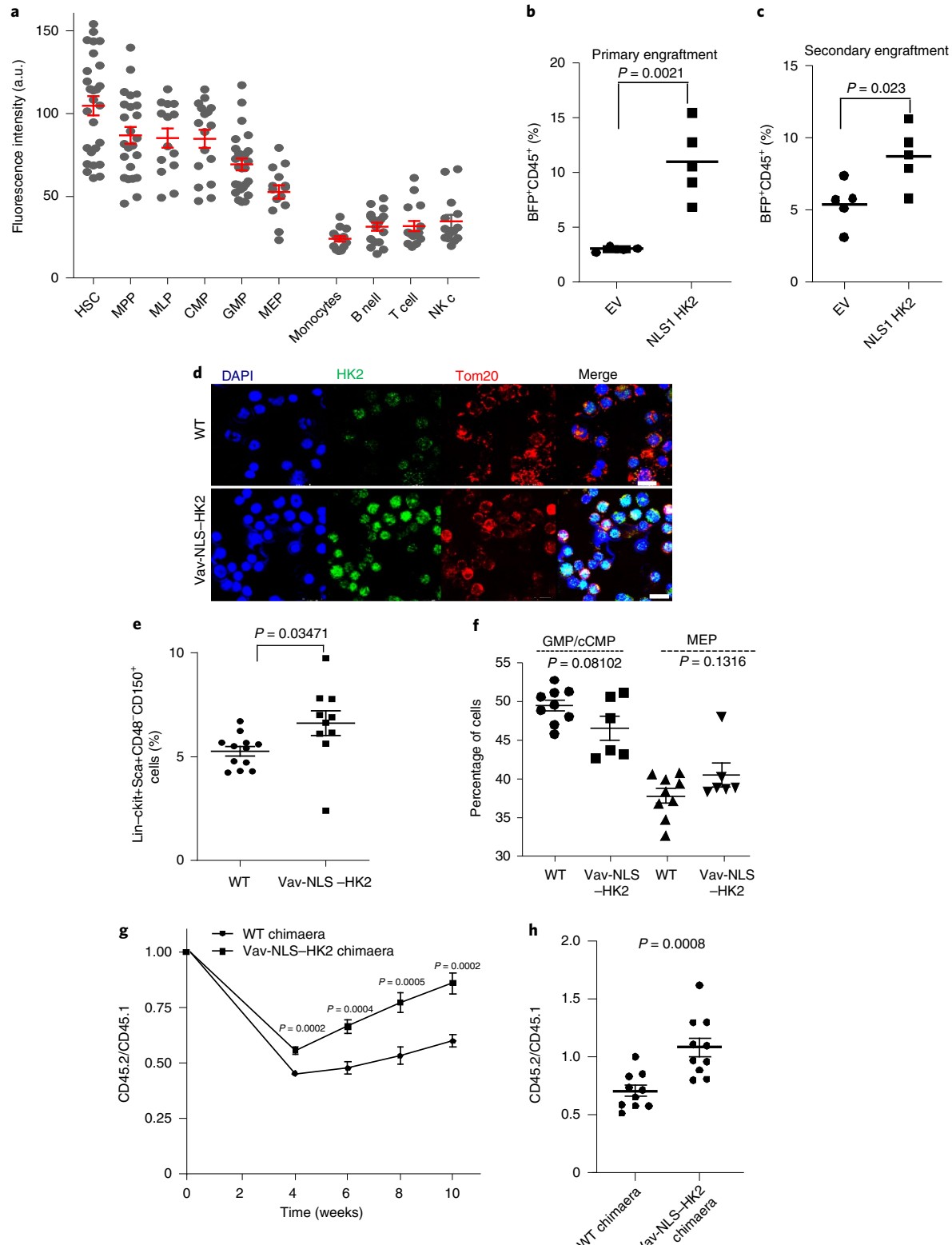

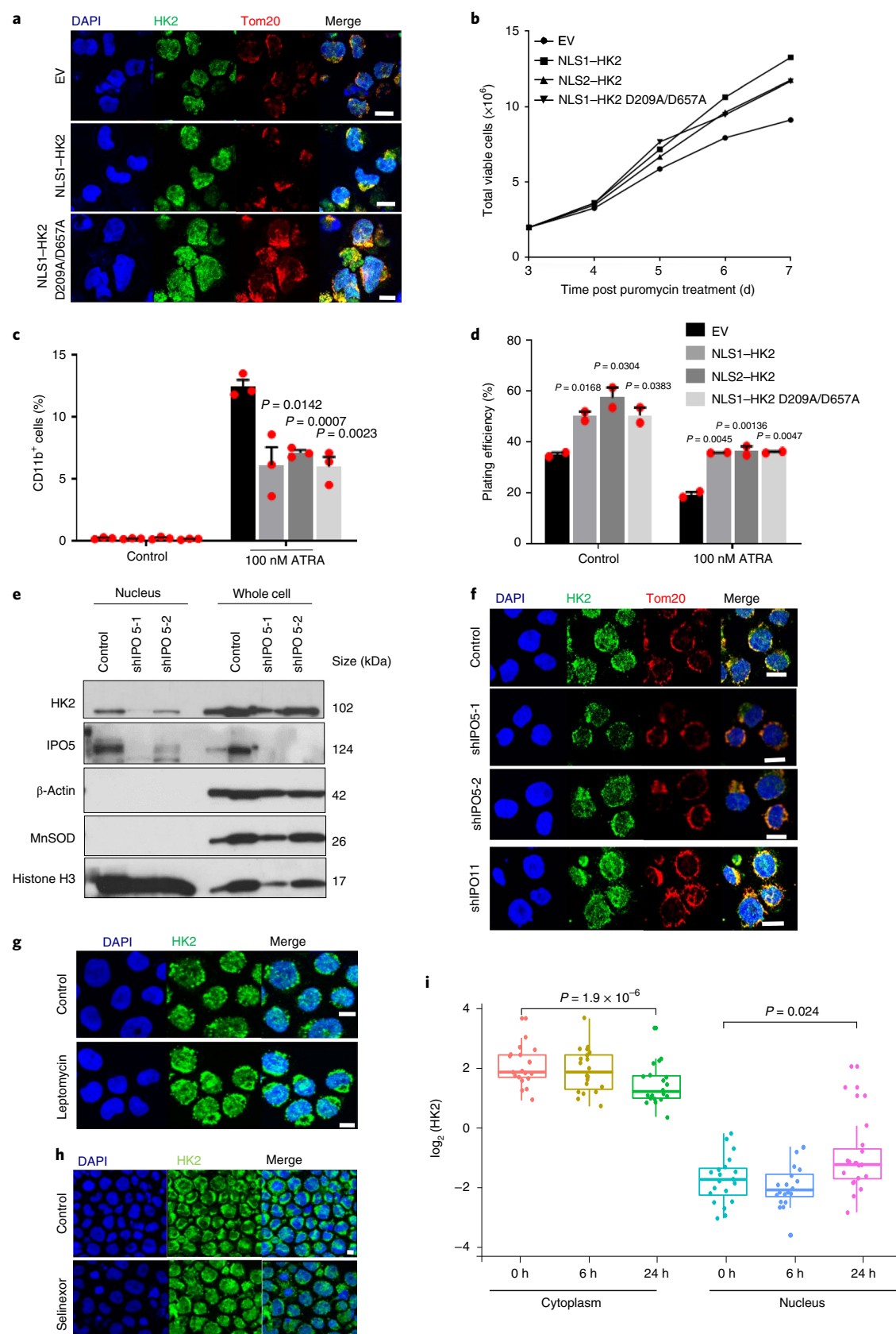

γH2AX foci (Fig. 6b,d,f and Extended Data Fig. 9a). We then examined primary patient samples and discovered that the expression of DNA-repair-pathway genes was upregulated in functionally defined stem cells compared with bulk AML cells (normalized enrichment score, NES = 1.74 and FDR = 0.057), and undifferentiated versus committed AML samples (NES = 2.041 and FDR ≤ 0.00001; Fig. 6j).

**Fig. 4 | Nuclear HK2 maintains stemness independently of its kinase activity and is mediated by active import/export. a**, Confocal microscopy images of HK2 and Tom20 staining in NB4 cells 5 d after transduction with NLS1–HK2, the kinase-dead NLS1–HK2 D209A/D657A mutant or EV. **b**, Cell viability of NB4 cells after transduction with NLS1–HK2, NLS2–HK2 or NLS1–HK2 D209A/D657A; $n = 2$ biological repeats. **c**, Expression of CD11b in NB4 cells transduced with NLS1–HK2, NLS2–HK2 or NLS1–HK2 D209A/D657A and treated with ATRA (100 nM); $n = 3$ biological repeats. **d**, Clonogenic growth of NB4 cells transduced with NLS1–HK2, NLS2–HK2 or NLS1–HK2 D209A/D657A and treated with ATRA (100 nM); $n = 3$ biological repeats. **e**, Immunoblot analysis of HK2 in the nuclear and whole-cell lysates of NB4 cells treated with shRNAs targeting *IPO5*. **f**, Confocal microscopy images of HK2 and Tom20 in NB4 cells after *IPO5* (shIPO5-1 and shIPO5-2; second and third rows) and *IPO11* (shIPO11; bottom) knockdown using shRNA. **g,h**, Representative confocal images of HK2 and Tom20 in NB4 cells treated with the XPO1 inhibitors leptomycin (**g**) and selinexor (**h**). **i**, HK2 expression in the cytoplasmic and nuclear fractions of various leukaemic cell lines ($n = 20$) following treatment with the XPO1 inhibitor KPT185 for 6 or 24 h, determined using RPPA analysis. Cytoplasm 0 h: minimum, 0.6462; maximum, 3.5142; and median, 1.6358. Nucleus 0 h: minimum, −3.5388; maximum, −0.5461; and median −2.1901. Cytoplasm 6 h: minimum, 0.4226; maximum, 3.5298; and median, 1.622. Nucleus 6 h: minimum, −4.130; maximum, −1.025; and median, −2.552. Cytoplasm 24 h: minimum, 0.0185; maximum, 3.1735; and median, 0.9156. Nucleus 24 h: minimum, −3.334; maximum, 1.817; and median, −1.1624. In the box-and-whisker plots, the horizontal lines mark the median, the box limits indicate the 25th and 75th percentiles, and the whiskers extend to 1.5× the interquartile range from the 25th and 75th percentiles. **a,e–h**, Images are representative of three biological repeats. **a,f–h**, Scale bars, 10 μm. **c,d**, Data represent the mean ± s.e.m. **c,d,i**, Statistical analyses were performed using a two-tailed unpaired Student's *t*-test (**c,d**) or a paired Wilcoxon rank-sum test.

We also sorted primary AML samples into functional stem and bulk populations based on ROS expression (ROS-low versus ROS-high), treated these fractions with daunorubicin and measured their DNA-damage response. Similar to the findings in 8227 cells, primary AML stem cells showed an increased DNA-damage response compared with bulk cells (Fig. 6k,l and Extended Data Fig. 10a–d).

Finally, we transduced 8227 cells with NLS1–HK2 and then sorted the transduced cells into stem and bulk populations based on CD34 and CD38 expression. The stem and bulk cells were treated with daunorubicin for 3 h and their DNA-damage repair markers were measured. In the bulk cells, overexpression of nuclear HK2 restored recruitment of the DNA-repair protein 53BP1. In addition, overexpression of nuclear HK2 in stem cells increased the recruitment of 53BP1 compared with EV stem cells (Extended Data Fig. 10e–g). Thus, AML stem cells demonstrate an enhanced DNA-damage response that seems to be partly mediated by nuclear HK2.

## Discussion

Mitochondrial metabolites regulate nuclear gene expression and thereby control stem cell function and differentiation. However, it is less appreciated how mitochondrial enzymes can moonlight in the nucleus to control these properties. In this study we discovered that the mitochondrial enzyme HK2 localizes to the nucleus and impacts the function of AML stem cells.

Metabolites and intermediates produced in the mitochondria serve as cofactors and substrates to modify DNA and histones. Through mitochondrial stress or mutation of mitochondrial enzymes, production of these metabolites can be altered, leading to deregulation of DNA and histone modifications, such as methylation and acetylation. For example, isocitrate dehydrogenase 2 (IDH2) Arg140 or Arg172 mutations are found in approximately 10% of patients with AML. IDH2 mutations result in the aberrant production of R-2-hydroxyglutarate from α-ketoglutarate. R-2-hydroxyglutarate inhibits α-ketoglutarate-dependent dioxygenases, including TET2, resulting in increased DNA methylation and a block in differentiation. Inhibition of mutant IDH2 with enasidenib decreases R-2-hydroxyglutarate production, restores TET2, reduces DNA methylation and relieves the block in myeloid differentiation.

Although most studies have focused on mitochondrial metabolites as regulators of epigenetic marks and gene expression, some previous studies have identified mitochondrial proteins that localize to the nucleus to control gene expression. However, these proteins impact nuclear-gene expression through their known metabolic enzymatic activity. For example, in response to mitochondrial stress, such as respiratory chain complex inhibitors or depletion of the mitochondrial outer membrane protein MTCH2, pyruvate dehydrogenase translocates from the mitochondria to the nucleus. In the nucleus, pyruvate dehydrogenase converts pyruvate to acetyl-CoA, leading to increased histone acetylation, which alters gene expression and promotes differentiation[36,37].

Here we demonstrated that HK2, independently of its known mitochondrial kinase activity, regulates stem cell function. We showed that nuclear HK2 interacts with the bHLH E-Box-binding protein MAX, regulates chromatin accessibility and maintains DNA integrity. In yeast, nuclear HK2 forms a complex with the transcription factor Mig1 to control the expression of the sucrose transporter gene *SUC2* (refs. [22,23,38,39]). However, mammalian cells do not have

**Fig. 5 | Nuclear HK2 modulates chromatin accessibility and is involved in the maintenance of DNA integrity. a**, Proteins that interact with NLS1–HK2, as determined by BioID coupled with mass spectrometry; $n = 3$ biological repeats. **b**, Intensity of the PLA signal of endogenous HK2 and MAX, SIRT1, IWS1, CTR9 and SPIN1 in NB4 cells. A.u., arbitrary units; $n = 278$ cells from three biological repeats. **c**, Chromatin accessibility, measured through ATAC-seq, following overexpression of NLS1–HK2 in NB4 cells; $n = 3$ biological repeats. **d**, LSC⁺ and LSC⁻ signatures in EV control NB4 cells. LSC⁻: minimum, 5.15; maximum, 12.81; and median, 8.27. LSC⁺: minimum, 5.990; maximum, 12.030; and median, 9.325. **e**, LSC⁺ and LSC⁻ signatures in NLS1–HK2 NB4 cells. LSC⁻: minimum, 6.840; maximum, 13.010; and median, 9.020. LSC⁺: minimum, 7.300; maximum, 12.340; and median, 9.895. **f**, ATAC-seq pathway enrichment analysis in EV control and NLS1–HK2 NB4 cells. The size of the pie chart slices is proportional to the FDR score, $-\log_{10}(\text{FDR})$, for each of the gene lists. Blue and pink lines pinpoint to pathways that overlap significantly with the HSC and granulocyte (GRAN) gene lists at FDR < 0.05 according to a Fisher's exact test. **g**, Enhanced enrichment of the pathways in NLS1–HK2 and NLS2–HK2 ChIP–seq compared with EV control ChIP–seq in NB4 cells. **h**, Consensus motif identified by HOMER DNA-binding-motif analysis significantly enriched in NLS1– and NLS2–HK2 peaks at FDR < 0.05. The *P* value shows significant enrichment of bHLH motifs, determined using a Fisher's exact test. The boxes with the dashed lines represent sequence similarity overlap. **i**, Enrichment of the consensus motif CACGTG in sequences of peaks associated with the selected pathways. The hypergeometric *P* value estimated using the Fisher's exact test indicate the significance of the enrichment of the motif when compared with random sequences. **c–e**, Statistical analyses were performed using a two-tailed unpaired Student's *t*-test. Data represent the mean ± s.e.m. **d,e**, In the box-and-whisker plots, the horizontal lines mark the median, the box limits indicate the 25th and 75th percentiles, and the whiskers extend to 1.5× the interquartile range from the 25th and 75th percentiles.

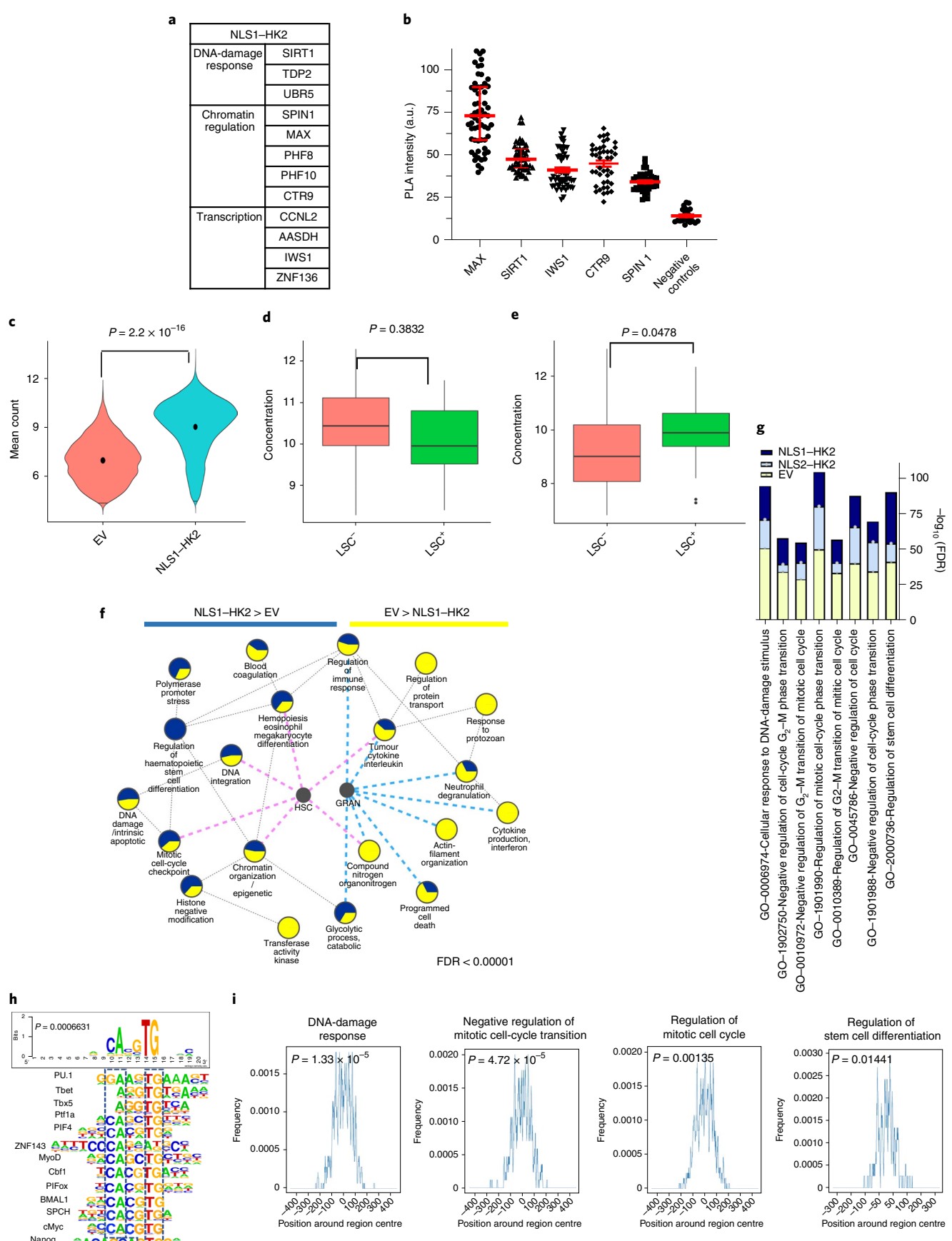

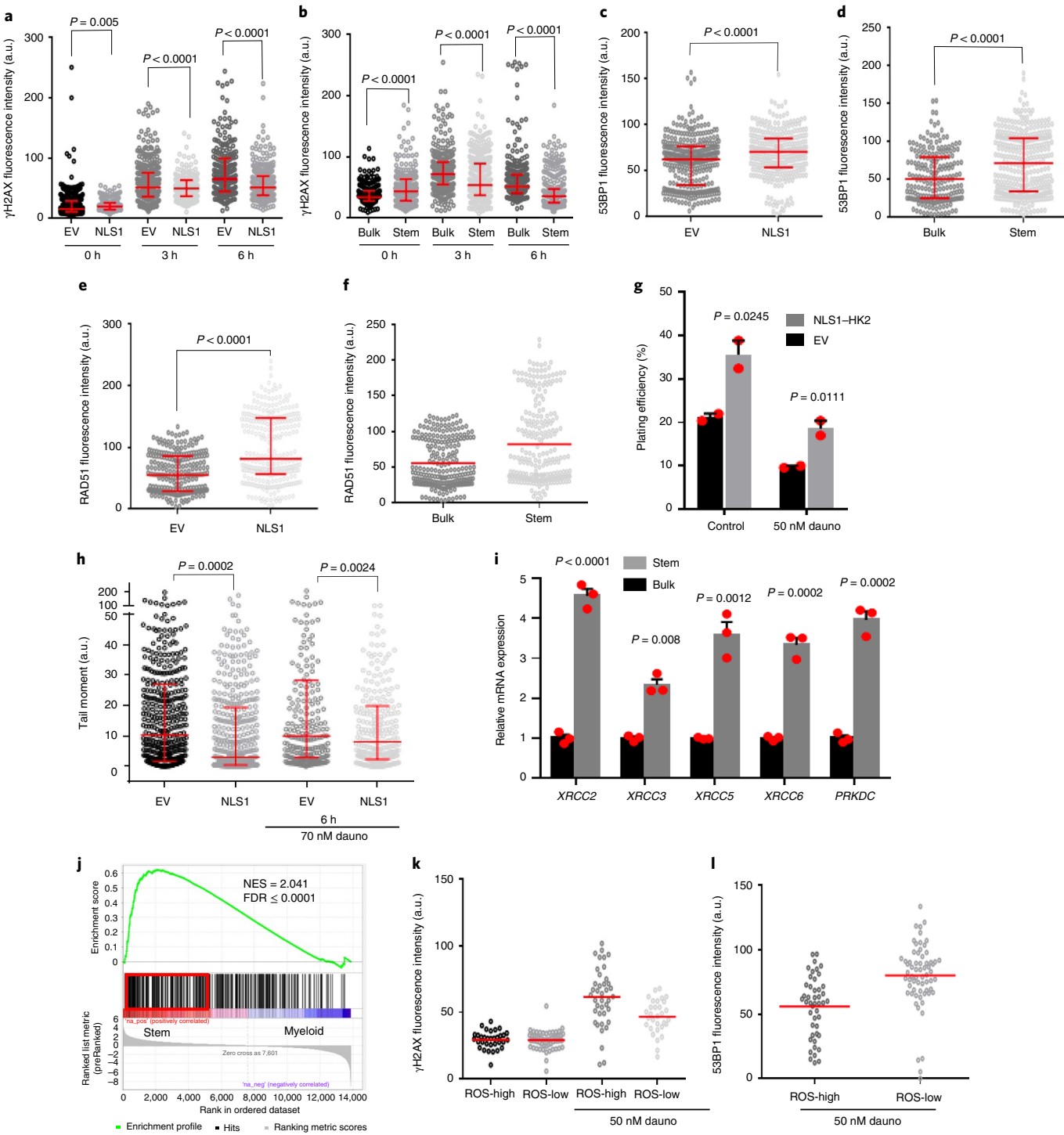

a Mig1 homologue. A recent study has shown that in glioma cells under metabolic stress, nuclear HK2 activates nuclear factor erythroid 2-related factor 2, a transcription factor that provides protection against oxidative stress[40].

Proteins are imported into the mitochondria through the TOM and TIM pathway of receptors and channels. However, how proteins translocate from the mitochondria and enter the nucleus is poorly understood. We discovered that nuclear HK2 is influenced by its phosphorylation state and requires active import and export via IPO5 and XPO1, respectively. In the future, it will be important to comprehensively map the pathways that control mitochondrial protein translocation and export. In addition, future studies will

determine how HK2 preferentially accumulates in stem cells compared with more differentiated cells.

We discovered that nuclear HK2 promotes stemness in AML and normal haematopoietic cells in vitro and in vivo. Using a transgenic mouse model, we showed that overexpression of nuclear HK2 increases the abundance of HSCs. In mouse models of AML, selective knockdown of nuclear HK2, decreased the engraftment of TEX cells into the mouse marrow. Although these cells can engraft the marrow of immune-deficient mice, they do not circulate in the blood after engraftment, so we could not measure peripheral blood levels of leukaemia in this model. Similarly, mice engrafted with AML cells overexpressing nuclear HK2 had increased marrow engraftment and

**Fig. 6 | Nuclear HK2 overexpression enhances the DNA-damage response and increases chemoresistance in AML. a**, Levels of γH2AX in NLS1–HK2 and EV control 8227 cells after treatment with 50 nM daunorubicin for 0, 3 and 6 h; n = 2,041 cells from four biological repeats were examined. **b**, Levels of γH2AX in stem and bulk 8227 cells after treatment with 50 nM daunorubicin for 0, 3 and 6 h; n = 1,786 cells from three biological repeats were examined. **c**, Levels of 53BP1 in NLS1–HK2 and EV control 8227 cells after treatment with 50 nM daunorubicin for 3 h; n = 645 cells from four biological repeats were examined. **d**, Levels of 53BP1 levels in stem and bulk 8227 cells after treatment with 50 nM daunorubicin for 3 h; n = 549 cells from three biological repeats were examined. **e**, Levels of RAD51 in NLS1–HK2 and EV control 8227 cells after treatment with 50 nM daunorubicin for 6 h; n = 551 cells from three biological repeats were examined. **f**, Levels of RAD51 in stem and bulk 8227 cells after treatment with 50 nM daunorubicin for 6 h; n = 477 cells from two biological repeats were examined. **g**, Clonogenic growth of NB4 cells transduced with NLS1–HK2 and treated with 50 nM daunorubicin for 3 h before plating. The colonies were counted 6 d after plating; n = 3 biological repeats. **h**, Comet assay in 8227 cells transduced with NLS1–HK2 before and after incubation with 70 nM daunorubicin for 6 h; n = 1,474 cells from three biological repeats were examined. **i**, Relative messenger RNA expression levels of genes associated with homologous recombination (*XRCC2* and *XRCC3*) and non-homologous end joining (*XRCC5*, *XRCC6* and *PRKDC*) in stem and bulk 8227 cells; n = 3 biological repeats. **j**, GSEA analysis of DNA-repair pathways in primary samples from patients with undifferentiated versus committed AML. The NES and FDR values were analysed using a modified Kolmogorov–Smirnov test. **k**, Levels of γH2AX in stem and bulk fractions of primary sample from a patient with AML (AML151258) after treatment with 50 nM daunorubicin for 0 and 3 h; n = 225 cells were examined from one of two biological samples. **l**, Levels of 53BP1 in the stem and bulk fractions of a primary sample from a patient with AML (AML151258) following treatment with 50 nM daunorubicin for 3 h; n = 111 cells were examined from one of two biological samples. **a–f,k,l**, The protein levels were determined using confocal microscopy. Statistical analyses were performed using a two-tailed unpaired Student's *t*-test in all panels except **j**, where a Fischer's exact *t*-test was performed. Data represent the median and interquartile range (**a–e,h**), the mean (**f,k,l**) or the mean ± s.e.m. (**g,i**). Dauno, daunorubicin; a.u., arbitrary units.

decreased survival compared with the controls (*P* = 0.054). It is important to note that in these survival studies, the mice were euthanized at the first sign of distress. However, the correlation between distress and engraftment is not exact. Similar to the clinical situation[41,42], mice can display distress with variable amounts of leukaemia in the marrow, potentially explaining the smaller difference in overall survival.

Research by our group and others has shown that LSCs have more open chromatin compared with bulk cells, which primes the cell for determination of fate and differentiation[31]. This open chromatin structure can enhance DNA-damage repair by increasing access to DNA-repair proteins[43]. We demonstrated that LSCs have a heightened DNA-damage response compared with bulk cells—a potential explanation for enhanced chemoresistance in LSCs. We showed that HK2 interacts with DNA damage-response proteins and overexpression of nuclear HK2 decreases the levels of double-strand DNA breaks and increases chemoresistance. Thus, in addition to participating in chromatin accessibility, HK2 positively influences the DNA-damage response.

Thus, in conclusion, we discovered a non-canonical mechanism by which a mitochondrial enzyme interacts with nuclear proteins to regulate stem cell function and differentiation. In addition, we highlight the role of mitochondrial enzymes in essential nuclear processes including chromatin accessibility and the DNA-damage response.

## Online content

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

## Methods

All experiments performed in this study were approved by the University Health Network ethical review committee. Further information on research design, statistics and technical information is available in the Nature Research Reporting Summary linked to this article.

**Cell lines.** OCI-AML2 cells were cultured in Iscove's modified Dulbecco's medium containing penicillin (100 U ml$^{-1}$) and streptomycin (100 µg ml$^{-1}$; Wisent) supplemented with 10% fetal bovine serum (FBS). NB4 and U937 cells were cultured in Roswell Park Memorial Institute 1640 medium containing penicillin (100 U ml$^{-1}$) and streptomycin (100 µg ml$^{-1}$; Wisent) supplemented with 10% FBS. TEX leukaemia cells obtained from J.E.D.'s laboratory[44] were maintained in Iscove's modified Dulbecco's medium with 20% FBS, 2 mM L-glutamine (Thermo Fisher Scientific, 25030081), human recombinant stem cell factor (SCF; 20 ng ml$^{-1}$; R&D Systems, 255-sc) and human recombinant interleukin (IL)-3 (2 ng ml$^{-1}$; R&D Systems, 203-IL). OCI-AML 8227 cells, obtained from J.E.D.'s laboratory[45], and 130578 cells, obtained from S. M. Chan's laboratory[16], were cultured in X-VIVO 10 medium (Lonza, 04-380Q) with 20% BSA–insulin-transferrin (BIT 9500 serum; Stem Cell Technologies, 09500) and growth factor cocktail (human Fms-related tyrosine kinase 3 ligand (Flt3-L; 50 ng ml$^{-1}$; PeproTech, 300-19), IL-6 (10 ng ml$^{-1}$; PeproTech, 200-06), SCF (50 ng ml$^{-1}$; PeproTech, 300-07), thrombopoietin (25 ng ml$^{-1}$; PeproTech, 300-18), IL-3 (10 ng ml$^{-1}$; PeproTech, 200-03) and granulocyte colony-stimulating factor (G-CSF; 10 ng ml$^{-1}$; Amgen, 121181-53-1)). The lentiviral packing cells (HEK 293T; American Type Culture Collection, CRL-11268) were cultured in DMEM medium with 10% FCS for seeding and DMEM with 10% FCS, penicillin (100 U ml$^{-1}$), streptomycin (100 µg ml$^{-1}$) and 1% BSA for harvesting of virus. All cell lines were maintained in humidified incubators at 37 °C supplemented with 5% $CO_2$.

**Primary AML and normal haematopoietic cells.** Primary human AML samples from peripheral blood or the bone marrow of both male and female patients with AML were collected after obtaining informed consent and cryopreserved at the Leukemia Tissue Bank at the Princess Margaret Cancer Centre/University Health Network (REB no. 01-0573). No compensation was provided. Ficoll-Paque differential density centrifugation was used to isolate AML cells. The primary AML cells were frozen in 50% FCS + 40% αMEM + 10% dimethylsulfoxide. The University Health Network institutional review board approved the collection and use of human tissue for this study (Research Ethics Board protocol no. 13-7163). As per regulation, all specimens were de-identified. Each experiment was performed using a single aliquot from a donor. G-CSF-mobilized peripheral blood was obtained from stem cell donations with written consent from the donors and in accordance with the ethical standards of the responsible committee on human experimentation (IRB permit 329/10). Cord blood cells were purchased from Stem Cell Technologies (cat no. 70007).

**Cell sorting.** 8227 cells were stained with CD34–APC-Cy7 (BioLegend) and CD38–PE-Cy7 (BD Biosciences) and washed with PBS before sorting CD34$^+$CD38$^-$ and CD34$^-$CD38$^+$ cells. Cells were FACS-sorted on a MoFlo system (Beckman Coulter). Primary AML specimens were stained with BB515–CD45 (BD Biosciences, 564585) to identify the blast population, PE–CD19 (BD Biosciences, 555413) and PE-Cy7–CD3 (BD Biosciences, 557749) to exclude the lymphocyte populations, 4,6-diamidino-2-phenylindole (DAPI; EMD Millipore, 278298) as a dead-cell marker and CellROX deep red (Thermo Fisher Scientific, C10422), and sorted using a Sony SH-800 system. ROS-low LSCs were identified as the cells with the 20% lowest ROS levels and the ROS-high blasts were identified as the cells with the highest 20% ROS levels, as recently described in detail[14].

**Nuclear isolation.** To isolate nuclei from cell lines and primary patient samples, 6–8 × 10$^6$ cells were washed in PBS (pH 7.4), resuspended in 750 µl nuclear isolation buffer (Sigma Aldrich, NUC-101), incubated on ice for 5 min and centrifuged at 500$g$ for 10 min at 4 °C. The supernatant was collected as the cytoplasmic fraction and then washed and centrifuged with lysis buffer. The supernatant from the second centrifugation was discarded and the pellet was resuspended in radioimmunoprecipitation assay buffer (RIPA) buffer with protease inhibitors. The protein concentration was determined using the Bradford Assay (Bio-Rad).

**Immunoblotting.** Nuclear and cytoplasmic isolates or total cell lysates from cell lines or primary patient samples were lysed using RIPA buffer and the Bradford assay (Bio-Rad) was used to determine the protein concentration. Equal amounts of protein were run on 10–12% SDS–PAGE gels and transferred to polyvinylidene difluoride membranes. The membranes were blocked with 5% milk or BSA in Tris-buffered saline with Tween-20 for 1 h and then incubated overnight with primary antibody dissolved in 5% milk in Tris-buffered saline with Tween-20 at 4 °C. The membranes were washed three times before incubation for 1 h at room temperature with secondary horseradish peroxidase-conjugated donkey anti-rabbit antibody (GE Healthcare, NA934) or sheep anti-mouse antibody (GE Healthcare, NA931). The primary antibodies used were (additional details are provided in the Nature Research Reporting Summary): anti-HK2 (Cell Signaling Technology, 2867; 1:1,000), anti-β-actin (Santa Cruz Biotechnology, sc-69879; 1:10,000), anti-histone 3 (Cell Signaling Technology, 4499; 1:5,000), anti-MnSOD

(Enzo, ADI-SOD-110-F; 1:3,000), anti-Flag M2 (Sigma, F3165; 1:500), anti-aldolase A (Santa Cruz Biotechnology, sc-390733; 1:500), anti-aconitase 2 (Cell Signaling Technology, 6922; 1:1,000), anti-citrate synthase (Abcam, ab129095; 1:1,000), anti-enolase (Cell Signaling Technology, 3810; 1:1,000), anti-GAPDH (Cell Signaling Technology, 2118; 1:1,000), anti-PFKP (Cell Signaling Technology, 5412; 1:500), anti-PGK (Abcam, ab113687; 1:1,000), anti-PKM2 (Cell Signaling Technology, 3198; 1:1,000), anti-GPI (Thermo Fisher Scientific, PA5-29665; 1:1,000), anti-hexokinase I (Cell Signaling Technology, 2024; 1:1,000), anti-hexokinase III (Abcam, ab91097; 1:1,000), anti-SDHA (Abcam, ab14715; 1:1,000) and anti-SDHB (Abcam, ab178423; 1:1,000).

**Confocal microscopy.** Glass slides were prepared by spinning down the cells using cytospin. The cells were fixed with 4% paraformaldehyde and permeabilized with 4% BSA in PBS with 0.1% Triton X-100. The cells were then stained with primary antibodies and fluorochrome-conjugated secondary antibodies for 1 h and 45 min, respectively. Tom20 (BD Biosciences, 612278; 1:400) was used as a marker for mitochondria; HK2 (Cell Signaling Technology, 2867; 1:200) and the nuclei were stained with DAPI (5 µg ml$^{-1}$). Antibodies to RAD51 (Abcam, ab63801; 1:600), γH2AXSer139 (EMD Millipore, 05-636; 1:300) and 53BP1 (Novus Biologics, NB100-304; 1:600) were used for DNA-damage analysis. Images were taken on a Leica SP8 or Olympus confocal microscope at ×60 magnification. Intensity and region-of-interest analyses were performed using Metamorph and ImageJ.

**RPPA.** Samples were collected from 25 patients with AML as well as from 15 AML cell lines. The MD Anderson Cancer Center Institutional Review Board approved the collection protocol, research usage profile and clinical protocols that these patients were treated with. The cell lines were cultured according to the American Type Culture Collection cell culture methods. Cells were collected and fractionated using a NE-PER nuclear and cytoplasmic extraction reagent kit (Thermo Fisher Scientific, 78835). The cell lines were treated with the XPO1 inhibitor KPT185 at 200 µg ml$^{-1}$ for 6 and 24 h. The protein expression levels of samples from the patients with acute leukaemia as well as the cell lines were determined using RPPA analysis. The methods and antibody validation techniques have been fully described in previous publications[46–48]. Briefly, the samples were printed onto slides in five (1:2) serial dilutions along with normalization and expression controls. The slides were probed with 298 validated primary antibodies, including an antibody to HK2 (Cell Signaling Technology, 2867; 1:100) and a secondary antibody to amplify the signal. The stained slides were analysed using the Microvigene software (Vigene Tech) to produce quantified data. SuperCurve algorithms were used to generate a single value from the five serial dilutions[49]. Loading controls[50] and topographical normalization[51] procedures were performed to account for variations in protein concentration and background staining. The HK2 expression levels were compared between pre- and post-treatment samples using the paired Wilcoxon rank-sum test. Plots were generated using R (version 1.3.959, 2009–2020, RStudio, Inc.).

**Nuclear HK2 overexpression.** For experiments overexpressing nuclear HK2, either the c-Myc (NLS1, PAAKRVKLD) or SV40 (NLS2, PKKKRKV) NLS was cloned in-frame to the human HK2 open reading frame (NM_000189.5) and subcloned into the pLentiEF1α vector (blasticidin antibiotic resistance). The empty pLentiEF1α vector served as a control. Lentiviral infections were performed as per protocol[52]. NB4 cells were seeded in T25 flasks (3 × 10$^6$ cells per flask). The culture was supplemented with protamine sulfate (1 µg ml$^{-1}$). The cells were transduced with pLentiEF1α, pLentiEF1α-NLS1–HK2 or pLentiEF1α-NLS2–HK2 viral stock, followed by overnight incubation (37 °C and 5% $CO_2$). The transduced cells were then selected by resuspending them in medium containing blasticidin (7.5 µg ml$^{-1}$).

**Transduction of 8227 and CD34-enriched cells.** CD34-enriched cord blood cells and 8227 cells were transduced with pLBC2-BS (from J.E.D.'s laboratory) vectors containing NLS1–HK2 driven by the SFFV promoter and BFP driven by a chimaeric EF1α/SV40 promoter. Transduction of 8227 cells and CD34-enriched cord blood cells was performed as described in a recent publication[53]. Twenty-four-well plates were coated with retronectin for 2 h at room temperature. The plates were then blocked with 2% BSA (wt/vol) for 30 min at room temperature. After the removal of BSA, 0.5 ml of concentrated virus particles in HBBS along with 25 mM HEPES was added to each well and the plates were centrifuged at 1,600$g$ for 5 h at room temperature to aid in the attachment of viral particles. After centrifugation, the viral particle solution was removed and 5 × 10$^5$ cells were added to each well in 1 ml X-VIVO 10 medium supplemented with 20% BIT 9500 serum substitute (Stem Cell Technologies) and growth factor cocktail. The plates were centrifuged again at 600$g$ for 10 min and transferred to a 37 °C incubator for 24 h. The cells were then resuspended in fresh medium at a concentration of 1 × 10$^6$ cells ml$^{-1}$ and seeded in 24-well plates (1 ml per well). After an additional 3 d, the transduction efficiency (percentage of BFP$^+$ cells) was determined by flow cytometry and expression was confirmed by confocal microscopy.

**Animals.** Immunodeficient NOD.Cg-Prkdc$^{scid}$ Il2rg$^{tm1Wjl}$ Tg(CMV-IL-3, CSF2,KITLG)1Eav/MloySzJ (NOD-SCID-GF) mice were obtained from C. J. Eaves and bred in our facility[54]. No statistical methods were used to

pre-determine the sample sizes but our sample sizes are similar to those reported in previous publications[31]. The mice were housed in micro isolator cages with temperature-controlled conditions under a 12-h light–dark cycle with access to drinking water and food. Only one experimental procedure was performed on each mouse and all mice were drug naive before the experiment. For the in vivo experiments with mice, the mice were grouped before treatment. The grouping and treatment of the mice was performed by an individual who was not involved in the analysis of the data from the experiment. Furthermore, all animal studies were performed in accordance with the Ontario Cancer Institute Animal Use Protocol (AUP) no. 1251.38 (NOD-SCID-GF and SCID mice).

**Mouse engraftment, tumour progression and survival analysis.** CD34[+]-enriched cord blood cells and 8227 cells were injected into the right femur of sublethally irradiated NOD-SCID-GF mice (male or female, 1:1 ratio; 5–6 weeks old) with human IL-3, GM-CSF and Steel factor[54]. The mice were killed 8 weeks after injection, the percentages of human BFP[+]CD45[+] cells were enumerated by flow cytometry and then human-cell engraftment was calculated, as described in earlier studies[31,37,55]. TEX cells were injected in the right femur of sublethally (2 Gy) irradiated NOD-SCID-GF mice[54] (male or female, 1:1 ratio). The mice were killed 5–6 weeks after injection and engraftment of human CD45 cells was measured in the left femur. OCI-AML2 cells were injected in the back flaps of male SCID mice and tumour volume was measured every 2–3 d. At the end of experiment, on day 21, the mice were euthanized and the tumours were weighed.

**Flow cytometry.** We co-stained 8227 cells with CD34–FITC (BD Biosciences) and CD38–PE (BD Biosciences). To measure differentiation in NB4 cells, the cells were stained with anti-CD11b (BD Biosciences, 340937). Flow cytometry data were acquired using a Fortessa X-20 system (BD Biosciences). Primary and secondary engraftment of AML was measured by staining for CD45–FITC (BD Biosciences, 347463) and data were acquired using a FACSCANTO II system (BD Biosciences). The data were analysed post acquisition using the FlowJO software (v7.7.1 and v10.7.1; Becton, Dickinson and Company; 2019).

**Colony-formation assays.** NB4 cells transduced with NLS1–HK2 or NLS2–HK2 were plated in duplicate 35-mm dishes (Nunclon) to a final volume of 1 ml per dish in MethoCult H4100 medium (StemCell Technologies) supplemented with 30% FBS. After incubation for 7–8 d at 37 °C and 5% $CO_2$ with 95% humidity, the number of colonies containing ten or more cells was counted on an inverted microscope. The plating efficiency was determined by counting the number of colonies that formed: (number of colonies formed ÷ number of cells inoculated) × 100.

**Cell growth and viability assays.** To assess the impact of 2-DG or olaparib on the growth and viability of the HK2 clones, MTS colorimetric assays were used. CellTiter 96 AQueous MTS reagent powder was purchased from Promega (cat. no. G1111). Following exposure to increasing concentrations of olaparib, the cells were incubated at 37 °C for 72 h. MTS solution (20 μl) was added to each sample (100 μl) in a 96-well plate (for a final MTS concentration of 0.33 mg ml[−1]). Cell viability was then assessed after 3 h of incubation at 37 °C by recording absorbance at a wavelength of 490 nm.

**Selective knockdown of nuclear *HK2*.** To selectively knockdown nuclear *HK2*, we overexpressed a mitochondria-tethered HK2 by fusing an OMMLS from OPA25 to the C terminus of human HK2 (OMMLS–HK2) and subcloned this construct into the pLentiEF1α vector. NB4 cells were transduced with pLentiEF1α OMMLS–HK2 viral particles, followed by overnight incubation. The following day, the cells were transduced with shRNA targeting a control sequence (GFP) or the 3′ UTR of *HK2* in a pLKO.1 vector (carrying the puromycin antibiotic-resistance gene). The transduced cells were cultured in selection medium containing both blasticidin (7.5 μg ml[−1]) and puromycin (1.5 μg ml[−1]). The 3′ UTR *HK2* shRNA sequences used in the study were 5′-CCGGCTTAGGGCAGTCAGTAGTATTCTCGAGAATAC-TACTGACTGCCCTAAGTTTTTG-3′ and 5′-CCGGCCAAAGACATCTCAGA-CATTGCTCGAGCAATGTCTGAGATGTCTTTGGTTTTTTG-3′.

**RNA sequencing.** RNA was isolated from 8227 OMMLS control and nuclear *HK2*-knockdown cells using an RNeasy plus mini kit (Qiagen). The quality of the total RNA samples was checked on an Agilent Bioanalyzer 2100 RNA Nano chip following Agilent Technologies' recommendation. The RNA concentration was measured using a Qubit RNA HS assay on a Qubit fluorometer (Thermo Fisher Scientific). The RNA library preparation was performed following the NEB next ultra directional library instructions.

*Differential gene expression analysis.* Before analysis, read adaptors and low-quality ends were removed using Trim Galore v. 0.4.0. Reads were aligned against hg38 using Tophat v. 2.0.11. Read counts per gene were obtained through htseq-count v. 0.6.1p2 in the mode 'intersection nonempty'. After removing genes whose counts per million reads were less than 0.5 in at least one sample, edgeR R package v. 3.16.5 was used to normalize the data using the trimmed mean of $M$ values method and to estimate differential expression by applying the generalized linear model likelihood ratio test between the control and knockdown. A score that ranks

genes in nuclear HK2-knockdown samples from the most upregulated to the most downregulated compared with control shRNA samples was calculated using the formula $-\log_{10}(P \text{ value}) \times \text{sign}(\log(\text{fold change}))$.

*Single-sample gene set enrichment analysis using LSC[+] or LSC[−] gene signatures.* Single-sample gene set enrichment analysis was run using the R package GSVA 1.30.0 using normalized counts per million and the LSC[+] or LSC[−] signatures as reference gene sets. Illumina beadchip transcriptomics data containing LSC[+]- and LSC[−]-sorted AML fractions were obtained from the Gene Expression Omnibus data portal (GSE76008)[19] and differential expression between the LSC[+] and LSC[−] fractions was calculated using a moderated *t*-test available in the limma R package 3.28.21 incorporating array batch effects in the linear model. A score that ranks genes from the top upregulated in the LSC[+] fractions to the top downregulated when compared with the LSC[−] fractions was calculated using the formula $-\log_{10}(P \text{ value}) \times \text{sign}(\log(\text{fold change}))$. The top-100 upregulated genes were used as the LSC[+]-specific gene list and the top-100 downregulated genes were used as the LSC[−] gene list. The gene expression data have been deposited in the Gene Expression Omnibus database under the accession code GSE176103.

**FACS of human stem and progenitor or mature cell populations.** Mononuclear cells ($1 \times 10^6$ per 100 μl) from cord blood were stained with the following antibodies (all from BD Biosciences, unless stated otherwise; the dilution used and catalogue numbers are in parentheses): anti-CD45RA–FITC (1:25; 555488), anti-CD90–APC (1:50; 561971), anti-CD135–Biotin (1:10; 624008), anti-CD38–PE-Cy7 (1:200; 335790), anti-CD10–Alexa Fluor 700 (1:10; 624040), anti-CD7–Pacific Blue (1:50; 642916), anti-CD45–V500 (1:200; 560777), anti-CD34–APC-Cy7 (1:100), anti-CD34–PerCP-Efluor 710 (1:100; eBioscience, 46-0344-42), anti-CD33–PE-Cy5 (1:100; Beckman Coulter, PNIM2647U), anti-CD19–PE (1:200), anti-CD3–FITC (1:100; 349201), anti-CD56–Alexa Fluor 647 (1:100; 557711) and Streptavidin–QD605 (1:200; Invitrogen, Q10101MP). Cells were sorted on a FACS Aria III system (BD Bioscience).

**CD34[+] haematopoietic stem and progenitor cell enrichment from cord blood.** CD34[+] haematopoietic stem and progenitor cells were enriched from freshly thawed cord blood samples by magnetic separation using CD34 microbeads (Miltenyi Biotec) as per the manufacturer's protocol and cultured in X-VIVO 10 (Lonza) medium with 20% BIT 9500 serum substitute (Stem Cell Technologies) and growth factor cocktail.

**Generation of transgenic lines.** The NLS–HK2 complementary DNA was cloned into the HS21/45-Vav vector[56], provided by P. D. Aplan. A fragment containing 5′ and 3′ Vav regulatory sequences with the cDNA was subsequently isolated and the construct was microinjected into zygotes obtained from C57BL6 mice. Founders were identified by Southern blot analysis using a human Vav-NLS–HK2 probe; the offspring were genotyped by PCR amplification of the transgene from tail-biopsy DNA. The lines were maintained by mating with wild-type C57BL6 (AUP, cat. no. 2244.16). Expression of NLS–HK2 in the transgenic mice was confirmed by western blotting and confocal microscopy of the bone marrow cells. The body weight and length of the mice were monitored and complete blood counts were analysed and enumerated using the HEMAVET 950FS system (Drew Scientific Inc.).

Bone marrow cells were harvested from both femurs by flushing with Iscove's modified Dulbecco's medium. Flow cytometry was used to determine the immunophenotype of a single-cell suspension prepared from the red-blood-cell-lysed bone marrow cells. The cells were stained with anti-haematopoietic lineage cocktail (Invitrogen, 22-7770-72), anti–mouse c-kit (BioLegend, 105814), anti–mouse Ly-6A/E (Sca-1) (Invitrogen, 15-5981-82), anti–mouse CD48 (eBioscience, 17-0481-82), anti–mouse CD150 (BioLegend, 115904), anti–mouse CD34 (eBioscience, 48-0341) and anti–mouse CD16/32 (BioLegend, 101328).

**Competitive repopulation assay.** C57BL/6-CD45.1 B6.SJL-Ptprca Pepcb/BoyJ (B6/SJL) mice were obtained from Jackson Laboratories. Bone marrow competitive repopulation assays were performed using the CD45.1 and CD45.2 congenic system, as described earlier[57]. Briefly, bone marrow cells from donor mice (CD45.2[+]; C57B6 background; Vav-NLS–HK2 or wild-type) and clonogenic competitor mice (CD45.1[+]; B6.SJL) were mixed in a 1:1 ratio and injected into the tail veins of irradiated recipient mice (CD45.1 B6.SJL). The mice received antibiotic-containing water for 2 weeks after irradiation. The efficacy of bone marrow reconstitution was determined in their peripheral blood at 4, 6, 8 and 10 weeks post transplantation and in the bone marrow 12 weeks after transplant.

**BioID.** NLS1–HK2 cDNA was cloned in-frame with BirA* fused to NLS1 into a tetracycline-inducible pcDNA5 FLP recombinase target/tetracycline operator (FRT/TO) expression vector, which was then transfected into Flp-In T-REx HEK293 cells. The cells were collected and pelleted (800 g for 3 min), the pellet was washed twice with PBS and the dried pellets were snap frozen. The pellets were lysed in 10 ml of modified RIPA lysis buffer at 4 °C for 1 h and then sonicated (30 s at 35% power; Sonic Dismembrator 500, Fisher Scientific) to disrupt the visible aggregates. The lysate was centrifuged at 35,000 g for 30 min.

The clarified supernatants were incubated with 30 l packed, pre-equilibrated streptavidin-Sepharose beads (GE) at 4 °C for 3 h on an end-over-end rotator. The beads were collected (800 g for 2 min) and washed six times with 50 mM ammonium bicarbonate (pH 8.3). The beads were then treated with L-1-tosylamide-2-phenylethyl chloromethyl ketone–trypsin (Promega) for 16 h at 37 °C on an end-over-end rotator. After 16 h, another 1 μl of L-1-tosylamide-2-phenylethyl chloromethyl ketone–trypsin was added and the sample was incubated in a water bath at 37 °C for 2 h. The supernatants were lyophilized and stored at 4 °C for downstream mass spectrometry analysis. Two biological and two technical replicates were completed and the NLS1-HK2 interactors were normalized to the BirA* spectral counts.

*Mass spectrometry data analysis.* For peptide and protein identification, Thermo RAW files were converted to .mzML format using ProteoWizard (v3.0.10800) and then searched using X! Tandem (Jackhammer TPP v2013.06.15.1)[58] and Comet (v2014.02 rev. 2)[59] against the human RefSeq v45 database (containing 36,113 entries). The search parameters specified a parent ion-mass tolerance of 10 ppm and a tandem mass spectrometry fragment-ion tolerance of 0.4 Da, with up to two missed cleavages allowed for trypsin (excluding Lys and Arg-Pro). Variable modifications included deamidation on Asn and Gln, oxidation on Met, diglycine on Lysine and acetylation on the protein N terminus in the search. Data were filtered through the trans-proteomic pipeline (v4.7 POLAR VORTEX rev 1) with general parameters set as –p0.05 -x20 –PPM. Proteins were identified with an iProphet cutoff of 0.9 and at least two unique peptides were analysed using Significance Analysis of Interactome Express (v. 3.6)[60,61]. Control runs (21 runs from cells expressing the Flag–BirA* epitope tag only) were collapsed to the two highest spectral counts for each prey and high-confidence interactors were defined as those with a Bayesian FDR of ≤0.01. ProHits-viz was used for baitbait Pearson's correlations and heatmap generation.

**PLA.** The PLA assay[28] was performed according to the manufacturer's protocol (Sigma). Cells were fixed with 4% paraformaldehyde, blocked and permeabilized with 3% BSA and 0.1% Triton X-100 for 30 min before the addition of primary antibodies. Antibodies to the following were used: HK2 (Santa Cruz Biotechnology, sc-374091; 1:100), MAX (Santa Cruz Biotechnology, sc-197; 1:200), SPIN1 (Cell Signaling Technology, 89139; 1:100), CTR9 (Cell Signaling Technology, 12619; 1:100) and IWS1 (Cell Signaling Technology, 5681; 1:100). Images were taken on a Leica SP8 confocal microscope at a magnification of ×60. Data were quantified as the PLA punctate intensity per nuclei.

**ATAC-seq.** *ATAC-seq library preparation.* Control, NLS1–HK2 or nuclear *HK2*-knockdown NB4 cell samples were prepared as described[62]. Briefly, 60,000 viable cells per sample were pelleted at 500 g for 5 min at 4 °C. The supernatant was removed and the cells were resuspended in 50 μl of cold resuspension buffer containing 0.1% NP-40, 0.1% Tween-20 and 0.01% digitonin. The cell suspension was incubated on ice for 3 min before being washed with 1 ml of cold resuspension buffer containing 0.1% Tween-20. The cells were mixed by inversion before pelleting at 500 r.c.f. for 5 min at 4 °C. The supernatant was removed, and the cells were resuspended in 50 μl of transposition mix and incubated at 37 °C for 1 h in a thermomixer (Eppendorf) set to 600 g. The transposition mix was purified using a Qiagen MinElute reaction clean-up kit and eluted in water (20 μl). A 1-μl volume from each sample was used for quantitative PCR (qPCR) to determine the optimal number of PCR cycles required for amplification without reaching saturation; based on the measured cycle number, the remaining 19 μl were amplified. Libraries were purified using AMPure XP beads (Beckman Coulter) using a double-sided bead clean-up protocol set to 0.7–1.0×. This clean-up was performed twice to remove the large-molecular-weight fragments. The purified libraries were evaluated for enrichment by qPCR using primers designed against open regions (*KAT6B* and *GAPDH*) compared against closed regions (*QML93* and *SLC22A3*). Samples that had a fold enrichment greater than ten were sequenced.

*Sequencing.* The libraries were quantified by qPCR, normalized and pooled to 1.25 nM. Each 1.25-nM pool was denatured using 4 μl of 0.2 N NaOH (Sigma) for 8 min at room temperature before being neutralized with 5 μl of 400 mM Tris–HCl (Sigma). Library pools were mixed with Illumina's XP master mix and loaded immediately onto a NovaSeq 6000 S1 flow cell. The samples were sequenced with the following run parameters: read 1, 50 cycles; read 2, 50 cycles; index 1, 8 cycles; and index 2, 0 cycles.

*Analysis.* The ATAC samples were pre-processed according to the ENCODE ATAC-seq pipeline. Single-end reads were aligned to the hg38 genome using Bowtie2 (ref. [63]) with the local parameter, reads with MAPQ scores <30 were filtered out using Samtools[64], duplicates were removed using Sambamba[65] and TN5 tagAlign shifted files were created. MACS2 (ref. [66]) was used to call peaks with the following parameters: -p 0.01–shift 75–extsize 150–nomodel -B–SPMR –keep-dup all–call-summits. Peaks were later filtered at a *q*-value threshold of 0.0001 for further analyses. Peak counts and sizes for each replicate were calculated using a custom Python script and Jaccard indices for similarities between called peaks were calculated using BEDTools[67]. Differentially accessible regions were calculated using

the DiffBind and EdgeR[68] packages in R. Regions with an FDR ≤ 0.05 were defined as significantly differentially accessible regions.

*Mapping of ATAC gene lists to LSC+ or LSC− signatures.* Differentially accessible regions were mapped to genes using the annotatePeak function of the R package ChIPseeker 1.22.1. Feature distribution was plotted using the function plotAnnoBar. Annotated regions at an FDR threshold of $1 \times 10^{-10}$ were mapped to published LSC+ or LSC− signatures (GSE76008) at an FDR threshold of 0.05. Quantile normalized mean (log-transformed) counts of these annotated regions were plotted on a violin plot in R and a *t*-test was run to estimate the difference in the mean score between the LSC+ and LSC− groups. Data have been deposited in the Gene Expression Omnibus database under the accession code GSE176071.

*Pathway analysis of ATAC-seq data.* Differentially accessible regions were separated in regions with higher binding affinity in the EV and regions with higher binding affinity in the NLS1–HK2 samples. The differentially accessible regions were subjected to pathway analysis using the GREAT tool version 4.0.4 (http://great.stanford.edu/public/html/). Pathways enriched at an FDR of 0.05 belonging to the category of Gene Ontology Biological Processes were visualized as a network using Cytoscape 3.8.1 and EnrichmentMap 3.3.1 and AutoAnnotate 1.3.3. Gene overlaps between HSC/stem or myeloid/granulocytes gene signatures were mapped using a hypergeometric test at an FDR of 0.05.

*Classification of ATAC gene lists as HSC and stem or myeloid and granulocytes.* Changes in gene expression between the LSC+ and LSC− fractions were mapped to the Gene Expression Omnibus dataset GSE24759 (DMAP)[20] containing Affymetrix GeneChip HT-HG_U133A Early Access Array gene expression data of 20 distinct haematopoietic cell states. The GSE24759 data were background corrected using Robust Multi-Array Average, quantile normalized using the expresso function of the affy Bioconductor package (affy_1.38.1, R 3.0.1) and array batch corrected using the ComBat function of the sva package (sva_3.6.0). Gene differential expression was calculated between the HSC and granulocyte population using limma *t*-test. Scatterplots show the LSC+/LSC− expression score ($-\log_{10}(P$ value)) on the *x* axis and the *t*-value of the HSC/granulocyte expression score.

**ChIP-seq.** *ChIP sample preparation.* Chromatin immunoprecipitation assays were performed in control NB4 cells and NB4 cells overexpressing nuclear HK2. Briefly, the cells were treated with 1% formaldehyde for 10 min at room temperature. After harvesting, the cells were pelleted and resuspended in cell lysis buffer (5 mM PIPES pH 8.0, 85 mM KCl, 0.5% NP-40, 1 mM phenylmethylsulfonyl fluoride, 1 mM sodium orthovanadate, 1 mM NaF, 1.0 μg ml⁻¹ leupeptin, 1 μg ml⁻¹ aprotinin and 25 mM β-glycerophosphate). After 10 min rotation at 4 °C, the cellular material was centrifuged at 2,000 g for 10 min to obtain the nuclei. The nuclear pellet was resuspended in MNase digestion buffer (10 nM Tris–HCl pH 7.5, 0.25 M sucrose, 75 mM NaCl plus the above-indicated phosphatase/protease inhibitors) and the $A_{260}$ value was measured. Crosslinked chromatin was sheared to fragments of 400–500 bp by sonication. To release the nuclear material, the samples were adjusted to 0.5% SDS and rotated for 1 h at room temperature. The insoluble material was pelleted at 2,000 g for 10 min and the soluble material was diluted to 0.1% SDS with RIPA buffer along with above-mentioned phosphatase/protease inhibitors. The G-Sepharose (Pierce) pre-cleared lysate was incubated with anti-HK2 (Santa Cruz Biotechnology, sc-374091; 1:100) overnight at 4 °C. Magnetic protein G Dynabeads (Invitrogen) were added for 2 h at 4 °C. The beads were pelleted, washed and the antibody–chromatin complexes were eluted[69].

*ChIP library preparation.* Samples were prepared as outlined by the Takara Bio ThruPLEX DNA-seq kit user guide. Briefly, the samples were normalized to 1 ng of input DNA and topped up to 10 μl with nuclease-free water. The samples were then mixed with 3 μl of template preparation master mix and incubated at 22 °C for 25 min, followed by 55 °C for 20 min in a Veriti 96-well thermocycler (Thermo Fisher Scientific). Following template preparation, the samples were mixed with 2 μl of library synthesis master mix and incubated at 22 °C for 40 min on the same cycler. The samples were prepared for library amplification as outlined in the user guide and indexed individually. From here, the PCR cycles were optimized by adding 0.75 μl of 10×SYBR green I nucleic acid gel stain (Thermo Fisher Scientific) to each sample. Once mixed, a 10-μl aliquot from each sample was run on a CFX96 touch real-time PCR cycler (Bio-Rad). The samples were normalized to one-third of their amplification curve and amplified on a Veriti 96-well thermocycler. The samples were topped up to 50 μl with nuclease-free water before bead clean-up with AMPure XP beads (Beckman Coulter). Final library sizing and quality control was evaluated using Agilent's high sensitivity DNA kit run on a 2100 Bioanalyzer (Agilent Technologies).

*Sequencing.* The libraries were quantified by qPCR and then normalized and pooled to 1.25 nM. Each 1.25-nM pool was denatured using 4 μl of 0.2 N NaOH (Sigma) for 8 min at room temperature before being neutralized with 5 μl of 400 mM Tris–HCl (Sigma). The library pools were mixed with Illumina's XP master mix and loaded immediately onto a NovaSeq 6000 S1 flow cell. The samples were sequenced with the following run parameters: read 1, 50 cycles; read 2, 50 cycles; index 1, 8 cycles; and index 2, 8 cycles.

*ChIP–seq data analysis.* The ChIP–seq analysis was performed in control, NLS1–HK2 and NLS2–HK2 NB4 cells. Before analysis, the read adaptors were removed using Trim Galore v. 0.4.0, removing reads that were smaller than 35 bp after trimming. In addition, a base-pair quality score cutoff ($q = 30$) was used for filtering low-quality base pairs. Reads were aligned against hg38 (UCSC version) using Bowtie2 v2.3.2. Secondary and supplementary alignments were removed and only primary alignments were kept. Alignment reads were de-duplicated to remove duplicate reads and keep unique reads using picard v. 1.9.1. Broad peaks were identified from the alignment files using MACS2 v. 2.1.1 with a cutoff score ($q$) < 0.05. The peaks were annotated with all the potential genomic features based on hg38 GENCODE v24 gene assembly, which was downloaded from the UCSC database. Data have been deposited in Gene Expression Omnibus database under the accession code GSE176072.

*ChIP–seq pathway analysis.* MACS2 called peaks at an FDR of 0.01 for individual samples and pooled samples were subjected to pathway analysis using the GREAT tool version 4.0.4 (http://great.stanford.edu/public/html/). Gene Ontology Biological Processes pathways enriched at an FDR of 0.05 in a minimum of four of six NLS1–HK2 or NLS2–HK2 samples were visualized as a network using Cytoscape 3.8.1, EnrichmentMap 3.3.1 and AutoAnnotate 1.3.3. Scores corresponding to the $-\log_{10}$-transformed FDR value of the overlapping peaks between individual and pooled samples were represented as a bar graph for select pathways. Overlapping peaks for the two replicates of MAX (ENCFF793GVV.bed) was compared to NLS-HK2 ChIP–seq using the function findOverlappingPeaks from the ChIPpeakAnno 3.20.1 R package.

*ChIP–seq motif enrichment analysis.* The findMotifsGenome algorithm from HOMER v4.7 was used to identify known enriched motifs in genomic regions in each sample. Significantly enriched motifs at an FDR of 0.05 that were shared between the three NLS1–HK2 and three NLS2–HK2 samples were retrieved and the consensus sequences of motifs were aligned using MAFFT (https://mafft.cbrc.jp/alignment/software/). TFmotifView (http://bardet.u-strasbg.fr/) was used to calculate the significance of the enrichment of the CACGTG motif in selected peaks using random sequences background and G + C content adjustment.

**DNA-damage induction.** Sorted cells from patients with AML or 8227 cells, or transduced NB4 or 8227 cells were treated with daunorubicin 50 nm for select time periods (3 and 6 h). The cells were then spun down and fixed with formaldehyde for confocal microscopy.

**Comet assay.** For the neutral Comet assay, equal amounts of cells per condition were treated with 70 nM daunorubicin for 6 h, embedded in agarose on slides and the assay was performed as per protocol[70]. Tail moment was quantified using the open comet software[71].

**RNA isolation and real-time qPCR with reverse transcription.** Total RNA was isolated from 8227 leukaemia cells, separated into stem and bulk populations, using an RNeasy plus mini kit (Qiagen) and cDNA was prepared using SuperScript IV reverse transcriptase (Thermo Fisher Scientific). Equal amounts of cDNA from each sample were added to a prepared master mix (Power SYBR Green PCR master mix; Applied Biosystems). Quantitative real-time PCR reactions were performed on an ABI Prism 7900 sequence detection system (Applied Biosystems). The relative abundance of a transcript was represented by the threshold cycle of amplification ($C_T$), which is inversely correlated to the amount of target RNA/first-strand cDNA being amplified. To normalize for equal amounts of cDNA, we assayed the transcript levels of the *18S* ribosomal RNA gene. The comparative $C_T$ method was calculated as per the manufacturer's instructions. The primers that were used are listed in Supplementary Table 1.

**DNA-repair pathway analysis in primary patient samples.** GSEA enrichment analysis was performed to identify changes in gene expression in DNA-damage-response pathways between LSCs and bulk primary AML cells as well as undifferentiated versus committed primary patient samples. The Gene Expression Omnibus dataset GSE76008 ($n = 227$) and a Princess Margaret Cancer Centre cohort ($n = 11$) were used.

**Statistical analysis and reproducibility.** GraphPad Prism 6.0 was used to perform the analyses. Statistical analyses were performed using an unpaired Student's *t*-test and one- or two-way ANOVA testing was used to compare mean values between multiple groups. The data distribution was assumed to be normal but this was not formally tested. The investigators were not blinded to allocation during experiments and outcome assessment. However, key experiments were reproduced independently by different individuals. Quantitative end points were used for measurements. No data were excluded. Figures 4b, 6f,k,l and Extended Data Figs. 2m–o, 5l, 9f,i, 10a–c represent images where experiments were performed less than three times.

**Reporting summary.** Further information on research design is available in the Nature Research Reporting Summary linked to this article.

## Data availability

RNA sequencing, ATAC-seq and ChIP–seq data have been deposited to the Gene EXpression Omnibus Database under the accession numbers GSE176103, GSE176071 and GSE176072, respectively. All other data supporting the findings of this study are available from the corresponding author on reasonable request. Source data are provided with this paper.

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

## Acknowledgements

We thank J. Flewelling (Princess Margaret Cancer Centre) for administrative assistance and the Leukemia Tissue Bank (Princess Margaret Cancer Centre) for providing the primary AML samples. This work was supported by the Canadian Institutes of Health Research, Ontario Institute for Cancer Research with funding provided by the Ontario Ministry of Research and Innovation, Princess Margaret Cancer Centre Foundation and Ministry of Long Term Health and Planning in the Province of Ontario. A.D.S. holds the Ronald N. Buick Chair in Oncology Research. G.E.T was supported by a Princess Margaret Research Fellowship and Lady Tata Memorial Trust Fellowship. G.E. is supported by a scholarship grant from the Garron Family Cancer Center, The Hospital for Sick Children and from the Hold'em for Life Oncology Challenge, University of Toronto.

## Author contributions

G.E.T., G.E. and A.D.S. conceptualized the project. G.E.T., G.E., L.G.-P., A.B., V.V., P.S.P., J.C., B.N., K.B.K., D.H.K., R.H., N.M., X.W., M.G., R.P.S., F.W.H., S.M.H. and C.O'B. contributed to the methodology section. G.E.T., G.E. and A.D.S. acquired funding for this project. Project administration was performed by A.D.S. and Jill Flewelling. This project was supervised by A.D.S., B.R., G.D.B., R.H., S.K. and J.E.D. Writing of the original draft was performed by G.E.T., G.E. and A.D.S. Review and editing of the paper was performed by G.E.T., G.E., L.G.-P., A.B., V.V., P.S.P., J.C., B.N., K.B.K., D.H.K., R.H., N.M., X.W., M.G., R.P.S., F.W.H., S.M.H., C.O'B., B.R., C.L.J., A.A., M.D.M., G.D.B., R.H., S.K., J.E.D. and A.D.S.

## Competing interests

A.D.S. has received research funding from Takeda Pharmaceuticals and Medivir AB as well as consulting fees/honorarium from Takeda, Novartis, Jazz and Otsuka Pharmaceuticals. A.D.S. is named on a patent application for the use of DNT cells to treat AML (US patent application no. US62/971,534). The remaining authors declare no competing interests.

## Additional information

**Extended data** is available for this paper at https://doi.org/10.1038/s41556-022-00925-9.

**Correspondence and requests for materials** should be addressed to Aaron D. Schimmer.

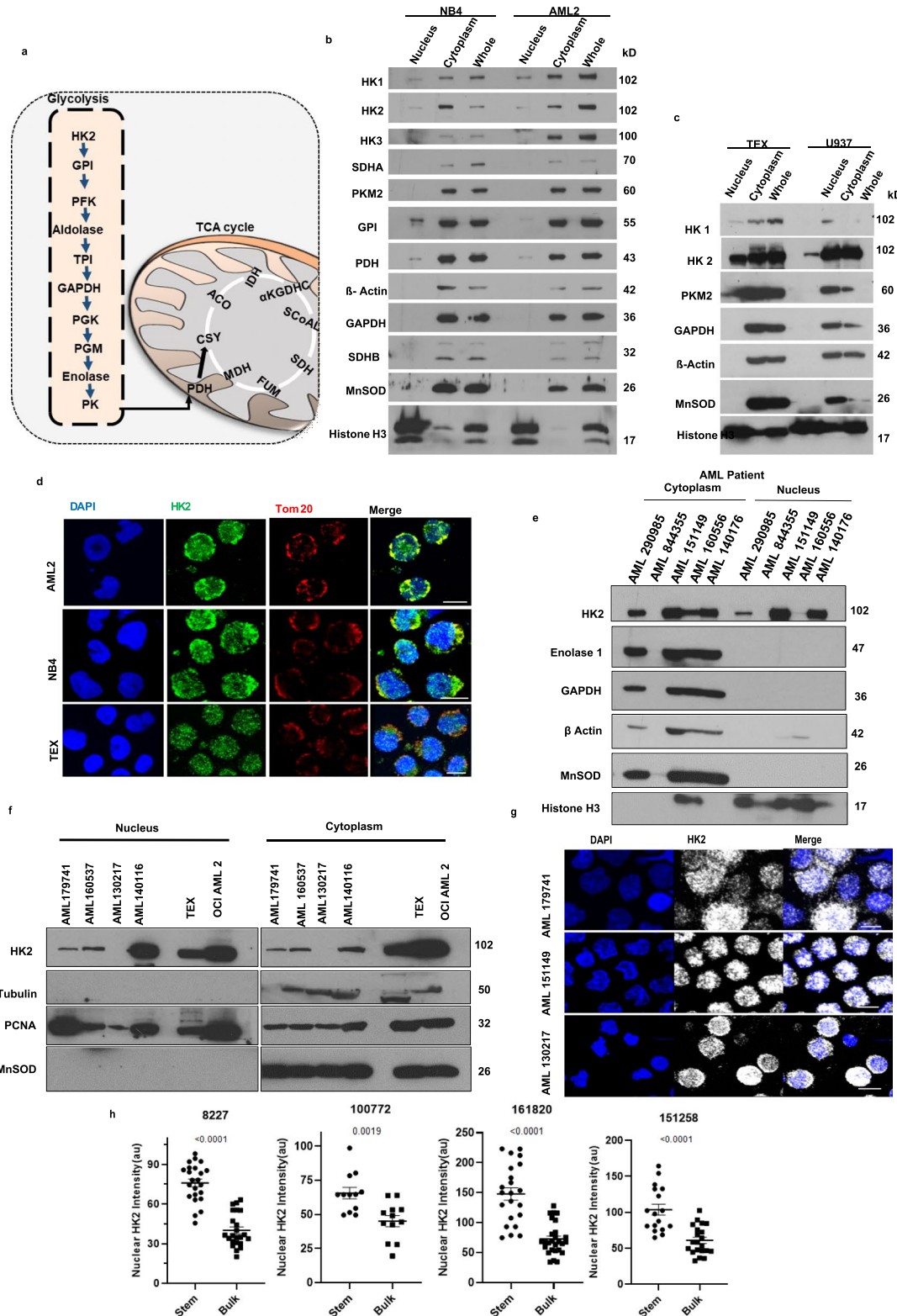

**Extended Data Fig. 1 | HK2 localizes to the nucleus in AML.** (a) Diagram of the glycolytic and TCA cycles. (b) Immunoblot analysis of glycolytic and TCA cycle metabolic enzymes in nuclear, cytoplasmic and whole-cell lysates of (b) OCI-AML2 and NB4 cells & (c) TEX and U937 cells. Representative immunoblot from 3 biologic repeats. (d) Representative confocal microscopy images of HK2 (green) and Tom20 in AML2, NB4 and TEX cells. Scale bar = 10μm. Representative immunoblot from 3 biologic repeats. (e) Expression of HK2 and the related glycolytic enzymes, enolase and GAPDH, by immunoblotting in cytoplasmic and nuclear fractions of primary AML patient samples n = 5 biologically independent samples. (f) Expression of HK2 by immunoblotting in cytoplasmic and nuclear fractions of primary AML patient samples. n = 4 biologically independent samples. (g) Representative confocal microscopy images of HK2 (white) in AML patient samples. Scale bar = 10μm. n = 3 biologically independent samples. (h) Fluorescence intensity analysis in 8227 cells and AML patient samples, separated into stem and bulk populations. n = 165 cells examined from 4 biologically independent samples. Statistical analyses for all experiments was performed using a two-tailed unpaired Student's t-test. Data shown represent mean +/- s.e.m.

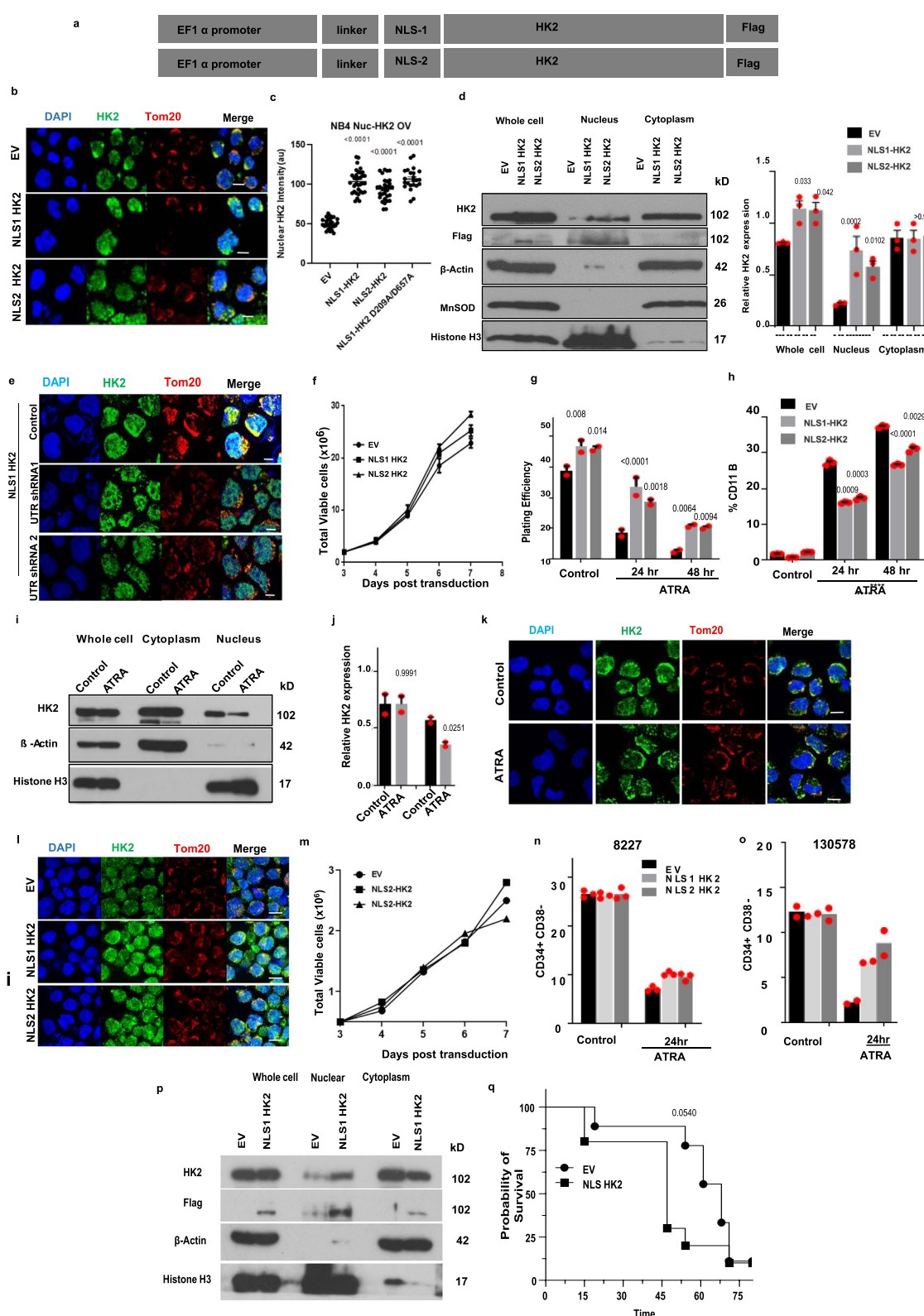

**Extended Data Fig. 2 | See next page for caption.**

**Extended Data Fig. 2 | Nuclear HK2 over-expression enhances stemness.** (a) cDNA construct of HK2 fused in frame with c-Myc (NLS1) or SV40 (NLS2) Nuclear Localizing Signal (NLS). (b) Representative confocal microscopy images of HK2 (green) or Tom20 (red) in NB4 cells 5 days after transduction with NLS-HK2 or EV. Scale bar= 10μm. Representative images from 3 biologic repeats. (c) Quantification of fluorescence intensity of nuclear HK2 in NB4 cells after transduction with NLS-HK2 or EV. n = 114 cells examined from 3 biologic repeats. (d) NB4 cells were transduced with NLS- HK2 or EV. The level of HK2 in the nucleus and whole-cell lysates was measured after 5 days of transduction using immunoblotting. The densitometry plots of relative HK2 expression were performed in subcellular lysates. n = 3 biologic repeats. (e) Representative confocal microscopy images of HK2 (green) or Tom20 (red) in NLS1-HK2 NB4 cells after transduction with shRNAs targeting the UTR of HK2. Scale bar= 10μm. Representative image from 3 biologic repeats. (f) Growth and viability of NB4 cells at increasing times after transduction with NLS-HK2. n = 5 biologic repeats. (g) Clonogenic growth of NB4 cells transduced with NLS-HK2 and treated with ATRA (100nM). n = 3 biologic repeats. (h) Expression of CD11b in NB4 cells transduced with NLS-HK2 and treated with ATRA (100nM). n = 4 biologic repeats. (i) NB4 cells were treated with ATRA for 72hrs and subcellular fractions were analysed for HK2 by immunoblot. n = 3 biologic repeats. (j) The densitometry plots of relative HK2 expression were performed in subcellular lysates. (k) Representative confocal microscopy images of HK2 (green) and Tom20 (red) in NB4 cells treated with ATRA. Scale bar= 10μm. Representative image from 2 biologic repeats. (l) Representative confocal microscopy images of HK2 (green) in 8227 cells 5 days after transduction with NLS-HK2 or EV. Scale bar= 10μm. Representative image from 3 biologic repeats. (m) Growth and viability of 8227 after transduction with NLS-HK2. n = 2 biologic repeats. (n) Percentage of CD34+CD38- cells in 8227 cells, n = 2 biologic repeats and (o) 130578 cells, n = 2 biologic repeats transduced with NLS-HK2 and treated with ATRA (100nM). (p) TEX cells were transduced with NLS1-HK2 or EV. Subcellular fractions were analysed for HK2 by immunoblot 5 days post transduction. (q) TEX cells transduced with NLS1-HK2 or EV were injected into NSGF mice. Survival of the mice was measured over 75 days (n = 9, EV mice, n = 10, NLS1-HK2 mice). Statistical analyses was performed using a two-tailed unpaired Student's t-test (d, j) and Ordinary Two-way ANOVA, Tukeys multiple comparison test (g, h). Data from represent mean +/- s.e.m, except (m-o) which represents the mean. A Kaplan Meier curve analysed survival using the gehan-breslow-wilcoxon test.

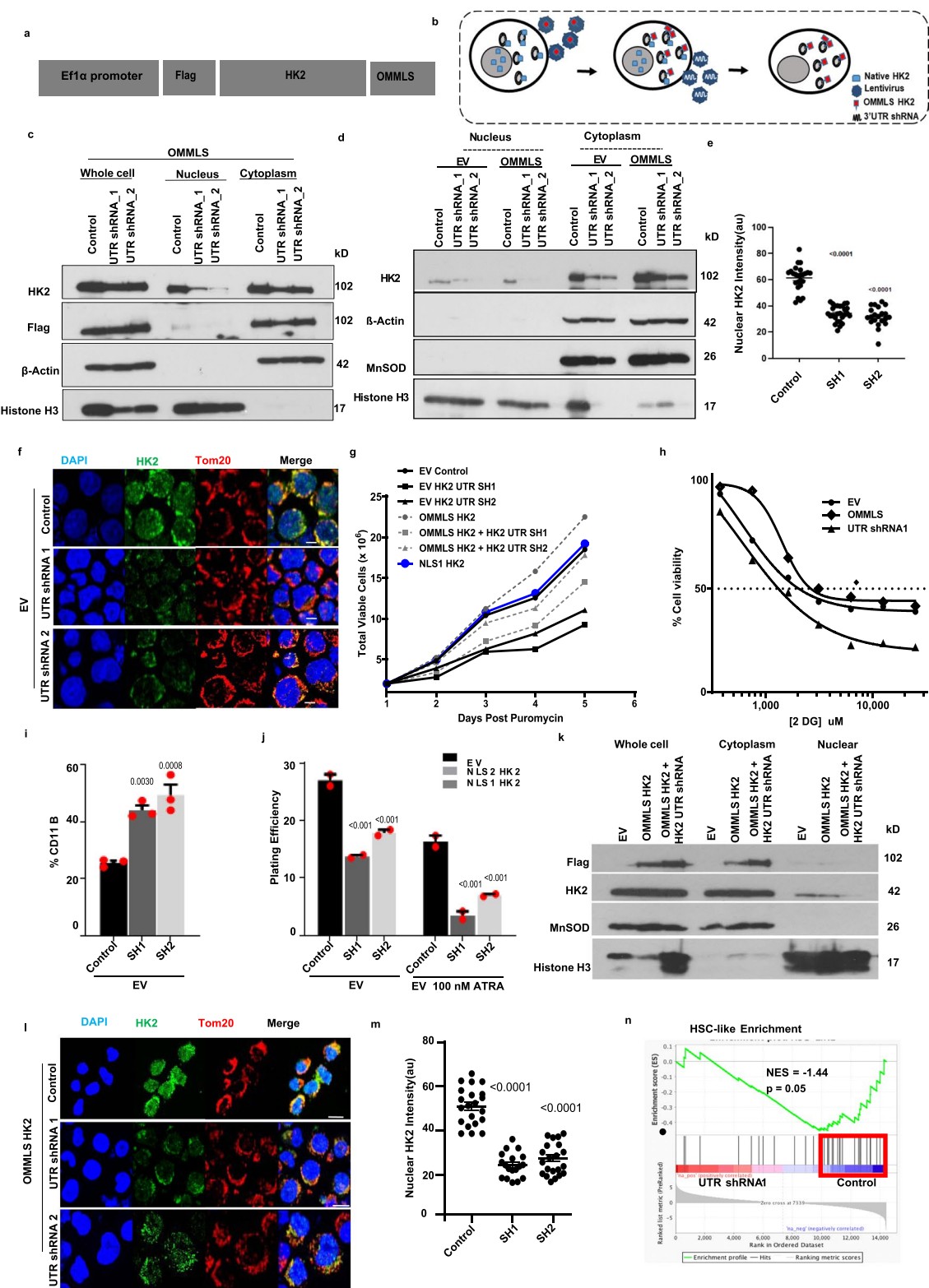

**Extended Data Fig. 3 | See next page for caption.**

**Extended Data Fig. 3 | Selective nuclear HK2 knockdown decreases stemness.** (a) cDNA construct of HK2 fused in frame with outer mitochondrial membrane localizing signal (OMMLS). (b) Experimental approach to selectively knockdown nuclear HK2. (c) & (d) NB4 cells overexpressing mitochondrial localized HK2 (OMMLS HK2) were transduced with shRNAs targeting the UTR of HK2. Levels of HK2 in the nucleus and whole cell were measured by immunoblot 5 after of transduction. Representative immunblot from 3 biologic repeats. (e) Quantification of fluorescence intensity of transduced OMMLS HK2 NB4 with shRNAs targeting the UTR of HK2. n = 73 cells examined from 3 biologic repeats. (f) Representative confocal microscopic images of HK2 (green) and Tom20 (red) in NB4 cells after transduction with shRNA targeting the UTR of HK2. Scale bar = 10μm. Representative images from 3 biologic repeats. (g) Cell growth and viability plot of OMMLS HK2 or EV NB4 cells after transduction with shRNAs targeting the UTR of HK2. n = 2 biologic repeats. (h) Cell viability of OMMLS-HK2, UTRsh1, and EV NB4 cells after treatment with increasing concentrations of 2-DG for 48hrs. n = 3 biologic repeats. (i) Expression of CD11b in EV NB4 cells after transduction with shRNA targeting the UTR of HK2 and treated with ATRA (100nM). n = 3 biologic repeats. (j) Clonogenic growth of EV NB4 cells after transduction with shRNA targeting the UTR of HK2 and treated with ATRA (100nM). n = 3 biologic repeats. (k) OMMLS HK2 AML2 cells were transduced with shRNAs targeting the UTR of HK2. Subcellular lysates were analysed by immunoblot 5 days after transduction. (l) Representative confocal microscopic images of HK2 (green) after transduction of 8227 cells with OMMLS-HK2 and UTR shRNAs targeting HK2. Scale bar = 10μm. Representative image from 3 biologic repeats. (m) Quantification of fluorescence intensity of nuclear HK2 in OMMLS HK2 8227 cells after transduction with shRNAs targeting the UTR of HK2. n = 63 cells examined from 3 biologic repeats. (n) Gene set enrichment analysis (GSEA) in 8227 cells transduced with OMMLS-HK2 and UTR shRNAs targeting HK2 for HSC gene signatures. The normalized enrichment scores (NES), and p value analysed using a modified Kolmogorov–Smirnov test. Statistical analyses for experiments (e, m) was performed using a two-tailed unpaired Student's t-test and Ordinary one-way ANOVA Sidaks multiple comparison (i-j). Data represent mean +/- s.e.m.

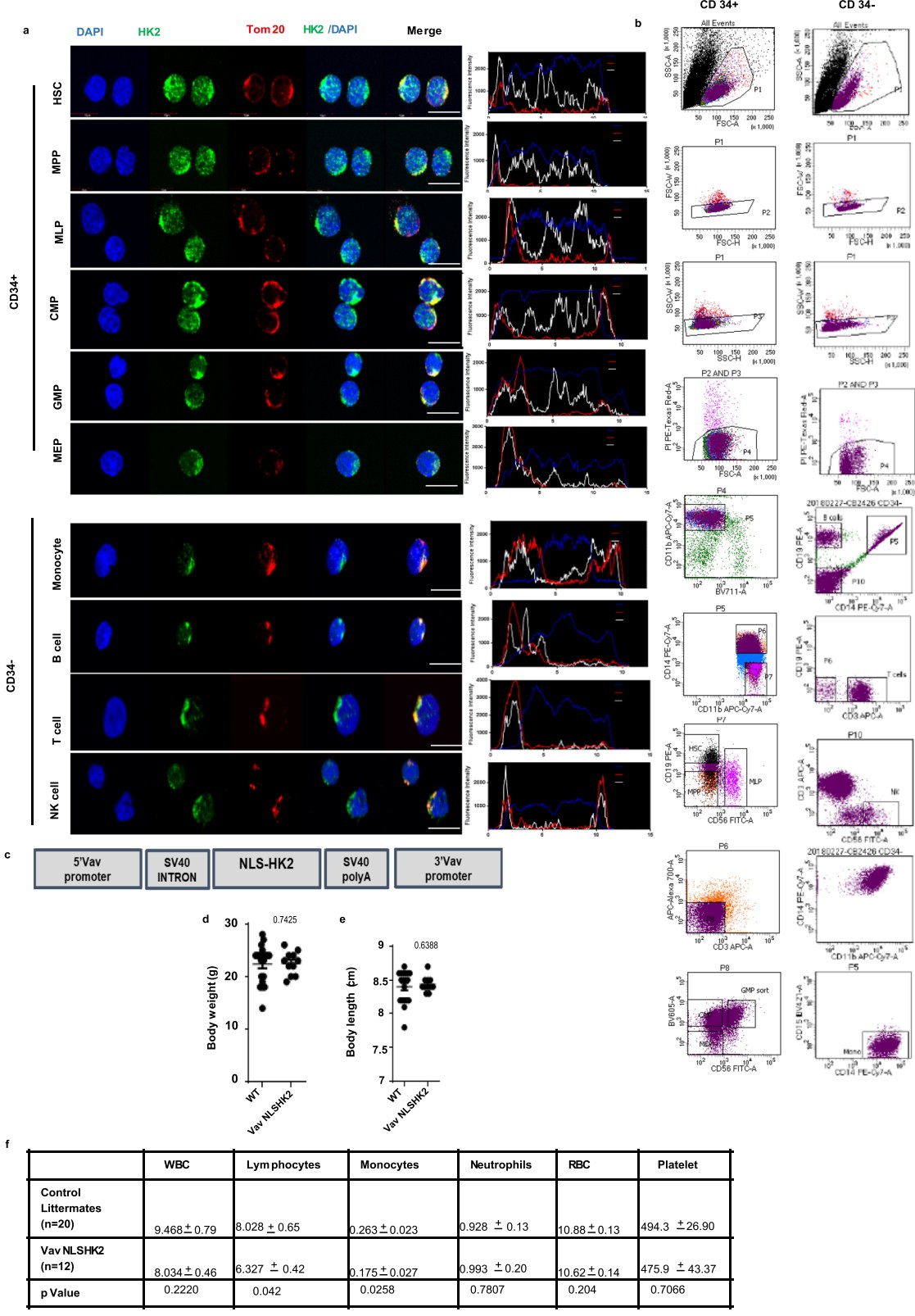

**f**

|  | WBC | Lymphocytes | Monocytes | Neutrophils | RBC | Platelet |
|---|---|---|---|---|---|---|
| Control Littermates (n=20) | 9.468 ± 0.79 | 8.028 ± 0.65 | 0.263 ± 0.023 | 0.928 ± 0.13 | 10.88 ± 0.13 | 494.3 ± 26.90 |
| Vav NLSHK2 (n=12) | 8.034 ± 0.46 | 6.327 ± 0.42 | 0.175 ± 0.027 | 0.993 ± 0.20 | 10.62 ± 0.14 | 475.9 ± 43.37 |
| p Value | 0.2220 | 0.042 | 0.0258 | 0.7807 | 0.204 | 0.7066 |

**Extended Data Fig. 4 | See next page for caption.**

**Extended Data Fig. 4 | HK2 localizes to nucleus of hematopoietic stem/progenitor cells.** (a) Representative confocal microscopy images of HK2 (green) and Tom20 (red) in FACS sorted CD34+ and CD34- cord blood cells. Scale bar = 10μm. The white (HK2), red (Tom20) and blue (DAPI) curve in the scan profiles represents the fluorescence intensity of HK2, Tom20 and DAPI along the plane. Representative images from 3 biologic samples. (b) Gating strategy for FACs sorted CD34+ and CD34- cord blood cells. (c) Vav promoter flanked cDNA construct of HK2 fused in frame with a nuclear localizing signal. (d) Body weight (n = 42 wild-type control mice, n = 22 Vav-NLS-HK2 mice) and body length (e) (n = 48 wild-type control mice, n = 31 Vav-NLS-HK2 mice) of Vav NLS-HK2 transgenic mice and control littermates. (f) Complete blood count analysis of Vav NLS-HK2 transgenic mice and control littermates (n = 21 wild-type control mice, n = 13 Vav-NLS-HK2 mice). Statistical analyses for experiments (d-e) was performed using an unpaired Student's t-test. Data represent mean +/- s.e.m.

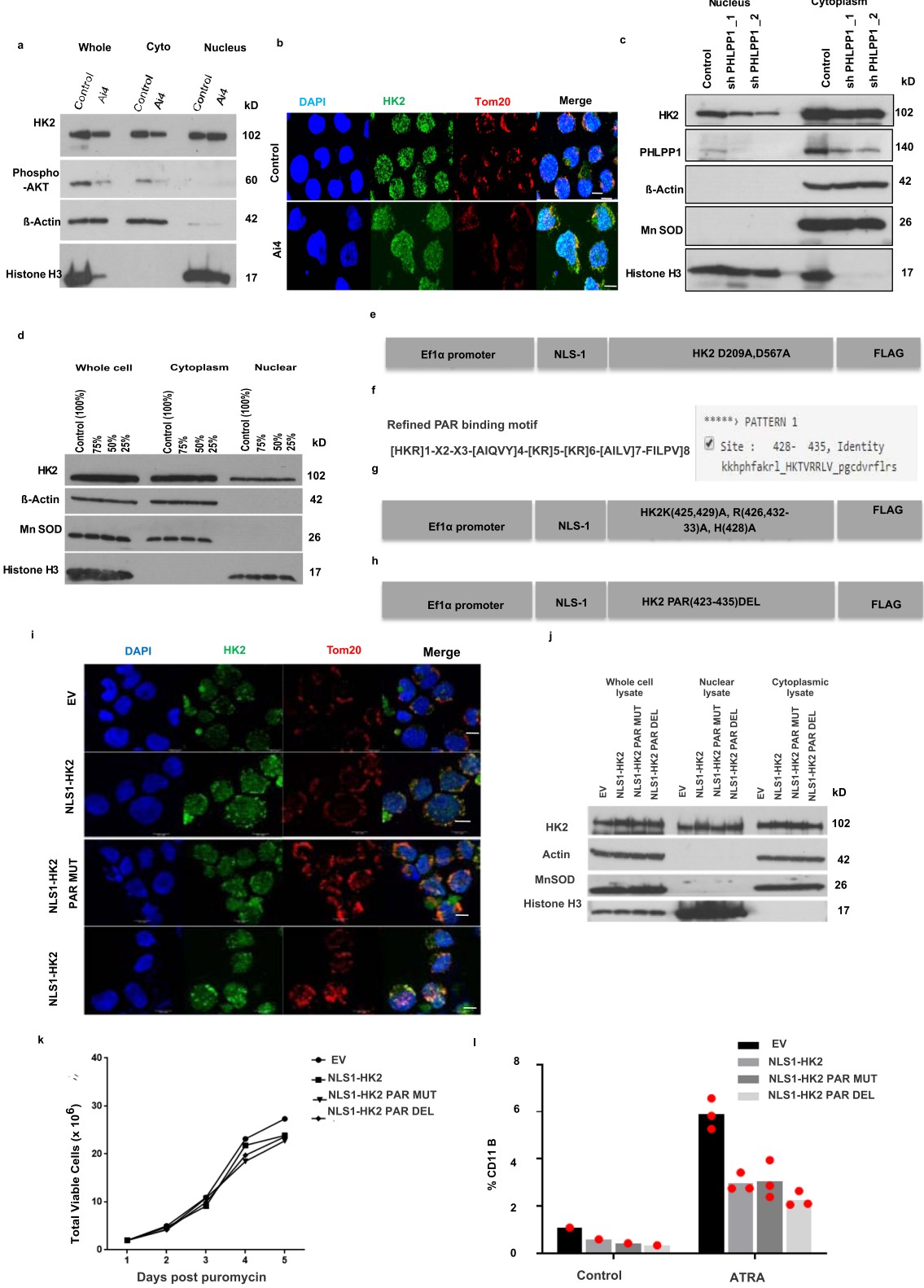

**Extended Data Fig. 5 | See next page for caption.**

**Extended Data Fig. 5 | HK2 maintains stemness independently of its metabolic function.** (a) Subcellular fractions of NB4 cells were measured by immunoblot analysis after treatment with DMSO or AKT inhibitor (Ai4) for 24hrs. Representative immunoblot from 3 biologic repeats. (b) Representative confocal images of HK2 (green) and Tom20 (red) in NB4 cells treated with AKT inhibitor (Ai4). Scale bar= 10μm. Representative image from 3 biologic repeats. (c) NB4 cells were transduced with shRNAs targeting PHLPP1 or control sequences. Five days after transduction, HK2 expression in the nucleus and cytoplasm was measured by immunoblotting. Representative immunoblot from 2 biologic repeats. (d) AML2 cells were grown with decreasing glucose concentrations in IMDM media for 24 hours and subcellular fractions were analysed. Representative immunoblot from 3 biologic repeats. (e) Construct of nuclear localizing HK2 kinase-dead double mutant with c-Myc nuclear localizing signal at the N-terminal region. The Aspartic acid residues at 209 and 567 were mutated to Alanine. (f) The predicted PAR binding motif in HK2 using Pattinprot. (g) Construct of nuclear localizing PAR mutant of HK2 with c-Myc nuclear localizing signal at the N-terminal region. (h) Construct of nuclear localizing PAR deletion of HK2 with c-Myc nuclear localizing signal at the N-terminal region of HK2. (i) Representative confocal microscopy images of HK2 (green) or Tom20 (red) in NB4 cells 5 days after transduction with NLS1-HK2, NLS1- HK2 Par mut, NLS1-HK2 Par_Del or EV. Scale bar = 10μm. Representative image from 2 biologic repeats. (j) Immunoblot analysis of subcellular fractions of NLS1-HK2, NLS1- HK2 Par_mut, NLS1-HK2 Par_Del cells transduced NB4 cells. (k) Cell growth and viability of NB4 cells after transduction with NLS1-HK2, NLS1- HK2 Par_mut, NLS1-HK2 Par_Del or EV. n = 1 independent experiment with 3 technical replicates. (l) Expression of CD11b in NB4 cells transduced with NLS1-HK2, NLS1- HK2 Par_mut, NLS1-HK2 Par_Del and treated with ATRA (100nM). n = 2 biologic repeats. Data from (l) represent mean.

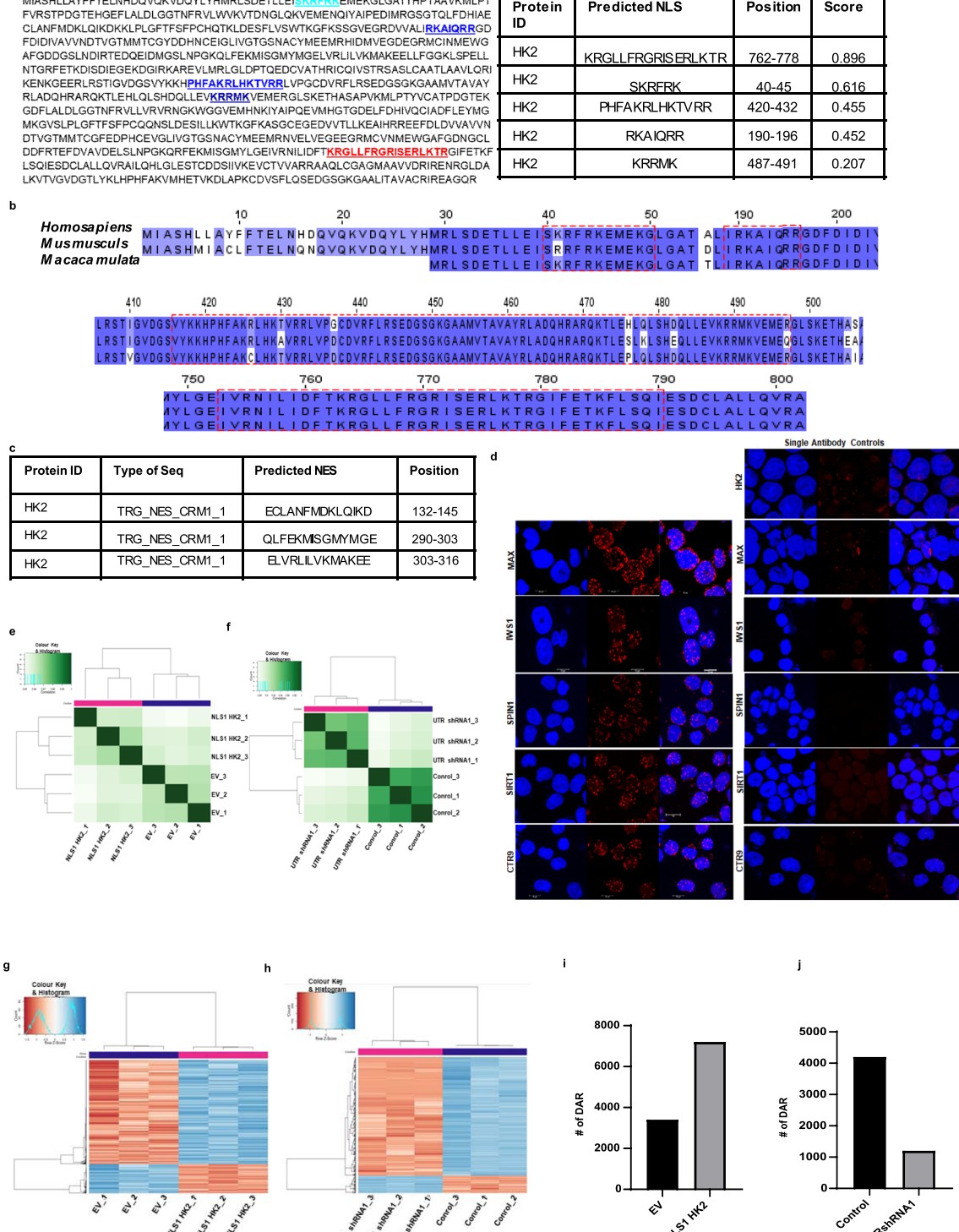

**Extended Data Fig. 6 | Nuclear HK2 modifies chromatin accessibility.** (a)The mammalian HK2 protein sequence was evaluated for nuclear localizing signals using the SeqNLS algorithm. (b) NLS conservation in HK2 of Homosapiens, Mus Musculus and Macaca mulata analysed using Clustal omega. (c)The predicted sequences and the positions of HK2 Nuclear Export Sequence by NESX1 library using Pattinprot and ELM library. (d) Representative confocal images of DuoLink Promity Assay interactions. Scale bar = 10μm. Representative image from 3 biologic repeats. (e) Clustering on peaks using DiffBind comparing NB4 cells transduced with NLS1-HK2 and EV. (f) Clustering on peaks using DiffBind comparing NB4 cells overexpressing OMMLS HK2 transduced with control sequences or shRNA targeting the UTR of HK2. (g) Heatmap plots of differentially accessible regions in NB4 cells overexpressing NLS1-HK2 or EV. (h) Heatmap plots of differentially accessible regions in NB4 cells overexpressing OMMLS HK2 transduced with control sequences or shRNA targeting the UTR of HK2. (i) Number of Differentially Accessible Regions in NB4 cells with NLS1-HK2 or EV measured by ATAC sequencing. (j) Number of Differentially Accessible Regions in NB4 cells overexpressing OMMLS HK2 transduced with control sequences or shRNA targeting the UTR of HK2 measured by ATAC sequencing.

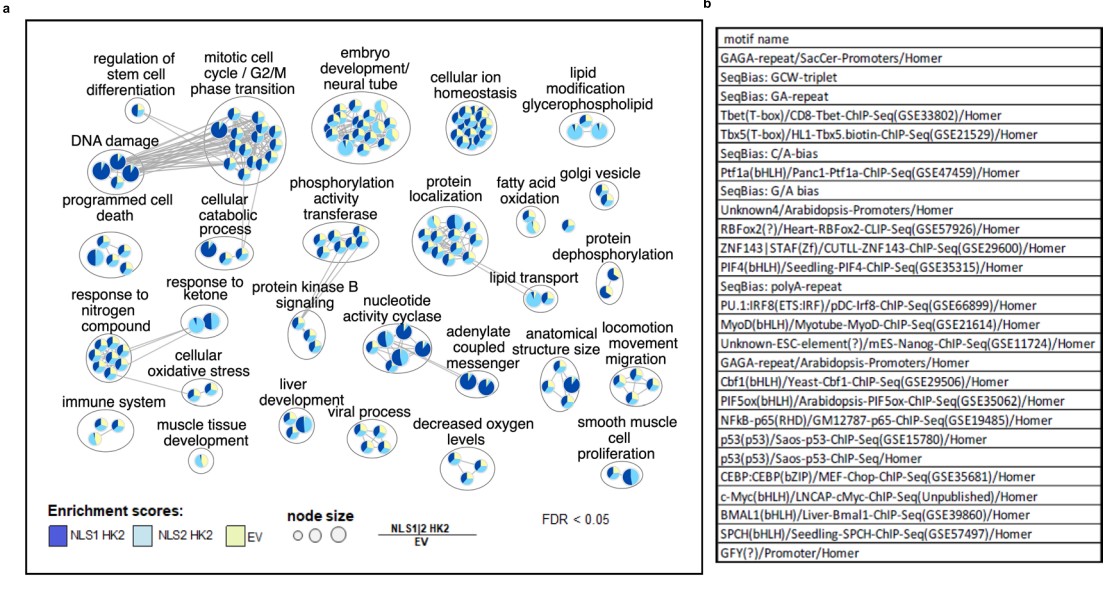

motif name

| motif name |
| --- |
| GAGA-repeat/SacCer-Promoters/Homer |
| SeqBias: GCW-triplet |
| SeqBias: GA-repeat |
| Tbet(T-box)/CD8-Tbet-ChIP-Seq(GSE33802)/Homer |
| Tbx5(T-box)/HL1-Tbx5.biotin-ChIP-Seq(GSE21529)/Homer |
| SeqBias: C/A-bias |
| Ptf1a(bHLH)/Panc1-Ptf1a-ChIP-Seq(GSE47459)/Homer |
| SeqBias: G/A bias |
| Unknown4/Arabidopsis-Promoters/Homer |
| RBFox2(?)/Heart-RBFox2-CLIP-Seq(GSE57926)/Homer |
| ZNF143|STAF(Zf)/CUTLL-ZNF143-ChIP-Seq(GSE29600)/Homer |
| PIF4(bHLH)/Seedling-PIF4-ChIP-Seq(GSE35315)/Homer |
| SeqBias: polyA-repeat |
| PU.1:IRF8(ETS:IRF)/pDC-Irf8-ChIP-Seq(GSE66899)/Homer |
| MyoD(bHLH)/Myotube-MyoD-ChIP-Seq(GSE21614)/Homer |
| Unknown-ESC-element(?)/mES-Nanog-ChIP-Seq(GSE11724)/Homer |
| GAGA-repeat/Arabidopsis-Promoters/Homer |
| Cbf1(bHLH)/Yeast-Cbf1-ChIP-Seq(GSE29506)/Homer |
| PIF5ox(bHLH)/Arabidopsis-PIF5ox-ChIP-Seq(GSE35062)/Homer |
| NFkB-p65(RHD)/GM12787-p65-ChIP-Seq(GSE19485)/Homer |
| p53(p53)/Saos-p53-ChIP-Seq(GSE15780)/Homer |
| p53(p53)/Saos-p53-ChIP-Seq/Homer |
| CEBP:CEBP(bZIP)/MEF-Chop-ChIP-Seq(GSE35681)/Homer |
| c-Myc(bHLH)/LNCAP-cMyc-ChIP-Seq(Unpublished)/Homer |
| BMAL1(bHLH)/Liver-Bmal1-ChIP-Seq(GSE39860)/Homer |
| SPCH(bHLH)/Seedling-SPCH-ChIP-Seq(GSE57497)/Homer |
| GFY(?)/Promoter/Homer |

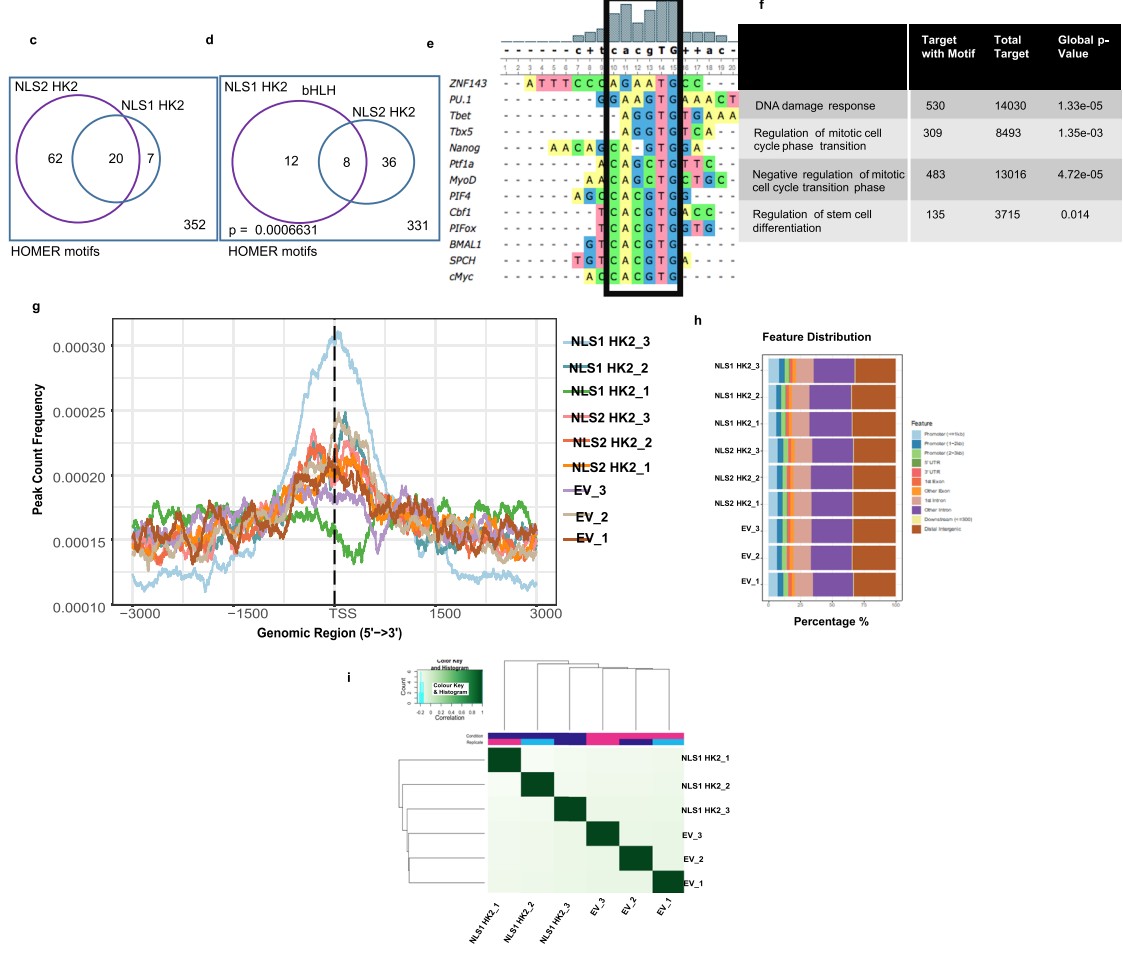

**Extended Data Fig. 7 | ChIP-seq of HK2 reveals a role in stem cell regulation and DNA damage response.** (a) Network representation of NLS1-HK2 and NLS2-HK2 ChIP–seq pathways with higher enrichment scores compared to EV. Node size is proportional to the ratio of NLS1/2 to EV at FDR < 0.05 using a Fisher's exact test. (b) Known motifs from HOMER significantly enriched in both NLS1 and NLS2-HK2 peaks at FDR<0.05. (c) Venn plot showing significantly enriched motifs in NLS1 HK2 and NLS2 HK2 at FDR<0.05 using Fisher's exact test. (d) Venn diagram shows bHLH motifs among the common HOMER motifs between NLS1 HK2 and NLS2 HK2. Fisher's exact test p value for the enrichment of this motif is 0.0006631. (e) Multiple alignment of the motifs using the UGene tool showing the frequency of each nucleotide identifying the consensus sequence. (f) Enrichment of consensus sequence in the specified pathways. (g) Peak distribution around TSS of genes in each individual ChIP–seq biological replicates. (h) Peak distribution of individual replicates relative to promoters, intronic or intergenic regions. (i) Clustering on identical peaks using the DiffBind tool shows that NLS1-HK2 and EV group separately.

**Extended Data Fig. 7 | ChIP-seq of HK2 reveals a role in stem cell regulation and DNA damage response.**

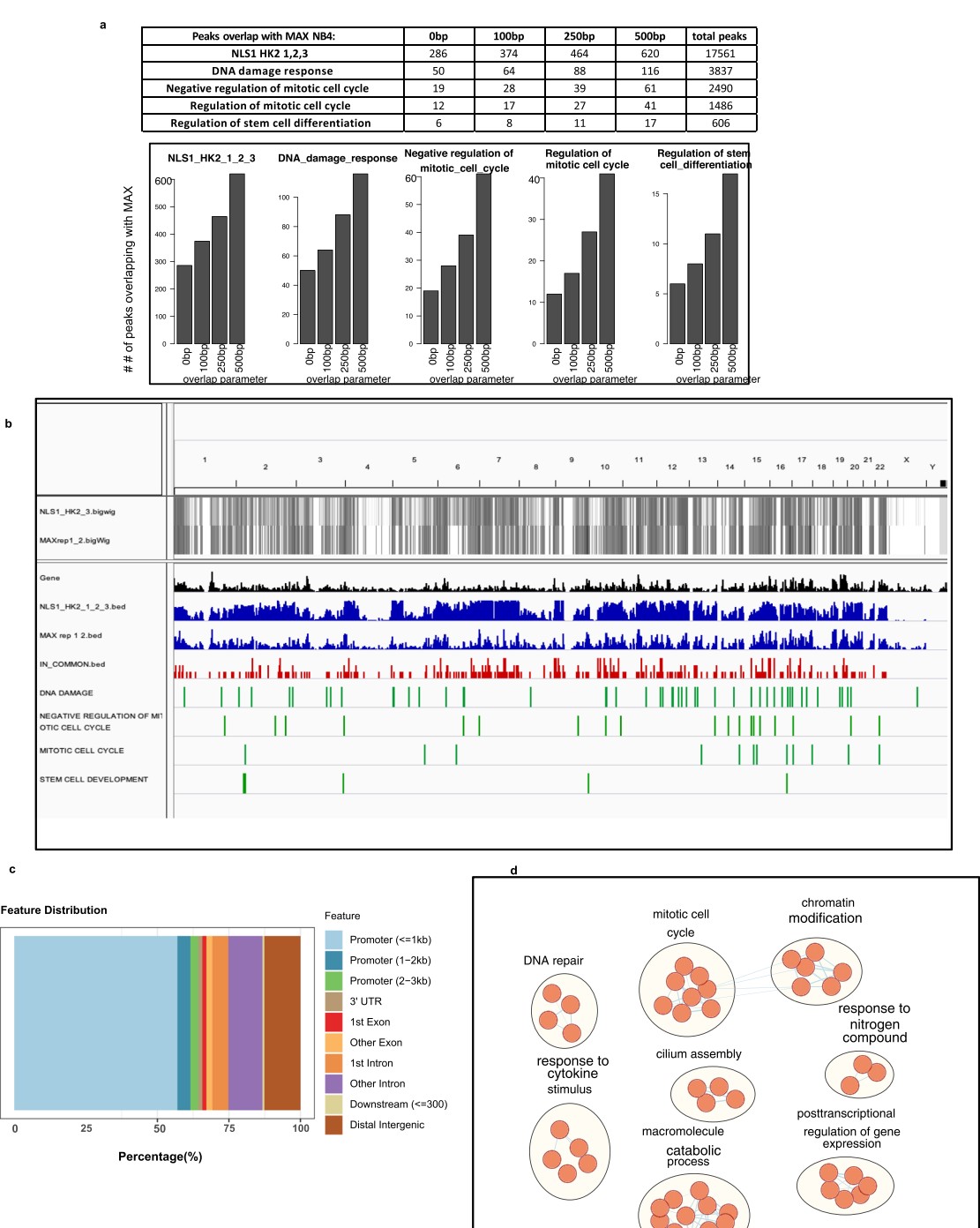

**Extended Data Fig. 8 | Comparative analysis of nuclear HK2 and MAX ChIP-seq.** (a) Number of overlapping peaks between NLS1 HK2 and MAX ChIP–seq peaks using different overlap parameters. (b) Visualization of overlapping peaks between NLS1-HK2 and MAX ChIP–seq peaks in NB4 cells on a full genome view (0bp parameter). (c) Distribution of the peaks in common between NLS1-HK2 and MAX (0bp parameter). (d) Pathways (GO BP) enriched in the peaks common between NLS1-HK2 and MAX (0bp parameter), at FDR <0.000001 using the binomial test.

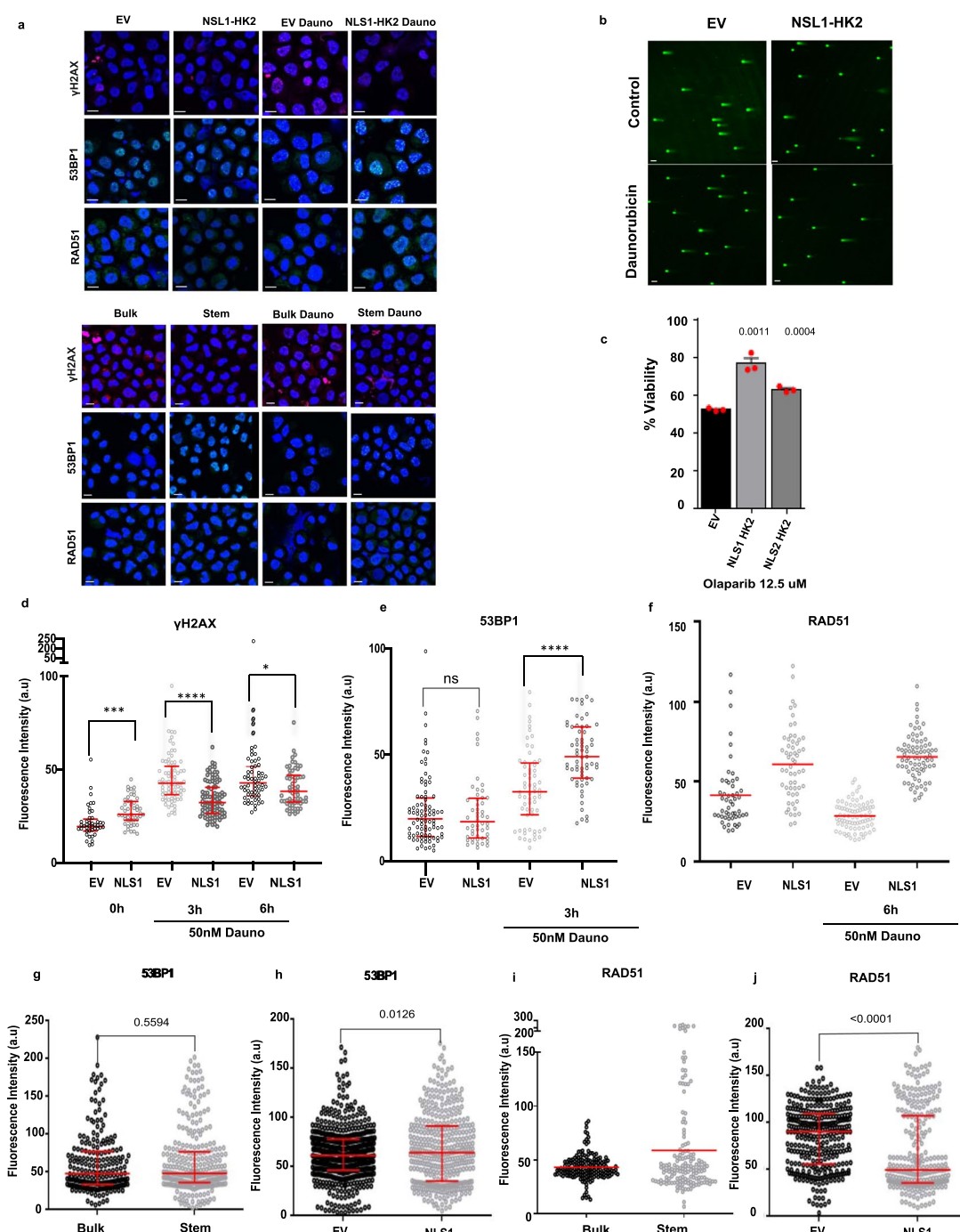

**Extended Data Fig. 9 | Nuclear HK2 enhances DNA repair.** (a) Representative confocal images of γH2AX and RAD51 expression at 6 hours, 53BP1 at 3 hours, in NLS1-HK2/EV transduced 8227 cells and sorted 8227 cells (stem/bulk) at baseline and after treatment with daunorubicin. Scale Bar = 10μm. Representative image from 3 biologic repeats. (b) Representative confocal images of comet assay in transduced 8227 cells. Scale bar = 70μm. Representative image from 3 biologic repeats. (c) Cell viability of EV, NLS1-HK2, NLS2-HK2 transduced NB4 cells after treatment with olaparib (12.5 μM) for 72 hours. n = 3 biologic repeats. (d) γH2AX levels were quantified by confocal microscopy in NLS1-HK2 NB4 cells after treatment with daunorubicin for 3 & 6 hours (50nM). n = 1536 cells examined over 3 biologic repeats. (e) 53BP1 levels were quantified by confocal microscopy after overexpressing NLS1-HK2 in NB4 cells and treating cells with daunorubicin for 3 hours (50nM). n = 1272 cells examined over 3 biologic repeats. (f) RAD51 levels were quantified by confocal microscopy in NLS1-HK2 NB4 cells after treatment with daunorubicin for 6 hours (50nM). n = 438 cells examined over 2 biologic repeats. (g) Baseline 53BP1 expression in 8227 stem and bulk controls. n = 510 cells examined over 3 biologic repeats. (h) Baseline 53BP1 expression in 8227 cells overexpressing NLS1- HK2. n = 879 cells examined over 4 biologic repeats. (i) Baseline RAD51 expression in 8227 stem and bulk controls . n = 331cells examined over 2 biologic repeats. (j) Baseline RAD51 expression in 8227 cells overexpressing NLS1- HK2. n = 671 cells examined over 3 biologic repeats. Statistical analyses for experiments was performed using an unpaired Student's t-test. Data shown in (d-e, g-h, j) represent median and interquartile range, data from (c) represent mean +/- s.e.m, data from (f) represent mean.

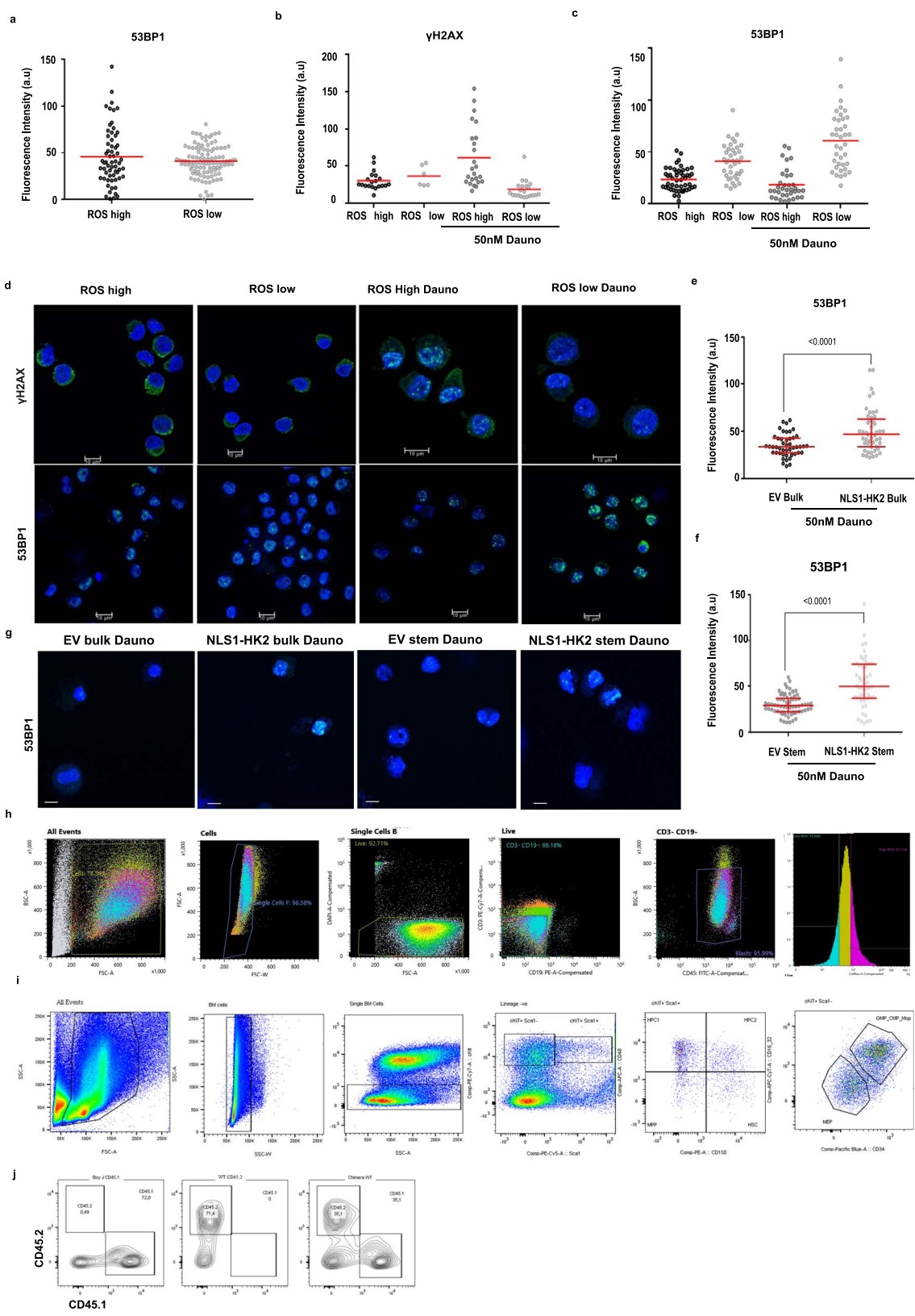

**Extended Data Fig. 10 | See next page for caption.**

**Extended Data Fig. 10 | AML stem cells demonstrate enhanced DNA repair.** (a) Baseline 53BP1 expression in stem and bulk primary patient control cells (AML151258). n = 170 cells examined from 1 of 2 biologic samples. (b) γH2AX levels were quantified by confocal microscopy in stem and bulk fractions of primary patient cells (AML161820) after treatment with daunorubicin for 0 and 3 hours (50nM). n = 71 cells examined from 2 of 2 biologic samples. (c) 53BP1 levels in stem and bulk primary patient cells (AML161820) at baseline and after treatment with daunorubicin for 3 hours (50nM). n = 161 cells examined from 2 of 2 biologic samples. (d) Representative confocal images of γH2AX and 53BP1 in primary patient cells (AML151258) sorted into stem and bulk populations based on ROS levels at baseline and 3 hours after treatment with daunorubicin. Scale Bar = 10μm. Representative images from 1 of 2 biologic samples. (e) 53BP1 levels in EV and NLS1-HK2 transduced sorted bulk 8227 cells and (f) sorted stem cells after 3-hour treatment with daunorubicin (50nM). n = 203 cells examined from 3 biologic replicates. (g) Representative confocal images of 53BP1 in transduced sorted 8227 cells 3 hours after treatment with daunorubicin (50nM). Scale bar = 10μm Representative images from 3 biologic replicates. (h) Gating strategy for ROS high and ROS low AML patient samples. (i) Gating strategy for mouse bone marrow hematopoietic analysis. (j) Gating strategy for competitive repopulation assay. Statistical analyses for experiments was performed using an unpaired Student's t-test. Data shown in (e-f) represent median and interquartile range, data from (a-c) represent mean.

# nature research

# Reporting Summary

Nature Research wishes to improve the reproducibility of the work that we publish. This form provides structure for consistency and transparency in reporting. For further information on Nature Research policies, see our Editorial Policies and the Editorial Policy Checklist.

## Statistics

For all statistical analyses, confirm that the following items are present in the figure legend, table legend, main text, or Methods section.

| n/a | Confirmed | |
|---|---|---|
| ☐ | ☒ | The exact sample size (*n*) for each experimental group/condition, given as a discrete number and unit of measurement |
| ☐ | ☒ | A statement on whether measurements were taken from distinct samples or whether the same sample was measured repeatedly |
| ☐ | ☒ | The statistical test(s) used AND whether they are one- or two-sided<br>*Only common tests should be described solely by name; describe more complex techniques in the Methods section.* |
| ☒ | ☐ | A description of all covariates tested |
| ☒ | ☐ | A description of any assumptions or corrections, such as tests of normality and adjustment for multiple comparisons |
| ☐ | ☒ | A full description of the statistical parameters including central tendency (e.g. means) or other basic estimates (e.g. regression coefficient) AND variation (e.g. standard deviation) or associated estimates of uncertainty (e.g. confidence intervals) |
| ☐ | ☒ | For null hypothesis testing, the test statistic (e.g. *F*, *t*, *r*) with confidence intervals, effect sizes, degrees of freedom and *P* value noted<br>*Give P values as exact values whenever suitable.* |
| ☒ | ☐ | For Bayesian analysis, information on the choice of priors and Markov chain Monte Carlo settings |
| ☒ | ☐ | For hierarchical and complex designs, identification of the appropriate level for tests and full reporting of outcomes |
| ☐ | ☒ | Estimates of effect sizes (e.g. Cohen's *d*, Pearson's *r*), indicating how they were calculated |

*Our web collection on statistics for biologists contains articles on many of the points above.*

## Software and code

Policy information about availability of computer code

| Data collection | Confocal Imaging: Leica X Software, Flow cytometry: BD FACS Diva Software, RPPA: Microvigene Software v3.0, MTS Assay: Softmax Pro v7.0.3. |
|---|---|
| Data analysis | Statistical Analysis: Graph Pad Prism, Flow cytometry: Flow Jo V7.7 & V10.7;<br>Confocal imaging: Image J 2.3.0, LAS AF, Metamorph v7.8;<br>Comet Assay: Open Comet 1.3.1,<br>Densitometry: Image Lab 5.2.1,  GREAT_4.0.4 (great.Stanford.edu)<br>RNA Seq:Reads were aligned against hg38 using Tophat v. 2.0.11. Read counts per gene were ob-tained through htseq-count v.0.6.1p2 in the mode ''intersection nonempty.'' EdgeR R package v.3.16.5 was used to normalize the data using the TMM (trimmed mean of M values) method and to estimate differential expression. R package GSVA_1.30.0 was used to analyze LSC+/LSC- signatures. Differential expression between LSC+ and LSC- fractions was calculated the limma R package 3.28.21.<br>BioID: RAW files were converted to the .mzML format using ProteoWizard (v3.0.10800) (43) and then searched using X! Tandem [X! TANDEM Jackhammer TPP (v2013.06.15.1)] 59 and Comet (v2014.02 rev.2) 60 against the human Human RefSeq v45 database (containing 36113 entries). Data were filtered through the trans-proteomic pipeline (TPP) (v4.7 POLAR VOR-TEX rev 1) .<br>ATAC seq: ATAC samples were preprocessed according to the ENCODE ATAC-seq pipeline. Single-end reads were aligned to the hg38 genome using Bowtie2 64, reads with MAPQ scores < 30 were filtered out with Samtools 65, duplicates were removed using Sambamba 66 and TN5 tagAlign shifted files were created. MACS2 67 was used to call peaks. Peak counts and sizes for each replicate were calculated using a custom Python script, and Jaccard indices for similarities between called peaks was calculated using BEDTools 68. Differentially accessible regions were calculated using the DiffBind and EdgeR 69 packages in R. Differentially accessible regions (DAR) were mapped to genes using the annotatePeak function of the R package ChIPseeker_1.22.1. Feature Distribution were plotted using the function plotAnnoBar. DARs were subjected to pathway analysis using the GREAT tool version 4.0.4. Pathways enriched at FDR 0.05 belonging to the category of GO Biological Processes (BP) were visualized as a network using Cytoscape 3.8.1 and EnrichmentMap 3.3.1and AutoAnnotate 1.3.3.<br>ChIP seq: Prior to analysis, read adapters were removed using Trim_Galore v. 0.4.0. Reads were aligned against hg38 (UCSC version) using Bowtie2 v2.3.2. Alignment reads were deduplicated to remove duplicate reads and keep unique reads us-ing picard v. 1.9.1. Broad peaks were |

identified from the alignment files using MACS2 v.2.1.1. Peaks were annotated with all the potential genomic features based on hg38 GENCODE v24 gene assembly which was downloaded from UCSC database. MACS2 called peaks at FRD 0.01 for individual samples and pooled samples were sub-jected to pathway analysis using the GREAT tool version 4.0.4. Overlapping peaks for the 2 replicates of MAX (ENCFF793GVV.bed) using the function findOverlappingPeaks from the ChIPpeakAnno_3.20.1 R package. The findMotifsGenome algorithm from HOMER v4.7 was used to identify known enriched motifs in genomic regions. consensus sequences of motifs were aligned using MAFFT (https://mafft.cbrc.jp/alignment/software/). TFmotifView (http://bardet.u-strasbg.fr/) was used to calculate the significance of motifs.

For manuscripts utilizing custom algorithms or software that are central to the research but not yet described in published literature, software must be made available to editors and reviewers. We strongly encourage code deposition in a community repository (e.g. GitHub). See the Nature Research guidelines for submitting code & software for further information.

## Data

Policy information about availability of data

All manuscripts must include a data availability statement. This statement should provide the following information, where applicable:

- Accession codes, unique identifiers, or web links for publicly available datasets
- A list of figures that have associated raw data
- A description of any restrictions on data availability

NCBI Accession number: NM_000189.5
BioGRID: https://thebiogrid.org/109346/summary/homo-sapiens/hk2.html
Omnibus dataset GSE24759
Gene Expression Omnibus dataset GSE76008
RNA seq-GSE176103
ATAC seq- GSE176071
ChIP seq- GSE176072

# Field-specific reporting

Please select the one below that is the best fit for your research. If you are not sure, read the appropriate sections before making your selection.

☒ Life sciences          ☐ Behavioural & social sciences          ☐ Ecological, evolutionary & environmental sciences

For a reference copy of the document with all sections, see nature.com/documents/nr-reporting-summary-flat.pdf

# Life sciences study design

All studies must disclose on these points even when the disclosure is negative.

| | |
|---|---|
| Sample size | No statistical methods were used to predetermine sample sizes, but our sample sizes are similar to those reported in previous publications. |
| Data exclusions | Data were not excluded |
| Replication | Data were replicated in at least 3 biologic replicates and technical replicates.  All experiments were reproducible. Key experiments were reproduced by different individuals |
| Randomization | For the in vivo experiments, the mice were randomly assigned to Control vs overexpression/knockdown models prior to intervention. The assignment and treatment of the mice was performed by an individual who was not involved in the analysis of the data from the experiment. For in vitro experiments, randomization was not applied to allocate samples into experimental groups. |
| Blinding | In vivo experiments were not blinded, however, experiments and analysis were performed independently.  For in vitro cell based experiments, the cell type and treatment condition were known because the experiments required the investigators to group the data between control and testing conditions to quantify differences. However, key experiments were reproduced independently by different individuals. |

# Reporting for specific materials, systems and methods

We require information from authors about some types of materials, experimental systems and methods used in many studies. Here, indicate whether each material, system or method listed is relevant to your study. If you are not sure if a list item applies to your research, read the appropriate section before selecting a response.

## Materials & experimental systems

| n/a | Involved in the study |
|---|---|
| ☐ | ☒ Antibodies |
| ☐ | ☒ Eukaryotic cell lines |
| ☒ | ☐ Palaeontology and archaeology |
| ☐ | ☒ Animals and other organisms |
| ☐ | ☒ Human research participants |
| ☒ | ☐ Clinical data |
| ☒ | ☐ Dual use research of concern |

## Methods

| n/a | Involved in the study |
|---|---|
| ☐ | ☒ ChIP-seq |
| ☐ | ☒ Flow cytometry |
| ☒ | ☐ MRI-based neuroimaging |

## Antibodies

| Antibodies used | Hexokinase II (C64G5) Rabbit mAb Cell signalling Cat #2867 WB 1:1000, IF 1:200<br>Mouse monoclonal anti- beta-Actin (AC-15)  Santa Cruz Biotechnology Cat# sc-69879 WB 1:10000<br>Rabbit polyclonal anti-beta Tubulin (H-235)  Santa Cruz Biotechnology  Cat# sc-9104 WB 1:5000<br>Histone H3 (D1H2) XP® Rabbit mAb Cell signalling Cat #4499 WB 1:5000<br>Mn SOD polyclonal antibody Enzo Cat#ADI-SOD-110-F WB 1:3000<br>Monoclonal ANTI-FLAG® M2 antibody produced in mouse Sigma Cat#F3165 WB 1:1000<br>53BP1 Antibody Novus Biologics Cat#NB100-304 IF 1:600<br>Anti-HK II Antibody (B-8)  Santa Cruz Biotechnology Cat#sc-374091 WB 1:1000<br>Anti-Aldolase A Antibody (C-10) Santa Cruz Biotechnology Cat#sc-390733 WB 1: 500<br>ACO2 Antibody Cell signalling Cat#6922 WB 1: 1000<br>Rabbit monoclonal anti-citrate synthase  Abcam  Cat# ab129095 WB 1: 1000<br>Enolase-1 Antibody Cell signalling Cat#3810 WB 1: 1000<br>GAPDH (14C10) Rabbit mAb Cell signalling Cat#2118 WB 1: 1000<br>PFKP Antibody  Cell signalling Cat#5412 WB 1: 500<br>IPO5 Antibody Thermo PA5-30076 WB 1: 1000<br>PHLPP1 Antibody Proteintech 22789-1-AP WB 1: 1000<br>PKM2 Antibody Abcam  Cat#3198 WB 1: 1000<br>GPI Antibody  Thermo Scientific Cat#PA5-29665 WB 1: 1000<br>Hexokinase I (C35C4) Rabbit mAb Cell signalling Cat#2024 WB 1: 1000<br>Anti-Hexokinase Type III/HK3 antibody  Abcam  Cat#ab91097 WB 1: 1000<br>Anti-Rad51 antibody Abcam  Cat#ab63801 IF 1:600<br>Anti-phospho-Histone H2A.X (Ser139) Antibody, clone JBW301 EMD Millipore Cat#05-636 IF 1:300<br>Anti-SDHA antibody [2E3GC12FB2AE2] Abcam  Cat#ab14715 WB 1: 1000<br>Anti-SDHB antibody [EPR13042(B)] Abcam  Cat#ab178423 WB 1: 1000<br>Mouse Anti-Tom20  Clone  29 BD Biosciences Cat#612278 IF 1: 400<br>Mouse anti-human CD45 BB515 (HI30)  BD Biosciences Cat#555413 Flow 1:100<br>Mouse anti-human CD3 PE-Cy7 (SP34-2)  BD Biosciences Cat#557749 Flow 1:100<br>Mouse anti-human CD19 PE (HIB19)  BD Biosciences Cat#555413 Flow 1:100<br>BB515 Rat Anti-Mouse Ly-6A/E  BD Biosciences  Cat# 565397 Flow 1:100<br>Pe/Cy7 Rat anti-mouse CD117 (c-Kit)  BioLegend  Cat# 105814  Flow 1:100<br>PE Rat anti-mouse CD150 (SLAM)  BioLegend  Cat# 115904  Flow 1:100<br>PE/Cy5 Anti-Mo-LY-6A/E (Sca-1) Invitrogen  Cat#15-5981-82 Flow 1:200<br>APC Hamster anti-mouse CD48  ThermorFisher Scientific  Cat# 17-0481-82  Flow 1:50<br>APC/Cy7 Rat anti-mouse CD16/32  BioLegend  Cat# 101328  Flow 1:200<br>eFluor 450 Rat anti-mouse CD34  ThermoreFisher Scientific  Cat# 48-0341-82  Flow 1:200<br>APC Mouse anti-human CD11b  BD Biosciences  Cat# 340937 Flow 1:200<br>APC Mouse anti-human CD14  BD Biosciences  Cat# 555399 Flow 1:100<br>CD34  BD Biosciences  Cat# 348053 Flow 1:200<br>CD38  BD Biosciences  Cat# 342371 Flow 1:100<br>FITC Mouse Anti-Human CD45RA  BD Biosciences  Cat# 555488 Flow 1:25<br>APC Mouse Anti-Human CD90 BD Biosciences  Cat#561971 Flow 1:50<br>PE-Cy™7 Mouse Anti-Human CD38 Clone  HB7 BD Biosciences  Cat#335790 Flow 1:200<br>V450 Mouse Anti-Human CD7 Clone  M-T701 BD Biosciences  Cat#642916 Flow 1:50<br>V500 Mouse Anti-Human CD45 Clone  HI30 BD Biosciences  Cat#560777 Flow 1:200<br>APC/Cyanine7 anti-human CD34 Clone 581 BioLegend  Cat#343513 Flow 1:100<br>CD34 Monoclonal Antibody (4H11), PerCP-eFluor 710 eBioscience Cat#46-0349-42 Flow 1:100<br>CD33-PE-Cy5  anti-human  Beckman Coulter Cat#PNIM2647U Flow 1:100<br>CD3 FITC Clone  SK7 BD Biosciences  Cat#349201 Flow 1:100<br>CD19 (Leu™-12) PE Clone  4G7  BD Biosciences  Cat#349209 Flow 1:200<br>Alexa Fluor®647 Mouse Anti-Human CD56 Clone  B159  BD Biosciences  Cat#557711 Flow 1:100<br>Qdot™ 605 Streptavidin Conjugate Invitrogen  Q10101MP Flow 1:200<br>Biotin anti-human CD135 (Flt-3/Flk-2) BioLegend  Cat#313312 Flow 1:10<br> anti-CD10-Alexa-700  BD Biosciences  624040 Flow 1:10<br>Pacific Blue™ anti-mouse CD45.1 Antibody  Biolegend  110721 Flow 1:100<br>APC/Cyanine7 anti-mouse CD45.2 Antibody  Biolegend  109823 Flow 1:100<br>MAX (Santa-Cruz, sc-197,1:200), SPIN1 (Cell signaling, 89139, ,1:100), CTR9 (Cell sig-naling, 12619, ,1:100), IWS1 (Cell signaling, 5681, ,1:100) |
|---|---|

| Validation | Antibodies were validated as per manufactures instructions. In addition, antibodies were validated by knocking down the target with shRNA, demonstrating a reduction in the band by immunoblotting. |
|---|---|

# Eukaryotic cell lines

Policy information about cell lines

| Cell line source(s) | OCI AML2 - Dr. Mark D. Minden, NB4 (ACC207) - DSMZ , K562 – ATCC , U937 - ATCC, TEX – Dr. John Dick, OCI-AML 8227 - Dr. John Dick, 130578 – Dr. Steven Chan, HL60 - ATCC, HEK293 - ATCC, KBM3- Dr. Michael Andreeff, KG1a - ATCC, ML2 -DSMZ, MOLM13 -Dr. Michael Andreeff, MOLM14 -Dr. Michael Andreeff, MV411 -ATCC, OCI-AML3 - DSMZ, OCI-AML5 - DSMZ, SKM1 - Dr. Garcia-Manero Guillermo, THP1 - ATCC, CCRF-CEM - Dr. Yiling Lu, Jukart - ATCC, MOLT4 - Dr. Michael Andreeff, TALL1 - Dr. Michael Andreeff |
|---|---|
| Authentication | Short Tandem Repeat (STR) Genotyping |
| Mycoplasma contamination | All cell lines tested negative for mycoplasma |
| Commonly misidentified lines (See ICLAC register) | No commonly misinterpreted cell lines were used in the study. |

# Animals and other organisms

Policy information about studies involving animals; ARRIVE guidelines recommended for reporting animal research

| Laboratory animals | NOD.Cg-Prkdcscid Il2rgtm1Wjl Tg(CMVIL3,CSF2,KITLG)1Eav/MloySzJ (NOD-SCID-GF) 6-12 week old male or female mice(1:1 ratio) were used for the engraftment studies. 5 week old male SCID mice were used in the tumor progression analysis. (AUP): # 1251.38 (NOD-SCID-GF and SCID mice). 5-10 week old male or female mice Vav NLS HK2 Mouse C57BL6 (CD45.2+) & 5 week old male B6.SJL-Ptprca Pepcb/BoyJ (CD45.1+) were used for the hematopoietic stem progenitor analysis and competitive repopulation assays. (AUP#2244.16) for B6 mice. The mice were housed in micro isolator cages with temperature-controlled conditions under a 12-hour light/dark cycle with access to drinking water and food. All animal studies were performed in accordance with the Ontario Cancer Institute Animal Use Protocol. |
|---|---|
| Wild animals | Not involved |
| Field-collected samples | Not involved |
| Ethics oversight | University Health Network ethical review committee & Ontario Cancer Institute Animal Use Protocol |

Note that full information on the approval of the study protocol must also be provided in the manuscript.

# Human research participants

Policy information about studies involving human research participants

| Population characteristics | This has been included in the Extended Data Table 2. |
|---|---|
| Recruitment | Primary human AML samples from peripheral blood or the bone marrow of male or female AML patients, were collected after obtaining informed consent. |
| Ethics oversight | University Health Network (REB # 01-0573) and MD Anderson Cancer Center Institutional Review Board reviewed and approved the collection protocol and the research usage protocol. |

Note that full information on the approval of the study protocol must also be provided in the manuscript.

# ChIP-seq

## Data deposition

☒ Confirm that both raw and final processed data have been deposited in a public database such as GEO.

☒ Confirm that you have deposited or provided access to graph files (e.g. BED files) for the called peaks.

| Data access links May remain private before publication. | Data have been deposited to Gene EXpression Omni-bus Database , RNA seq - GSE176103, ATAC seq - GSE176071 & ChIP seq - GSE176072. |
|---|---|
| Files in database submission | RNA Seq<br>PROCESSED DATA FILES<br>8227_GFP_2.tsv<br>8227_GFP_3.tsv<br>8227_SH1_1.tsv<br>8227_SH1_2.tsv<br>8227_SH1_3.tsv |

```
RAW FILES
8227_GFP_2_S4_L002_R1_001.fastq.gz
8227_GFP_2_S4_L001_R2_001.fastq.gz
8227_GFP_3_S7_L001_R1_001.fastq.gz
8227_GFP_3_S7_L001_R2_001.fastq.gz
8227_SH1_1_S2_L001_R1_001.fastq.gz
8227_SH1_1_S2_L001_R2_001.fastq.gz
8227_SH1_2_S5_L001_R1_001.fastq.gz
8227_SH1_2_S5_L001_R2_001.fastq.gz
8227_SH1_3_S8_L001_R1_001.fastq.gz
8227_SH1_3_S8_L001_R2_001.fastq.gz
PAIRED-END EXPERIMENTS
8227_GFP_2_S4_L002_R1_001.fastq.gz
8227_GFP_3_S7_L001_R1_001.fastq.gz
8227_SH1_1_S2_L001_R1_001.fastq.gz
8227_SH1_2_S5_L001_R1_001.fastq.gz
8227_SH1_3_S8_L001_R1_001.fastq.gz

ATAC Seq:
PROCESSED DATA FILES
EFa_1_S18_L001_R1_001.trim.merged.nodup.no_chrM_MT.tn5.pval0.01.300K.bfilt.narrowPeak.gz
EFa_2_S22_L001_R1_001.trim.merged.nodup.no_chrM_MT.tn5.pval0.01.300K.bfilt.narrowPeak.gz
Efa_3_S28_L001_R1_001.trim.merged.nodup.no_chrM_MT.tn5.pval0.01.300K.bfilt.narrowPeak.gz
PAA1_S17_L001_R1_001.trim.merged.nodup.no_chrM_MT.tn5.pval0.01.300K.bfilt.narrowPeak.gz
PAA2_S21_L001_R1_001.trim.merged.nodup.no_chrM_MT.tn5.pval0.01.300K.bfilt.narrowPeak.gz
PAA2_S21_L001_R1_001.trim.merged.nodup.no_chrM_MT.tn5.pval0.01.300K.bfilt.narrowPeak.gz
OM_GFP1_S20_L001_R1_001.trim.merged.nodup.no_chrM_MT.tn5.pval0.01.300K.bfilt.narrowPeak.gz
OM_GFP2_S24_L001_R1_001.trim.merged.nodup.no_chrM_MT.tn5.pval0.01.300K.bfilt.narrowPeak.gz
OM_GFP3_S26_L001_R1_001.trim.merged.nodup.no_chrM_MT.tn5.pval0.01.300K.bfilt.narrowPeak.gz
OM_SH1_1_S19_L001_R1_001.trim.merged.nodup.no_chrM_MT.tn5.pval0.01.300K.bfilt.narrowPeak.gz
OM_SH1_2_S23_L001_R1_001.trim.merged.nodup.no_chrM_MT.tn5.pval0.01.300K.bfilt.narrowPeak.gz
OM_SH1_3_S25_L001_R1_001.trim.merged.nodup.no_chrM_MT.tn5.pval0.01.300K.bfilt.narrowPeak.gz
RAW FILES
EFa_1_S18_L001_R1_001.fastq.gz
EFa_2_S22_L001_R1_001.fastq.gz
Efa_3_S28_L001_R1_001.fastq.gz
PAA1_S17_L001_R1_001.fastq.gz
PAA2_S21_L001_R1_001.fastq.gz
PAA3_S27_L001_R1_001.fastq.gz
OM_GFP1_S20_L001_R1_001.fastq.gz
OM_GFP2_S24_L001_R1_001.fastq.gz
OM_GFP3_S26_L001_R1_001.fastq.gz
OM_SH1_1_S19_L001_R1_001.fastq.gz
OM_SH1_2_S23_L001_R1_001.fastq.gz
OM_SH1_3_S25_L001_R1_001.fastq.gz
EFa_1_S18_L001_R2_001.fastq.gz
EFa_2_S22_L001_R2_001.fastq.gz
Efa_3_S28_L001_R2_001.fastq.gz
PAA1_S17_L001_R2_001.fastq.gz
PAA2_S21_L001_R2_001.fastq.gz
PAA3_S27_L001_R2_001.fastq.gz
OM_GFP1_S20_L001_R2_001.fastq.gz
OM_GFP2_S24_L001_R2_001.fastq.gz
OM_GFP3_S26_L001_R2_001.fastq.gz
OM_SH1_1_S19_L001_R2_001.fastq.gz
OM_SH1_2_S23_L001_R2_001.fastq.gz
OM_SH1_3_S25_L001_R2_001.fastq.gz

ChIP Seq:
PROCESSED DATA FILES
EV1_S1_L002_R1_001.nodup.pval0.01.500K.bfilt.narrowPeak.gz
EV2_S2_L002_R1_001.nodup.pval0.01.500K.bfilt.narrowPeak.gz
EV3_S3_L002_R1_001.nodup.pval0.01.500K.bfilt.narrowPeak.gz
PAA1_S4_L002_R1_001.nodup.pval0.01.500K.bfilt.narrowPeak.gz
PAA2_S5_L002_R1_001.nodup.pval0.01.500K.bfilt.narrowPeak.gz
PAA3_S6_L002_R1_001.nodup.pval0.01.500K.bfilt.narrowPeak.gz
NES1_S7_L002_R1_001.nodup.pval0.01.500K.bfilt.narrowPeak.gz
NES2_S8_L002_R1_001.nodup.pval0.01.500K.bfilt.narrowPeak.gz
NES3_S9_L002_R1_001.nodup.pval0.01.500K.bfilt.narrowPeak.gz
RAW FILES
EV1_S1_L002_R1_001.fastq.gz
EV1_S1_L002_R2_001.fastq.gz
EV2_S2_L002_R1_001.fastq.gz
EV2_S2_L002_R2_001.fastq.gz
EV3_S3_L002_R1_001.fastq.gz
EV3_S3_L002_R2_001.fastq.gz
```

PAA1_S4_L002_R1_001.fastq.gz
PAA1_S4_L002_R2_001.fastq.gz
PAA2_S5_L002_R1_001.fastq.gz
PAA2_S5_L002_R2_001.fastq.gz
PAA3_S6_L002_R1_001.fastq.gz
PAA3_S6_L002_R2_001.fastq.gz
NES1_S7_L002_R1_001.fastq.gz
NES1_S7_L002_R2_001.fastq.gz
NES2_S8_L002_R1_001.fastq.gz
NES2_S8_L002_R2_001.fastq.gz
NES3_S9_L002_R1_001.fastq.gz
NES3_S9_L002_R2_001.fastq.gz
MD5 (EV1_S1_L002_R1_001.fastq.gz) = 858e3c60355a007941cdfae01c4e6dd5
MD5 (EV1_S1_L002_R2_001.fastq.gz) = 9aa16e7bdebc1b6c5726b5a661fae0d2
MD5 (EV2_S2_L002_R1_001.fastq.gz) = c31cde4e4432927c898db8552a72b933
MD5 (EV2_S2_L002_R2_001.fastq.gz) = afc751195418de45ee5693c98eee7503
MD5 (EV3_S3_L002_R1_001.fastq.gz) = f721682b3625a639b56d781578021486
MD5 (EV3_S3_L002_R2_001.fastq.gz) = 968e879d179ee484bb92838300c7b7b7
MD5 (NES1_S7_L002_R1_001.fastq.gz) = 8af893be7084d8c62b4385ca70ddc7c3
MD5 (NES1_S7_L002_R2_001.fastq.gz) = 89be07d16e0cbce3dae91079e256bb30
MD5 (NES2_S8_L002_R1_001.fastq.gz) = 4e26266ea133b039b32334e70acfe816
MD5 (NES2_S8_L002_R2_001.fastq.gz) = 7a1404ff4ab3ac57c355037829fc4127
MD5 (NES3_S9_L002_R1_001.fastq.gz) = 7dac594817d0948ef66dc8fe203df874
MD5 (NES3_S9_L002_R2_001.fastq.gz) = 8aad1f3bb01c865e2b3e7a582dac5f0e
MD5 (PAA1_S4_L002_R1_001.fastq.gz) = d27f6f772303bbf7f1a8d444846316af
MD5 (PAA1_S4_L002_R2_001.fastq.gz) = 0b6d7e99f5b1860629d8d0cfdba0c001
MD5 (PAA2_S5_L002_R1_001.fastq.gz) = 25f1e3b0119eb4e7bdebe1ef2883fc06
MD5 (PAA2_S5_L002_R2_001.fastq.gz) = 1171d9dd8867e4eb13acde33c62d356e
MD5 (PAA3_S6_L002_R1_001.fastq.gz) = e48999515a6c319f820744810da8a53b
MD5 (PAA3_S6_L002_R2_001.fastq.gz) = 83098efad1946c696748a17844863c1d
MD5 (EV1_S1_L002_R1_001.nodup.pval0.01.500K.bfilt.narrowPeak.gz) = 878140f7492a73495ebce5e86f8b52b3
MD5 (EV2_S2_L002_R1_001.nodup.pval0.01.500K.bfilt.narrowPeak.gz) = 720ed3c41f86486b27e667608e3335ba
MD5 (EV3_S3_L002_R1_001.nodup.pval0.01.500K.bfilt.narrowPeak.gz) = c4e936481bbd1da6097042d8d22b5981
MD5 (NES1_S7_L002_R1_001.nodup.pval0.01.500K.bfilt.narrowPeak.gz) = fad39929407ad1803b26847af57a35df
MD5 (NES2_S8_L002_R1_001.nodup.pval0.01.500K.bfilt.narrowPeak.gz) = aea2fb7e6bc5509d62edd40c50dacf9d
MD5 (NES3_S9_L002_R1_001.nodup.pval0.01.500K.bfilt.narrowPeak.gz) = 9d77cfc1f24e57a76dc6e19b9fdfb5f9
MD5 (PAA1_S4_L002_R1_001.nodup.pval0.01.500K.bfilt.narrowPeak.gz) = 8d54ae9ec993498e43c57607d8fd571c
MD5 (PAA2_S5_L002_R1_001.nodup.pval0.01.500K.bfilt.narrowPeak.gz) = 357dfae969bbe0bd36423f07857c22f9
MD5 (PAA3_S6_L002_R1_001.nodup.pval0.01.500K.bfilt.narrowPeak.gz) = 91f188340369355731bb8a8d93a94a51

Genome browser session
(e.g. UCSC)

GEO

## Methodology

**Replicates**

3 Biological replicates

**Sequencing depth**

```
              rep1      rep2      rep3
EV   Total Reads 89178502 67951346 68028648
     Mapped Reads 85343440 66582031 66131862
NLS1 HK2
     Total Reads 60189652 77018382 75795786
     Mapped Reads 56367586 74916850 72362159
NLS2 HK2
      Total Reads 63210334 63827492 70883392
      Mapped Reads 61136782 62631788 69070112
```

**Antibodies**

Anti-HK II Antibody (B-8), Santa Cruz Biotechnology Cat#sc-374091

**Peak calling parameters**

Peak-calling on replicates and pooled data was carried out using MACS2 (v2.1.0). A p-value threshold of 0.01 was used for MACS2. Output files from MACS2 peak-calling include narrowpeak files, fold-enrichment and -log10(p-value) bigwig files. Once all bed files were converted to bigbed format, naïve overlap thresholding for the MACS2 peak calls was run.

**Data quality**

The initial quality control metrics for the ChIP-seq data were obtained using the tool FastQC (v0.11.5). All samples were processed using the official ChIP-seq pipeline specification of the Encyclopedia of DNA Elements (ENCODE) consortium, with QC report generated confirming quality metrics including alignment/peak statistics .The single-end BAM files were converted to tagAlign format (BED 3+3 format) and cross-correlation QC scores were calculated using phantompeakqualtools (v1.2).

**Software**

Read adapters were removed using Trim_Galore v. 0.4.0 with removing reads that have length less than 35 bp after trimming. In addition, a base pair quality score cutoff (q=30) was used for filtering low quality base pairs. Reads were aligned against hg38 (UCSC version) using Bowtie2 v2.3.2. Secondary and supplementary alignments were removed, and only primary alignments were kept. Alignment reads were deduplicated to remove duplicate reads and keep unique reads using picard v. 1.9.1. Broad peaks were

identified from the alignment files using MACS2 v.2.1.1 with a cutoff score q<0.05.MACS2 called peaks at FRD 0.01 for individual samples and pooled samples were sub-jected to pathway analysis using the GREAT tool version 4.0.4 GO BP pathways enriched at FDR 0.05 were visualized as a network using Cytoscape 3.8.1 and EnrichmentMap 3.3.1 and Au-toAnnotate 1.3.3. The findMotifsGenome algorithm from HOMER v4.7 was used to identify known enriched motifs in genomic regions in each individual samples.

# Flow Cytometry

## Plots

Confirm that:

☒ The axis labels state the marker and fluorochrome used (e.g. CD4-FITC).

☒ The axis scales are clearly visible. Include numbers along axes only for bottom left plot of group (a 'group' is an analysis of identical markers).

☒ All plots are contour plots with outliers or pseudocolor plots.

☒ A numerical value for number of cells or percentage (with statistics) is provided.

## Methodology

| | |
|---|---|
| Sample preparation | Cells were dispensed in FACS buffer stained with specific antibodies incubated, washed and resuspended in FACS buffer |
| Instrument | BD Forttessa X-20, BD FACS Aria 3, Sony SH-800, Cell Sorter,  MoFlo XDP Cell Sorter |
| Software | FlowJO_V10.7.1/ v7.7.1 |

Cell population abundance

| | Starting cell count | Stem cell | Bulk cells |
|---|---|---|---|
| OCI_AML_8227 | 80-100 million | ~5-10 million | ~12-15 million |
| AML_patient samples | 50-200 million | ~ 500,000-1 million | ~500,000-1 million |
| Cord blood samples CD34+ | ~5-6 million | | |
| Cord blood samples CD34- | ~40-60 million | | |

Gating strategy

The gating strategy was set as follows.
OCI_AML_8227 cells : FSC/SSC (represents the distribution of cells in the light scatter based on size and intracellular composition, respectively) to FSC Height/FSC width to SSC Height/SSC width (doublet discrimination) live gate ( DAPI negative,  viable cells within the sample analyzed) to CD34-APC-CY7/CD38-PE CY5 (identifies selective subsets, CD34+CD38- cells as Stem like and CD34-CD38+ as AML blasts).
AML Patient sample ROS High/Low: SC/SSC (represents the distribution of cells in the light scatter based on size and intracellular composition, respectively) to FSC Height/FSC width to SSC Height/SSC width (doublet discrimination) live gate ( DAPI negative, the fraction of viable cells within the sample analyzed) to B PE-CD1/ PE-Cy7-CD3  to ex-clude the lymphocyte populations B515-CD45 (to identify the blast population) to CellROX deep red (ROS-low LSCs - the cells with the 20% lowest ROS levels and the ROS-high blasts - the cells with the highest 20% ROS levels).
Cord Blood :
HSPC (CD34+) SORT
HSC CD34+CD38-CD45RA-CD90+CD49f+
MPP CD34+CD38-CD45RA-CD90-CD49f-
MLP CD34+CD38-CD45RA+
CMP CD34+CD38+CD7+CD10-FLT3+CD45RA-CD71-
GMP CD34+CD38+CD7+CD10-FLT3+CD45RA+CD71-CD19-
MEP CD34+CD38-CD7-FLT3-CD45RA-CD71+CD19- (CD7 and CD10)

Mature cells SORT
B CD19+
T CD3+
NK CD19-CD3-CD56+
Gran CD19-CD3-CD14+CD15+
Mono CD19-CD3-CD14+CD15-

☒ Tick this box to confirm that a figure exemplifying the gating strategy is provided in the Supplementary Information.

