## [Peer Review File · Nature Cell Biology]

Peer Review Information

Journal: Nature Cell Biology

Manuscript Title: The metabolic enzyme Hexokinase 2 localizes to the nucleus in AML and normal hematopoietic stem/progenitor cells to maintain stemness

Corresponding author name(s): Aaron Schimmer

Reviewer Comments & Decisions:

Decision Letter, initial version:

Dear Dr Schimmer,

Your manuscript, "The mitochondrial metabolic enzyme Hexokinase 2 regulates stem cell function and differentiation by increasing chromatin openness and the accessibility of stem cell genes", has now been seen by 3 referees, who are experts in LSCs, chromatin, metabolism (referee 1); cancer metabolism, mitochondria (referee 2); and leukemia, stem cells, metabolism (referee 3). As you will see from their comments (attached below) they find this work of potential interest, but have raised substantial concerns, which in our view would need to be addressed with considerable revisions before we can consider publication in Nature Cell Biology.

Nature Cell Biology editors discuss the referee reports in detail within the editorial team, including the chief editor, to identify key referee points that should be addressed with priority, and requests that are overruled as being beyond the scope of the current study. To guide the scope of the revisions, I have listed these points below. We are committed to providing a fair and constructive peer-review process, so please feel free to contact me if you would like to discuss any of the referee comments further.

In particular, it would be essential to address the following concerns:

(A) Additional experimental evidence should be provided to strengthen the role of HK2 in stem cell function and DNA damage repair, as highlighted by all three referees.

Referee 1 notes:

"5. The presence of increased phenotypic HSCs in transgenic Vav NLS-HK2 mice is striking but does not provide any evidence about functionality. The authors should perform competitive repopulation to demonstrate that the increase in phenotypic HSCs is associated with increased HSC activity.

6. The nuclear and cytoplasmic detection of HK2 (Figures 1B, 2B, 2C, 2F, ...) should be quantified on a

large number of cells in addition to providing a representative image. This has been done partly in figure 3A, although it would be better to display the nuclear fluorescent intensity value for each cell instead of just an average.”

Referee 2 notes:

“1. The abstract statement “nuclear HK2 increased repair of DNA damage and contributed to the mechanism by which LSCs resist DNA damaging agents” is not well-supported. None of the data presented constitutes a DNA repair assay, and there are a lot of reasons that these marks can be elevated or suppressed.

2. Related to the preceding point, if nuclear HK2 is indeed impacting stemness, it will be quite hard to predict whether nuclear HK2 is directly or indirectly genomic integrity because, as the authors point out, stem-like cells tend to resist genotoxic stress. This is of course a very hard question to answer, and therefore it would seem most appropriate for the authors to moderate their conclusions to reflect these limitations.

3. The title states, “Hexokinase 2 regulates stem cell function and differentiation by increasing chromatin openness and the accessibility of stem cell genes.” Here the authors are implying causality where no evidence exists. The authors have evidence that forced expression of nuclear HK2 results in several effects, among which are increased stemness and increased “openness” (a correlative relationship). However, it is not clear that the openness is requisite for “stem cell function and differentiation” or even that one event precedes the other.”

“1. The evidence that HK2 localizes to the nucleus is fairly strong, although it is difficult to appreciate the degree of co-localization in many of the figures. The authors should provide quantification for these fluorescent images with proper statistical tests to demonstrate that their conclusions are rigorously supported.”

Referee 3 notes:

“1. While the emphasis of the paper is on leukemic and normal stem cell function the assays used for these functions are very limited. The important experiments using knock-down of HK2 relied on In vitro and gene expression assays but not in vivo assays that more rigorously define stem cell function. The in vivo assays that were done rely on overexpression of nuclear HK2, but it is unclear if the levels resemble physiologic levels or if there are associated metabolic effects that may account for the functional changes. Firmer evidence for reliance on HK2 for stem cell function would give a stronger context for the elegant biochemistry that follows.”

“6. The authors show that nuclear HK2 enhances DNA damage repair, and at the same time that stem cell fraction has superior DNA damage repair compared with bulk AML cells. Because stem cells have a higher expression of nuclear HK2 than bulk cells, the authors state that AML stem cells have increased rates of DNA damage repair mediated by nuclear HK2. While this is a clear possibility, in order to conclude this the authors should modify nuclear HK2 levels in sorted stem and bulk fractions to rescue/restore their DNA damage repair capabilities.

7. Previous studies show that glucose levels can change the levels of nuclear HK2 (Sheikh T et al., 2018, Neary CL et al., 2013). Can the authors be sure that the differences in nuclear HK2 observed in stem vs bulk AML cells are not due to the media conditions?”

(B) The overall relevance of HK2 in AML and disease outcomes should be better supported with additional data, as requested by referees 1 and 3.

Referee 1 notes:

"1. It is not clear if nuclear HK2 is observed in all subtypes of AMLs. Is it more prevalent in AMLs with a more immature phenotype (M0/M1), or is it associated with a stem population in all the different AML subtypes?"

2. The authors demonstrate that nuclear HK2 over-expression in AML cells results in increased engraftment (Figure 1F). Does this translate into an acceleration of leukemia development and decreased survival?

3. Nuclear HK2 interactors have been identified in human embryonic kidney (HEK) cells. The authors suggest that these interactions are similar in human AML cells. This assumption is questionable. At least some of these interactions should be further demonstrated in AML cells, by Duolink proximity ligation assay (PLA), for example."

Referee 3 notes:

"4. The authors have shown that phosphorylation of HK2 is inversely related to nuclear localization and inhibiting AKT increases nuclear HK2. Since overall expression of HK2 does not seem to be increased in AML per public databases and AML is generally associated with hyperactivity of AKT/mTOR, how do AML cells have higher nuclear HK2?"

5. Human data that correlate HK2 levels with a better prognosis:
https://servers.binf.ku.dk/bloodspot/?gene=HK2&dataset=normal_human_v2_with_AMLs should be discussed given the proposed relationship of nuclear HK2 to therapy resistance suggested by the authors."

(C) Conclusions about the interaction of HK2 with nuclear proteins should be further validated and mechanistic insights improved, as requested by referees 1 and 3.

Referee 1 notes:

"[as above] 3. Nuclear HK2 interactors have been identified in human embryonic kidney (HEK) cells. The authors suggest that these interactions are similar in human AML cells. This assumption is questionable. At least some of these interactions should be further demonstrated in AML cells, by Duolink proximity ligation assay (PLA), for example.

4. The authors suggest that HK2 interacts with MAX at the chromatin. This should be further documented by comparing binding peaks of HK2 and MAX in AML CHIP-seq."

Referee 3 notes:

"2. The biochemical assays regarding interaction with other nuclear proteins and DNA depend upon overexpression of HK2. While these are interesting, it is not clear that these interactions occur at physiologic levels of HK2. Validation of the interactions with co-immunoprecipitation and CHIP experiments without overexpression are needed."

(D) The text should be edited to more adequately reflect the current literature and highlight the advance of this work over previous papers.

Referee 2 notes:

"2. The observation that HK2 localizes to the nucleus has been made before in mammalian cells, and the authors should place their work in the proper context. For example, there is a report that HK2 binds and

influences NRF2. See PMID: 29414774 and PMID: 20346347 for some examples of studies involving nuclear HK2.”

Referee 3 notes:

“3. In terms of novelty, the authors mentioned in the discussion that nuclear HK2 has been reported in yeast. However, nuclear localization of HK2 has also been described in glioma and other cancer cells (Sheikh T et al., 2018, Neary CL et al., 2013). In addition, some of the mechanism described in regard to HK2 nuclear import and export have also been reported; Akt inhibition-mediated nuclear localization (Neary CL et al., 2013) and XpoI-mediated nuclear export (Neary CL et al., 2010, Peláez R et al., 2009). While the findings found here relate specifically to AML cells, the authors should reference where appropriate these previous discoveries. These prior studies clearly impact the novelty of the findings here and further emphasize the importance of defining the cell biologic and functional consequences of HK2 modulation.”

(E) All other referee concerns pertaining to strengthening existing data, providing controls, methodological details, clarifications and textual changes should also be addressed.

(F) Finally please pay close attention to our guidelines on statistical and methodological reporting (listed below) as failure to do so may delay the reconsideration of the revised manuscript. In particular please provide:

We would be happy to consider a revised manuscript that would satisfactorily address these points, unless a similar paper is published elsewhere, or is accepted for publication in Nature Cell Biology in the meantime.

- ensure that it conforms to our format instructions and publication policies (see below and <https://www.nature.com/nature/for-authors>).

- provide a point-by-point rebuttal to the full referee reports verbatim, as provided at the end of this letter.

- provide the completed Reporting Summary (found here <https://www.nature.com/documents/nr-reporting-summary.pdf>). This is essential for reconsideration of the manuscript will be available to editors and referees in the event of peer review. For more information see <http://www.nature.com/authors/policies/availability.html> or contact me.

When submitting the revised version of your manuscript, please pay close attention to our [href="https://www.nature.com/nature-research/editorial-policies/image-integrity">Digital Image Integrity Guidelines](https://www.nature.com/nature-research/editorial-policies/image-integrity). and to the following points below:

Nature Cell Biology is committed to improving transparency in authorship. As part of our efforts in this direction, we are now requesting that all authors identified as 'corresponding author' on published papers create and link their Open Researcher and Contributor Identifier (ORCID) with their account on the Manuscript Tracking System (MTS), prior to acceptance. ORCID helps the scientific community achieve unambiguous attribution of all scholarly contributions. You can create and link your ORCID from the home page of the MTS by clicking on 'Modify my Springer Nature account'. For more information please visit www.springernature.com/orcid.

This journal strongly supports public availability of data. Please place the data used in your paper into a public data repository, or alternatively, present the data as Supplementary Information. If data can only be shared on request, please explain why in your Data Availability Statement, and also in the correspondence with your editor. Please note that for some data types, deposition in a public repository is mandatory - more information on our data deposition policies and available repositories appears below.

[REDACTED]

We would like to receive a revised submission within six months.

We hope that you will find our referees' comments, and editorial guidance helpful. Please do not hesitate to contact me if there is anything you would like to discuss.

With best wishes,

Christine.

Christine Weber, PhD
Senior Editor
Nature Cell Biology
E-mail: christine.weber@nature.com
Phone: +44 (0)207 843 4924

Reviewers' Comments:

Reviewer #1:
Remarks to the Author:

In this manuscript, the authors demonstrate that the mitochondrial metabolic enzyme Hexokinase 2 (HK2) localizes to the nucleus in leukemic and normal hematopoietic stem cells, where it regulates stem cell function and differentiation by increasing chromatin openness and the accessibility of stem cell genes. The authors also suggest that nuclear HK2 increases DNA damage repair rate and confers resistance to chemotherapy. Thus, in addition to regulating chromatin openness, HK2 also positively regulates the DNA damage response.

The manuscript is very well written and the findings are novel and exciting.

I have several comments and concerns.

1. It is not clear if nuclear HK2 is observed in all subtypes of AMLs. Is it more prevalent in AMLs with a more immature phenotype (M0/M1), or is it associated with a stem population in all the different AML subtypes?
2. The authors demonstrate that nuclear HK2 over-expression in AML cells results in increased engraftment (Figure 1F). Does this translate into an acceleration of leukemia development and decreased survival?
3. Nuclear HK2 interactors have been identified in human embryonic kidney (HEK) cells. The authors suggest that these interactions are similar in human AML cells. This assumption is questionable. At least some of these interactions should be further demonstrated in AML cells, by Duolink proximity ligation assay (PLA), for example.
4. The authors suggest that HK2 interacts with MAX at the chromatin. This should be further documented by comparing binding peaks of HK2 and MAX in AML ChIP-seq.
5. The presence of increased phenotypic HSCs in transgenic Vav NLS-HK2 mice is striking but does not provide any evidence about functionality. The authors should perform competitive repopulation to demonstrate that the increase in phenotypic HSCs is associated with increased HSC activity.
6. The nuclear and cytoplasmic detection of HK2 (Figures 1B, 2B, 2C, 2F, ...) should be quantified on a large number of cells in addition to providing a representative image. This has been done partly in figure 3A, although it would be better to display the nuclear fluorescent intensity value for each cell instead of just an average.

Reviewer #2:

Remarks to the Author:

In their manuscript Thomas et al describe the identification of glycolytic enzyme HK2 as a nuclear factor with activity in modulating gene expression. The authors determine that HK2 is unique among a panel of metabolic enzymes in its nuclear localization and that the degree of HK2 nuclear localization correlates with stemness in cancer cell models. Forced expression of nuclear HK2 increases the ability of leukemic cells to proliferate in animals, while forced expression of mitochondrially localized HK2 in the absence of nuclear HK2 reduces stemness. The authors conclude that these nuclear effects are independent of HK2 catalytic activity as the same phenotype is observed in an HK2 catalytic mutant. The authors go on to identify IPO5 as a protein involved in HK2 subcellular localization. They then perform transcriptomic analysis to identify the putative HK2 nuclear binding site and identify DNA repair as an HK2 regulated process. They provide evidence that this change in gene expression has functional relevance as forced expression of HK2 in the nucleus impacts pH2AX and RAD51 levels.

Overall, I think the authors have made an interesting set of observations. My major criticism, detailed below, is that the authors tend to over-interpret their data, implying an understanding of mechanism when their (many powerful) observations are largely correlative. Correcting this deficiency could entail

moderating their conclusions or providing more mechanistic studies. Therefore, I have a few points that I think should be addressed by the authors.

Major Points

1. The abstract statement "nuclear HK2 increased repair of DNA damage and contributed to the mechanism by which LSCs resist DNA damaging agents" is not well-supported. None of the data presented constitutes a DNA repair assay, and there are a lot of reasons that these marks can be elevated or suppressed.
2. Related to the preceding point, if nuclear HK2 is indeed impacting stemness, it will be quite hard to predict whether nuclear HK2 is directly or indirectly genomic integrity because, as the authors point out, stem-like cells tend to resist genotoxic stress. This is of course a very hard question to answer, and therefore it would seem most appropriate for the authors to moderate their conclusions to reflect these limitations.
3. The title states, "Hexokinase 2 regulates stem cell function and differentiation by increasing chromatin openness and the accessibility of stem cell genes." Here the authors are implying causality where no evidence exists. The authors have evidence that forced expression of nuclear HK2 results in several effects, among which are increased stemness and increased "openness" (a correlative relationship). However, it is not clear that the openness is requisite for "stem cell function and differentiation" or even that one event precedes the other.

Minor Points

1. The evidence that HK2 localizes to the nucleus is fairly strong, although it is difficult to appreciate the degree of co-localization in many of the figures. The authors should provide quantification for these fluorescent images with proper statistical tests to demonstrate that their conclusions are rigorously supported.
2. The observation that HK2 localizes to the nucleus has been made before in mammalian cells, and the authors should place their work in the proper context. For example, there is a report that HK2 binds and influences NRF2. See PMID: 29414774 and PMID: 20346347 for some examples of studies involving nuclear HK2.
3. I have to admit that I am having a lot of trouble interpreting Figure 5E-G. Why are nodes enriched in both the control and NLS HK2 conditions? I would have expected some comparative analysis of EV versus NLS HK2.

Reviewer #3:

Remarks to the Author:

Thomas GE et al., described an interesting non-canonical role for hexokinase 2 (HK2) in the regulation of AML stem cells. The authors found an accumulation of HK2 in the nucleus of AML cells and used shRNA and overexpression of nuclear HK2 to demonstrate that HK2 maintains their differentiation block and clonogenic capabilities in vitro. Chromatin analysis with overexpression of nuclear HK2 showed increased accessibility at regions linked to LSC signature and DNA damage repair genes. Finally, overexpressing nuclear HK2 enhanced the repair of DNA damage of AML cells treated with Daunorubicin.

The results are interesting but there are a number of concerns:

1. While the emphasis of the paper is on leukemic and normal stem cell function the assays used for these functions are very limited. The important experiments using knock-down of HK2 relied on In vitro and gene expression assays but not in vivo assays that more rigorously define stem cell function. The in vivo assays that were done rely on overexpression of nuclear HK2, but it is unclear if the levels resemble physiologic levels or if there are associated metabolic effects that may account for the functional changes. Firmer evidence for reliance on HK2 for stem cell function would give a stronger context for the elegant biochemistry that follows.
2. The biochemical assays regarding interaction with other nuclear proteins and DNA depend upon

overexpression of HK2. While these are interesting, it is not clear that these interactions occur at physiologic levels of HK2. Validation of the interactions with co-immunoprecipitation and CHIP experiments without overexpression are needed.

3. In terms of novelty, the authors mentioned in the discussion that nuclear HK2 has been reported in yeast. However, nuclear localization of HK2 has also been described in glioma and other cancer cells (Sheikh T et al., 2018, Neary CL et al., 2013). In addition, some of the mechanism described in regard to HK2 nuclear import and export have also been reported; Akt inhibition-mediated nuclear localization (Neary CL et al., 2013) and XpoI-mediated nuclear export (Neary CL et al., 2010, Peláez R et al., 2009). While the findings found here relate specifically to AML cells, the authors should reference where appropriate these previous discoveries. These prior studies clearly impact the novelty of the findings here and further emphasize the importance of defining the cell biologic and functional consequences of HK2 modulation.

4. The authors have shown that phosphorylation of HK2 is inversely related to nuclear localization and inhibiting AKT increases nuclear HK2. Since overall expression of HK2 does not seem to be increased in AML per public databases and AML is generally associated with hyperactivity of AKT/mTOR, how do AML cells have higher nuclear HK2?

5. Human data that correlate HK2 levels with a better prognosis: https://servers.binf.ku.dk/bloodspot/?gene=HK2&dataset=normal_human_v2_with_AMLs should be discussed given the proposed relationship of nuclear HK2 to therapy resistance suggested by the authors.

6. The authors show that nuclear HK2 enhances DNA damage repair, and at the same time that stem cell fraction has superior DNA damage repair compared with bulk AML cells. Because stem cells have a higher expression of nuclear HK2 than bulk cells, the authors state that AML stem cells have increased rates of DNA damage repair mediated by nuclear HK2. While this is a clear possibility, in order to conclude this the authors should modify nuclear HK2 levels in sorted stem and bulk fractions to rescue/restore their DNA damage repair capabilities.

7. Previous studies show that glucose levels can change the levels of nuclear HK2 (Sheikh T et al., 2018, Neary CL et al., 2013). Can the authors be sure that the differences in nuclear HK2 observed in stem vs bulk AML cells are not due to the media conditions?

Methods should be written concisely, but should contain all elements necessary to allow interpretation and replication of the results. As a guideline, Methods sections typically do not exceed 3,000 words. The Methods should be divided into subsections listing reagents and techniques. When citing previous methods, accurate references should be provided and any alterations should be noted. Information must be provided about: antibody dilutions, company names, catalogue numbers and clone numbers for monoclonal antibodies; sequences of RNAi and cDNA probes/primers or company names and catalogue numbers if reagents are commercial; cell line names, sources and information on cell line identity and authentication. Animal studies and experiments involving human subjects must be reported in detail, identifying the committees approving the protocols. For studies involving human subjects/samples, a

statement must be included confirming that informed consent was obtained. Statistical analyses and information on the reproducibility of experimental results should be provided in a section titled "Statistics and Reproducibility".

All Nature Cell Biology manuscripts submitted on or after March 21 2016 must include a Data availability statement as a separate section after Methods but before references, under the heading "Data Availability". For Springer Nature policies on data availability see <http://www.nature.com/authors/policies/availability.html>; for more information on this particular policy see <http://www.nature.com/authors/policies/data/data-availability-statements-data-citations.pdf>. The Data availability statement should include:

- Accession codes for primary datasets (generated during the study under consideration and designated as "primary accessions") and secondary datasets (published datasets reanalysed during the study under consideration, designated as "referenced accessions"). For primary accessions data should be made public to coincide with publication of the manuscript. A list of data types for which submission to community-endorsed public repositories is mandated (including sequence, structure, microarray, deep sequencing data) can be found here <http://www.nature.com/authors/policies/availability.html#data>.
- Unique identifiers (accession codes, DOIs or other unique persistent identifier) and hyperlinks for datasets deposited in an approved repository, but for which data deposition is not mandated (see here for details <http://www.nature.com/sdata/data-policies/repositories>).
- At a minimum, please include a statement confirming that all relevant data are available from the authors, and/or are included with the manuscript (e.g. as source data or supplementary information), listing which data are included (e.g. by figure panels and data types) and mentioning any restrictions on availability.
- If a dataset has a Digital Object Identifier (DOI) as its unique identifier, we strongly encourage including this in the Reference list and citing the dataset in the Methods.

We recommend that you upload the step-by-step protocols used in this manuscript to the Protocol Exchange. More details can found at www.nature.com/protocolexchange/about.

All imaging data should be accompanied by scale bars, which should be defined in the legend. Cropped images of gels/blots are acceptable, but need to be accompanied by size markers, and to retain visible background signal within the linear range (i.e. should not be saturated). The boundaries of panels with low background have to be demarked with black lines. Splicing of panels should only be considered if unavoidable, and must be clearly marked on the figure, and noted in the legend with a statement on whether the samples were obtained and processed simultaneously. Quantitative comparisons between samples on different gels/blots are discouraged; if this is unavoidable, it should only be performed for samples derived from the same experiment with gels/blots were processed in parallel, which needs to be stated in the legend.

Figures should be provided at approximately the size that they are to be printed at (single column is 86 mm, double column is 170 mm) and should not exceed an A4 page (8.5 x 11"). Reduction to the scale

that will be used on the page is not necessary, but multi-panel figures should be sized so that the whole figure can be reduced by the same amount at the smallest size at which essential details in each panel are visible. In the interest of our colour-blind readers we ask that you avoid using red and green for contrast in figures. Replacing red with magenta and green with turquoise are two possible colour-safe alternatives. Lines with widths of less than 1 point should be avoided. Sans serif typefaces, such as Helvetica (preferred) or Arial should be used. All text that forms part of a figure should be rewritable and removable.

- For line art, graphs, charts and schematics we prefer Adobe Illustrator (.AI), Encapsulated PostScript (.EPS) or Portable Document Format (.PDF). Files should be saved or exported as such directly from the application in which they were made, to allow us to restyle them according to our journal house style.
- We accept PowerPoint (.PPT) files if they are fully editable. However, please refrain from adding PowerPoint graphical effects to objects, as this results in them outputting poor quality raster art. Text used for PowerPoint figures should be Helvetica (preferred) or Arial.
- We do not recommend using Adobe Photoshop for designing figures, but we can accept Photoshop generated (.PSD or .TIFF) files only if each element included in the figure (text, labels, pictures, graphs, arrows and scale bars) are on separate layers. All text should be editable in 'type layers' and line-art such as graphs and other simple schematics should be preserved and embedded within 'vector smart objects' - not flattened raster/bitmap graphics.
- Some programs can generate Postscript by 'printing to file' (found in the Print dialogue). If using an application not listed above, save the file in PostScript format or email our Art Editor, Allen Beattie for advice (a.beattie@nature.com).

SUPPLEMENTARY INFORMATION – Supplementary information is material directly relevant to the conclusion of a paper, but which cannot be included in the printed version in order to keep the manuscript concise and accessible to the general reader. Supplementary information is an integral part of a Nature Cell Biology publication and should be prepared and presented with as much care as the main display item, but it must not include non-essential data or text, which may be removed at the editor's discretion. All supplementary material is fully peer-reviewed and published online as part of the HTML version of the manuscript. Supplementary Figures and Supplementary Notes are appended at the

end of the main PDF of the published manuscript.

The total number of Supplementary Figures (not including the “unprocessed scans” Supplementary Figure) should not exceed the number of main display items (figures and/or tables (see our Guide to Authors and March 2012 editorial <http://www.nature.com/ncb/authors/submit/index.html#suppinfo>; <http://www.nature.com/ncb/journal/v14/n3/index.html#ed>). No restrictions apply to Supplementary Tables or Videos, but we advise authors to be selective in including supplemental data.

GUIDELINES FOR EXPERIMENTAL AND STATISTICAL REPORTING

REPORTING REQUIREMENTS – We are trying to improve the quality of methods and statistics reporting in our papers. To that end, we are now asking authors to complete a reporting summary that collects information on experimental design and reagents. The Reporting Summary can be found here <https://www.nature.com/documents/nr-reporting-summary.pdf> If you would like to reference the guidance text as you complete the template, please access these flattened versions at <http://www.nature.com/authors/policies/availability.html>.

We strongly recommend the presentation of source data for graphical and statistical analyses as a separate Supplementary Table, and request that source data for all independent repeats are provided

when representative experiments of multiple independent repeats, or averages of two independent experiments are presented. This supplementary table should be in Excel format, with data for different figures provided as different sheets within a single Excel file. It should be labelled and numbered as one of the supplementary tables, titled "Statistics Source Data", and mentioned in all relevant figure legends.

Author Rebuttal to Initial comments

Reviewers' Comments

Reviewer #1:

Remarks to the Author:

In this manuscript, the authors demonstrate that the mitochondrial metabolic enzyme Hexokinase 2 (HK2) localizes to the nucleus in leukemic and normal hematopoietic stem cells, where it regulates stem cell function and differentiation by increasing chromatin openness and the accessibility of stem cell genes. The authors also suggest that nuclear HK2 increases DNA damage repair rate and confers resistance to chemotherapy. Thus, in addition to regulating chromatin openness, HK2 also positively regulates the DNA damage response.

The manuscript is very well written and the findings are novel and exciting.

I have several comments and concerns.

1. It is not clear if nuclear HK2 is observed in all subtypes of AMLs. Is it more prevalent in AMLs with a more immature phenotype (M0/M1), or is it associated with a stem population in all the different AML subtypes?

Response: Thank you for this comment. We agree that it is an interesting question. Unfortunately, the FAB subtype classification was not available for the samples we used for RPPA analysis where we measured the expression of nuclear HK2 protein. The public databases that include FAB classification are based on gene expression and do not have nuclear and cytoplasmic protein expression. However, using normal cord blood cells fractionated into different cellular compartments, we demonstrated that nuclear HK2 was higher in the stem/progenitor cells compared to more mature cells (Fig 3A, S4A).

2. The authors demonstrate that nuclear HK2 over-expression in AML cells results in increased engraftment (Figure 1F). Does this translate into an acceleration of leukemia development and decreased survival?

Response: We agree with the reviewer that measuring the impact of nuclear HK2 on leukemia development and survival in mice is important. In response, we performed two new in vivo experiments. First, we over-expressed nuclear HK2 in TEX leukemia cells and injected the cells into NSGF mice. Survival was measured over time. Over-expression of nuclear HK2 decreased survival, compared to empty vector (EV) controls. These data are included in Fig S2P-Q.

Second, we selectively knocked down nuclear HK2 cells in OCI-AML2 cells and xenografted the cells into SCID mice. Compared to wild type controls, selective knockdown of nuclear HK2 delayed the growth of leukemia in vivo. These data are shown in Fig 2G, S2K-L.

3. Nuclear HK2 interactors have been identified in human embryonic kidney (HEK) cells. The authors suggest that these interactions are similar in human AML cells. This assumption is questionable. At least some of these interactions should be further demonstrated in AML cells, by Duolink proximity ligation assay (PLA), for example.

Response: Thank you for this suggestion. We agree it is important to validate in AML cells the interactions identified in HEK cells. We are grateful to the reviewers for suggesting the PLA assay and we agree it is a good method to validate the hits from BioID.

In response to the reviewer's comment, we performed Duolink proximity ligation assay in AML cells to confirm nuclear HK2 interactors identified as hits in our BioID assay (MAX, SIRT1, IWS1, CTR9 and SPIN1). AML cells were incubated with primary HK2 antibody and antibodies to MAX, SIRT1, IWS1, CTR9 and SPIN1 or single antibody controls. Cells were then stained with PLA probes and hybridized DNA was amplified and visualized on confocal microscopy.

Our PLA results demonstrated that in AML cells, endogenous HK2 interacts with key proteins identified on BIO ID. We also quantified the intensity of the fluorescent signal. We included these new data in Fig 5B, S8A.

4. The authors suggest that HK2 interacts with MAX at the chromatin. This should be further documented by comparing binding peaks of HK2 and MAX in AML ChIP-seq.

Response: Thank you for this comment. As described above, we used the PLA assay to confirm the interaction between HK2 and MAX.

In response to this specific suggestion, we performed ChIP-seq with HK2 in NB4 cells and compared with the MAX ChIP-seq data in NB4 cells from the ENCODE ENCSR000EHS dataset and used the same read alignment protocol. Specifically, we compared DNA binding sites between endogenous HK2 and MAX. The overlap pattern across the whole genome between HK2 and MAX was visualized using IGV_2.8.4. Even with a very high 0 base pair (bp) gap distance stringency analysis, we identified 286 common peaks between these two proteins and the number of common peaks increased as the stringency of the analysis decreased. The majority of these peaks were located in promoter regions of genes.

These common peaks were identified in pathways important for DNA damage response, regulation of stem cell differentiation, and mitotic cell cycle at 0bp gap distance. The very low false discovery rate (FDR) threshold (0.000001) for this GSEA pathway analysis indicates a very significant functional overlap between HK2 and MAX. We have shown these new data in Fig S9J-M.

A

Peaks overlap with MAXNB4:	0bp	100bp	250bp	500bp	total peaks
NLS1 HK2 1,2,3	286	374	464	620	17561
DNA damage response	50	64	88	116	3837
Negative regulation of mitotic cell cycle	19	28	39	61	2490
Regulation of mitotic cell cycle	12	17	27	41	1486
Regulation of stem cell differentiation	6	8	11	17	606

B

C

D

5. The presence of increased phenotypic HSCs in transgenic Vav NLS-HK2 mice is striking but does not provide any evidence about functionality. The authors should perform competitive repopulation to demonstrate that the increase in phenotypic HSCs is associated with increased HSC activity.

Response: Thank you for this suggestion. We agree that additional experiments to confirm functionality of hematopoietic-expressed nuclear HK2 cells are important to solidify our conclusions.

We performed a competitive repopulation assay in transgenic VAV NLS-HK2 mice. For a competitive transplant, bone marrow cells from donor mice (CD45.2+; C57B6 background; VAV-NLS HK2 OR Wildtype) and clonogenic competitor mice (CD45.1+; B6.SJL) were mixed in a 1:1 ratio and injected into the tail vein of irradiated recipient mice (CD45.1 B6.SJL). Marrow and blood reconstitution was determined in peripheral blood (PB) samples taken 4, 6, 8, 10 weeks post-transplant and in the bone marrow at the end of the experiment, 12 weeks after transplant.

Compared to wildtype chimera mouse, VAV NLS-HK2 chimera mice demonstrated increased repopulation of peripheral blood at 4, 6, 8, and 10 weeks. We also demonstrated that nuclear HK2 increased the repopulation efficiency in the bone marrow at 12 weeks post transplantation. We included these data in Fig 4G-H.

6. The nuclear and cytoplasmic detection of HK2 (Figures 1B, 2B, 2C, 2F, ...) should be quantified on a large number of cells in addition to providing a representative image. This has been done partly in figure 3A, although it would be better to display the nuclear fluorescent intensity value for each cell instead of just an average.

Response: Thank you for this comment and we agree with the recommendation. In response, we quantified nuclear HK2 levels by fluorescent intensity and performed region of interest (ROI) analysis using ImageJ.

Fluorescent intensity analysis confirmed that nuclear HK2 levels were significantly higher in stem vs bulk 8227 cells and primary patient samples. Nuclear HK2 levels were also significantly higher in transduced over-expressed NB4 cells (NLS1-HK2, NLS2-HK2 and

NLS1-D290A/D657A). Finally, nuclear HK2 levels were lower in NB4 and 8227 after selective knockdown of nuclear HK2.

Reviewer #2:

Remarks to the Author:

In their manuscript Thomas et al describe the identification of glycolytic enzyme HK2 as a nuclear factor with activity in modulating gene expression. The authors determine that HK2 is unique among a panel of metabolic enzymes in its nuclear localization and that the degree of HK2 nuclear localization correlates with stemness in cancer cell models. Forced expression of nuclear HK2 increases the ability of leukemic cells to proliferate in animals, while forced expression of mitochondrially localized HK2 in the absence of nuclear HK2 reduces stemness. The authors conclude that these nuclear effects are independent of HK2 catalytic activity as the same phenotype is observed in an HK2 catalytic mutant. The authors go on to identify IPO5 as a protein involved in HK2 subcellular localization. They then perform transcriptomic analysis to identify the putative HK2 nuclear binding site and identify DNA repair as an HK2 regulated process. They provide evidence that this change in gene expression has functional relevance as forced expression of HK2 in the nucleus impacts pH2AX and RAD51 levels.

Overall, I think the authors have made an interesting set of observations. My major criticism, detailed below, is that the authors tend to over-interpret their data, implying an understanding of mechanism when their (many powerful) observations are largely correlative. Correcting this deficiency could entail moderating their conclusions or providing more mechanistic studies. Therefore, I have a few points that I think should be addressed by the authors.

Major Points

1. The abstract statement “nuclear HK2 increased repair of DNA damage and contributed to the mechanism by which LSCs resist DNA damaging agents” is not well-supported. None of the data presented constitutes a DNA repair assay, and there are a lot of reasons that these marks can be elevated or suppressed.

Response: Thank you for this comment. We agree that it is important to measure functional DNA repair in addition to markers of DNA damage. In response, we performed a comet assay to quantify and analyze DNA damage in individual cells, using single cell gel electrophoresis. We confirmed a decrease in double strand breaks in cells over-expressing nuclear HK2 after exposure to daunorubicin, compared to control cells. The results of the comet assay are shown in Fig 6H, S10B.

Furthermore, to address the reviewer’s concern, we changed the wording in the paper so that our statements regarding DNA repair are softer. The abstract now reads “over-expression of nuclear HK2 resulted in decreased double strand breaks and conferred chemoresistance’ (Page no 2: line no: 36-38). In addition, we changed the wording in the discussion portion to ensure the statements we make are reflective of the available supportive evidence (Page no 14: line no:376-381).

2. Related to the preceding point, if nuclear HK2 is indeed impacting stemness, it will be quite hard to predict whether nuclear HK2 is directly or indirectly genomic integrity because, as the authors point out, stem-like cells tend to resist genotoxic stress. This is of course a very hard question to answer, and therefore it would seem most appropriate for the authors to moderate their conclusions to reflect these limitations.

Response: Thank you for this comment. We agree that it is difficult to distinguish causation from correlation, when interpreting some of our analyses.

We appreciate the reviewer’s comments and in order not to overinflate our results we changed the wording in the abstract, results and discussion section (highlighted in the manuscript).

3. The title states, “Hexokinase 2 regulates stem cell function and differentiation by increasing chromatin openness and the accessibility of stem cell genes.” Here the authors are implying causality where no evidence exists. The authors have evidence that forced expression of nuclear HK2 results in several effects, among which are increased stemness and increased “openness” (a correlative relationship). However, it is not clear that the openness is requisite for “stem cell function and differentiation” or even that one event precedes the other.

Response: We thank the reviewer for this comment. We agree that the functional effects observed do not imply causality. We changed the title in accordance with these comments. The new title reads:

“The metabolic enzyme Hexokinase 2 localizes to the nucleus in AML and normal hematopoietic stem/progenitor cells to maintain stemness”

Minor Points

1. The evidence that HK2 localizes to the nucleus is fairly strong, although it is difficult to appreciate the degree of co-localization in many of the figures. The authors should provide quantification for these fluorescent images with proper statistical tests to demonstrate that their conclusions are rigorously supported.

Response: Thank you for this comment. We quantified nuclear HK2 levels by fluorescent intensity and performed region of interest (ROI) analysis using ImageJ.

Fluorescent intensity analysis confirms that nuclear HK2 levels were significantly higher in stem vs bulk 8227 cells and primary patient samples. Nuclear HK2 levels were also significantly higher in transduced over-expressed NB4 cells (NLS1-HK2, NLS2-HK2 and NLS1-D290A/D657A). Finally, nuclear HK2 levels were significantly lower in NB4 and 8227 after the selective knockdown of nuclear HK2. We have added these new data to Fig S1H.

2. The observation that HK2 localizes to the nucleus has been made before in mammalian cells, and the authors should place their work in the proper context. For example, there is a report that HK2 binds and influences NRF2. See PMID: 29414774 and PMID: 20346347 for some examples of studies involving nuclear HK2.

Response: Thank you for highlighting these papers. We included them in the results section and discussion (page no:14 line no: 361-365).

3. I have to admit that I am having a lot of trouble interpreting Figure 5E-G. Why are nodes enriched in both the control and NLS HK2 conditions? I would have expected some comparative analysis of EV versus NLS HK2.

Response: We apologize for the confusion and we created a new figure for the ATAC-seq data. Genes near peaks of open chromatin were identified in cases of endogenous HK2 levels (EV) which represents the signal at the physiological level and over-expressed nuclear HK2 levels (NLS1-HK2). The enrichment map is showing all pathways represented as nodes that are significant in at least one condition at FDR <0.000001. The size of the pie chart slices indicates the score for each sample, and the color indicates blue for NLS1-HK2 and yellow for EV. Both NLS1-HK2 and EV have different pattern of pathway enrichments (left and right parts of the figure respectively). We saw regions associated with regulation of stem cell differentiation, cell cycle transition and DNA damage response pathways in EV samples but these pathways were identified at increased levels when we over-expressed nuclear HK2 (figure A).

Regarding the ChIP-seq data in figure B & C, ChIP-seq peaks significant at FDR 0.01 were retrieved for each individual NLS1 HK2, NLS2 HK2 and endogenous EV samples. Genes near peaks were identified and were analyzed for pathway enrichment independently using GREAT. The enrichment map is showing all pathways represented as nodes that are significant at FDR 0.05 and that have a higher enrichment score in NLS1 HK2 or NLS2 HK2 compared to EV. The node pie chart slices indicate the enrichment score for each pooled NLS1 HK2, NLS2 HK2 and EV. The pathways form clusters of nodes when they share a lot of genes in common and are annotated as functional modules. Similar to the ATAC-seq enrichment results, we saw peaks associated with regulation of stem cell differentiation, cell cycle transition and DNA damage response pathways in EV samples but these pathways and peaks were identified at increased levels when we over-expressed nuclear HK2.

For figure C, the bar graph shows pathway enrichment scores of pooled ChIP-seq samples in endogenous (EV) and overexpression (NSL1-HK2, NLS2-HK2) conditions, for some pathways of interest that were present in figure B.

Reviewer #3:

Remarks to the Author:

Thomas GE et al., described an interesting non-canonical role for hexokinase 2 (HK2) in the regulation of AML stem cells. The authors found an accumulation of HK2 in the nucleus of AML cells and used shRNA and overexpression of nuclear HK2 to demonstrate that HK2 maintains their differentiation block and clonogenic capabilities in vitro. Chromatin analysis with overexpression of nuclear HK2 showed increased accessibility at regions linked to LSC signature and DNA damage repair genes. Finally, overexpressing nuclear HK2 enhanced the repair of DNA damage of AML cells treated with Daunorubicin.

The results are interesting but there are a number of concerns:

1. While the emphasis of the paper is on leukemic and normal stem cell function the assays used for these functions are very limited. The important experiments using knock-down of HK2 relied on in vitro and gene expression assays but not in vivo assays that more rigorously define stem cell function. The in vivo assays that were done rely on overexpression of nuclear HK2, but it is unclear if the levels resemble physiologic levels or if there are associated metabolic effects that may account for the functional changes. Firmer evidence for reliance on HK2 for stem cell function would give a stronger context for the elegant biochemistry that follows.

Response: We thank the reviewer for this important point. We had shown that selective knockdown of HK2 decreased clonogenic growth of AML cells before and after treatment with ATRA and reduced the number of leukemic stem cells (CD34+CD38-), both before and after ATRA treatment. Knockdown of nuclear HK2 also decreased expression of LSC+ and HSC+ genes signatures.

However, we agree that it is important to provide additional in vivo experiments and data demonstrating the importance of HK2 for stem cell function. Therefore, we conducted new in vivo experiments.

We over-expressed nuclear HK2 in TEX AML cells, engrafted the cells into mouse marrow and measured the survival capacity. Over-expression of nuclear HK2 decreased survival compared to wild type cells.

We also performed competitive repopulation assays using mouse marrow over-expressing nuclear HK2. Bone marrow cells from donor mice (CD45.2+; C57B6 background; VAV-NLS HK2 OR Wildtype) and clonogenic competitor mice (CD45.1+; B6.SJL) were mixed in a 1:1 ratio and injected into the tail vein of irradiated recipient mice (CD45.1 B6.SJL). Marrow and blood reconstitution was determined in peripheral blood (PB) samples taken 4, 6, 8, and 10 weeks post-transplant and in the bone marrow at the end of the experiment, 12 weeks after transplant.

Compared to wildtype chimera mouse, VAV NLS-HK2 chimera mice demonstrated increased repopulation of peripheral blood at week 4, 6, 8, and 10. We demonstrated that nuclear HK2 increased the repopulation efficiency in the bone marrow samples at week 12 post transplantation. We included these data in Fig 4G-H.

We also performed new in vivo experiments to assess the impact of selective knockdown of nuclear HK2. We selectively knocked down nuclear HK2 cells in OCI-AML2 cells and xenografted the cells into SCID mice. Compared to wild type controls, selective knockdown of nuclear HK2 delayed the growth of leukemia in vivo. These data are shown in Fig 2G, S2K-L.

Finally, we selectively knocked down nuclear HK2 in TEX cells injected cells into the femurs of NSGF mice. Five weeks later, engraftment of TEX cells into the mouse marrow was assessed by flow cytometry. Knockdown of nuclear HK2 decreased AML engraftment into mouse marrow. These data are shown in Fig 2H.

We hoped that we could also transduce primary AML cells to selectively knockdown nuclear HK2 and perform primary and secondary transplants. However, given the low transduction efficiency in patient samples it was not possible for us to transduce the primary cells with 2 vectors to express mitochondrial HK2 while knocking down nuclear

HK2.

2. The biochemical assays regarding interaction with other nuclear proteins and DNA depend upon overexpression of HK2. While these are interesting, it is not clear that these interactions occur at physiologic levels of HK2. Validation of the interactions with co-immunoprecipitation and CHIP experiments without overexpression are needed.

Response: Thank you for this comment. We agree it is important to confirm the hits identified in BioID interact with endogenous HK2. In response, we performed Duolink proximity assay in AML cells to assess the interaction of endogenous HK2 with hits from our BioID assay (MAX, SIRT1, IWS1, CTR9 and SPIN1). Using this PLA assay, we validated the endogenous interactions. We included these results in Fig Fig 5B, S8A.

In addition, we performed ChIP-seq with endogenous HK2 which represents the physiological level of nuclear HK2. In this context, we saw peaks associated with regulation of stem cell differentiation, cell cycle transition and DNA damage response pathways. The same pathways and peaks were identified when we over-expressed nuclear HK2, albeit at increased levels.

3. In terms of novelty, the authors mentioned in the discussion that nuclear HK2 has been reported in yeast. However, nuclear localization of HK2 has also been described in glioma and other cancer cells (Sheikh T et al., 2018, Neary CL et al., 2013). In addition, some of the mechanism described in regard to HK2 nuclear import and export have also been reported; Akt inhibition-mediated nuclear localization (Neary CL et al., 2013) and Xpo1-mediated nuclear export (Neary CL et al., 2010, Peláez R et al., 2009). While the findings found here relate specifically to AML cells, the authors should reference where appropriate these previous discoveries. These prior studies clearly impact the novelty of the findings here and further emphasize the importance of defining the cell biologic and functional consequences of HK2 modulation.

Response: Thank you for this comment. We agree with the reviewers comment and have updated the results section to include the recommended references (Page no:9, line no: 226-228).

4. The authors have shown that phosphorylation of HK2 is inversely related to nuclear localization and inhibiting AKT increases nuclear HK2. Since overall expression of HK2 does not seem to be increased in AML per public databases and AML is generally associated with hyperactivity of AKT/mTOR, how do AML cells have higher nuclear HK2?

Response: Thank you for this comment. The reviewer raises an interesting point. The determinants of nuclear HK2 levels are likely multifactorial. In addition, to its phosphorylation state, nuclear HK2 is also regulated by the importin, IPO5, and exportin, XPO1. We demonstrated that IPO5 is increased in AML cells.

We also note that the public datasets contain gene expression data, but do not contain protein expression in specific subcellular compartments.

5. Human data that correlate HK2 levels with a better prognosis:

https://servers.binf.ku.dk/bloodspot/?gene=HK2&dataset=normal_human_v2_with_AMLs should be discussed given the proposed relationship of nuclear HK2 to therapy resistance suggested by the authors.

Response: We thank the reviewer for highlighting these data. The Bloodspot data contains gene expression data and mRNA expression may not correlate well with nuclear protein levels. Therefore, we sought to determine whether levels of nuclear HK2 correlate with survival in AML patients. We analyzed the relationship between nuclear HK2 levels as measured by RPPA and the cumulative probability of survival. Although our sample size was relatively small (n=25) patients with increased total HK2 protein levels and increased nuclear HK2 protein levels trended towards decreased survival. Increased patient sample numbers would help clarify this relationship further – this needs to be investigated in a larger dataset in the future.

6. The authors show that nuclear HK2 enhances DNA damage repair, and at the same time that stem cell fraction has superior DNA damage repair compared with bulk AML cells. Because stem cells have a higher expression of nuclear HK2 than bulk cells, the authors state that AML stem cells have increased rates of DNA damage repair mediated by nuclear HK2. While this is a clear possibility, in order to conclude this the authors should modify nuclear HK2 levels in sorted stem and bulk fractions to rescue/restore their DNA damage repair capabilities.

Response: Thank you for this comment. We agree that the suggested experiment is important. In response, we transduced 8227 cells with NLS1-HK2 to increase the nuclear levels of HK2. 8227 cells are a low passage primary AML model that has stem cell properties and are organized in a hierarchy of stem and bulk cells with the stem cells residing in the CD34+CD38- fraction. We then sorted the transduced cells into stem and bulk populations. Stem and bulk cells were treated with daunorubicin for 3 hours and DNA damage repair markers were measured. In the bulk cells, over-expressing nuclear HK2 restored recruitment of the DNA repair protein 53BP1. Additionally, over-expressing nuclear HK2 in the stem cells increased recruitment of 53BP1 compared to EV stem cells. These data are show in Fig S100-P.

7. Previous studies show that glucose levels can change the levels of nuclear HK2 (Sheikh T et al., 2018, Neary CL et al., 2013). Can the authors be sure that the differences in nuclear HK2 observed in stem vs bulk AML cells are not due to the media conditions?

Response: The reviewer raises an interesting question. In response, we investigated whether changing the glucose composition of the media alters HK2 localization. Unlike yeast cells, decreasing the amount of glucose in the media did not have an effect on HK2 localization (Fig S5D).

In addition, the stem and bulk AML and normal hematopoietic cells in our experiments were exposed to the same media conditions throughout the experiments or were directly obtained from patients or mice. Thus, the media glucose levels were unlikely to have an impact.

Decision Letter, first revision:

Our ref: NCB-T45558A

11th February 2022

Dear Dr. Schimmer,

Thank you for submitting your revised manuscript "The metabolic enzyme Hexokinase 2 localizes to the nucleus in AML and normal hematopoietic stem/progenitor cells to maintain stemness" (NCB-T45558A). It has now been seen by the original referees and their comments are below. The reviewers find that the paper has improved in revision, and therefore we'll be happy in principle to publish it in Nature Cell Biology, pending minor revisions to satisfy the referees' final requests and to comply with our editorial and formatting guidelines.

In particular, from the remaining points of referee 1, we have decided that we do not need you to further address point 3 regarding correlations between nuclear HK2 and survival in patients based on database information.

On the contrary, we would please request that you address points 1 and 2 of this referee textually. In particular, we would like you to discuss in the manuscript the potential caveats of the method you use to model leukemia for this particular request, as per the point 1 of referee 1. In regard to point 2 of referee 1, we would please like you to explain and discuss in the manuscript your observation that distress and AML engraftment do not necessarily correlate and provide the citations supporting this mirrors the patient situation, as this is an important issue that scientists of the field would like to know.

You will also receive a list with all the changes you are required to do per our journal style and policies, before we can proceed with full acceptance and publication.

Thank you again for your interest in Nature Cell Biology Please do not hesitate to contact me if you have any questions.

Best wishes,
Stelios

Stylianos Lefkopoulos, PhD
He/him/his
Associate Editor
Nature Cell Biology
Springer Nature
Heidelberger Platz 3, 14197 Berlin, Germany

E-mail: stylianos.lefkopoulos@springernature.com
Twitter: @s_lefkopoulos

Reviewer #1 (Remarks to the Author):

The authors have generated and included new data in the revision of the manuscript that addressed most of my concerns. I have, however, still a few remaining concerns and some queries for clarification.

1. The authors have now added data showing that knockdown of nuclear HK2 in OCI-AML2 delays the growth and engraftment of leukemia in vivo (Fig 2G, H, and S2K-L). I expected the authors to perform intra-venous or intra-femoral injections and follow up the accumulation of human leukemic cells in the blood and survival. Instead the authors injected the cells subcutaneously into flanks of SCID mice and measured tumor volume. Could the authors explain the rationale for this approach and why it is a good model for leukemia? Figures S2K and S2L do not seem to be the correct ones.
2. The authors have now added data showing that overexpression of nuclear HK2 in TEX leukemia decreases survival compared to empty vectors. These data are not statistically significant. Was leukemia confirmed as the reason for death or observed when terminal endpoints were reached? How were the cells injected?
3. The authors indicate in their rebuttal that they cannot associate nuclear HK2 with any leukemia subtypes as the FAB subtype classification is not available for the samples used for RPPA analysis. Would it be possible to use the 25 patient samples they analyzed to evaluate the correlation between nuclear HK2 and survival (answer to point 5 of reviewer 3) to investigate if nuclear HK2 is associated with specific types of leukemia subtypes?

Reviewer #2 (Remarks to the Author):

The authors have addressed all of my concerns.

Reviewer #3 (Remarks to the Author):

The authors have done an excellent job in responding to concerns.

25th February 2022

Dear Dr. Schimmer,

Thank you for your patience as we've prepared the guidelines for final submission of your Nature Cell Biology manuscript, "The metabolic enzyme Hexokinase 2 localizes to the nucleus in AML and normal hematopoietic stem/progenitor cells to maintain stemness" (NCB-T45558A). Please carefully follow the step-by-step instructions provided in the attached file, and add a response in each row of the table to indicate the changes that you have made. Please also check and comment on any additional marked-up edits we have proposed within the text. Ensuring that each point is addressed will help to ensure that your revised manuscript can be swiftly handed over to our production team.

We would like to start working on your revised paper, with all of the requested files and forms, as soon as possible (preferably within one week). Please get in contact with us if you anticipate delays.

If you have not done so already, please alert us to any related manuscripts from your group that are

under consideration or in press at other journals, or are being written up for submission to other journals (see: <https://www.nature.com/nature-research/editorial-policies/plagiarism#policy-on-duplicate-publication> for details).

In recognition of the time and expertise our reviewers provide to Nature Cell Biology's editorial process, we would like to formally acknowledge their contribution to the external peer review of your manuscript entitled "The metabolic enzyme Hexokinase 2 localizes to the nucleus in AML and normal hematopoietic stem/progenitor cells to maintain stemness". For those reviewers who give their assent, we will be publishing their names alongside the published article.

Nature Cell Biology offers a Transparent Peer Review option for new original research manuscripts submitted after December 1st, 2019. As part of this initiative, we encourage our authors to support increased transparency into the peer review process by agreeing to have the reviewer comments, author rebuttal letters, and editorial decision letters published as a Supplementary item. When you submit your final files please clearly state in your cover letter whether or not you would like to participate in this initiative. Please note that failure to state your preference will result in delays in accepting your manuscript for publication.

Cover suggestions

As you prepare your final files we encourage you to consider whether you have any images or illustrations that may be appropriate for use on the cover of Nature Cell Biology.

Nature Cell Biology has now transitioned to a unified Rights Collection system which will allow our Author Services team to quickly and easily collect the rights and permissions required to publish your work. Approximately 10 days after your paper is formally accepted, you will receive an email in providing you with a link to complete the grant of rights. If your paper is eligible for Open Access, our Author Services team will also be in touch regarding any additional information that may be required to arrange payment for your article.

Please note that *Nature Cell Biology* is a Transformative Journal (TJ). Authors may publish their research with us through the traditional subscription access route or make their paper immediately open access through payment of an article-processing charge (APC). Authors will not be required to make a final decision about access to their article until it has been accepted. Find out more about Transformative Journals

Authors may need to take specific actions to achieve compliance with funder and institutional open access mandates. For submissions from January 2021, if your research is supported by a funder that requires immediate open access (e.g. according to Plan S principles) then you should select the gold OA route, and we will direct you to the compliant route where possible. For authors selecting the subscription publication route our standard licensing terms will need to be accepted, including our self-

archiving policies. Those standard licensing terms will supersede any other terms that the author or any third party may assert apply to any version of the manuscript.

Please use the following link for uploading these materials:
[REDACTED]

Best regards,

Nyx Hills
Staff
Nature Cell Biology

On behalf of

Stylianos Lefkopoulos, PhD
He/him/his
Associate Editor
Nature Cell Biology
Springer Nature
Heidelberger Platz 3, 14197 Berlin, Germany

E-mail: stylianos.lefkopoulos@springernature.com
Twitter: @s_lefkopoulos

Reviewer #1:

Remarks to the Author:

The authors have generated and included new data in the revision of the manuscript that addressed most of my concerns. I have, however, still a few remaining concerns and some queries for clarification.

1. The authors have now added data showing that knockdown of nuclear HK2 in OCI-AML2 delays the growth and engraftment of leukemia in vivo (Fig 2G, H, and S2K-L). I expected the authors to perform intra-venous or intra-femoral injections and follow up the accumulation of human leukemic cells in the blood and survival. Instead the authors injected the cells subcutaneously into flanks of SCID mice and measured tumor volume. Could the authors explain the rationale for this approach and why it is a good model for leukemia? Figures S2K and S2L do not seem to be the correct ones.

2. The authors have now added data showing that overexpression of nuclear HK2 in TEX leukemia decreases survival compared to empty vectors. These data are not statistically significant. Was leukemia confirmed as the reason for death or observed when terminal endpoints were reached? How were the cells injected?

3. The authors indicate in their rebuttal that they cannot associate nuclear HK2 with any leukemia subtypes as the FAB subtype classification is not available for the samples used for RPPA analysis. Would it be possible to use the 25 patient samples they analyzed to evaluate the correlation between nuclear HK2 and survival (answer to point 5 of reviewer 3) to investigate if nuclear HK2 is associated with specific types of leukemia subtypes?

Reviewer #2:

Remarks to the Author:

The authors have addressed all of my concerns.

Reviewer #3:

Remarks to the Author:

The authors have done an excellent job in responding to concerns.

Author Rebuttal, first revision:

Point-by-point response for Reviewer #1

The authors have generated and included new data in the revision of the manuscript that addressed most of my concerns. I have, however, still a few remaining concerns and some queries for clarification.

- 1. The authors have now added data showing that knockdown of nuclear HK2 in OCI-AML2 delays the growth and engraftment of leukemia in vivo (Fig 2G, H, and S2K-L). I expected the authors to perform intra-venous or intra-femoral injections and follow up the accumulation of human leukemic cells in the blood and survival. Instead the authors injected the cells subcutaneously into flanks of SCID mice and measured tumor volume. Could the authors explain the rationale for this approach and why it is a good model for leukemia? Figures S2K and S2L do not seem to be the correct ones.*

Response: We apologize for the numbering error - Supplemental figures S2K-L should be labelled as S3K-L. We thank the reviewer for catching this error. We agree with the reviewer that measuring AML engraftment after knockdown of nuclear HK2 is important and we have performed this experiment. As shown in figure 2H, we selectively knocked down nuclear HK2 in TEX cells and injected the cells into the femur of mice. 8 weeks after injection, we measured AML engraftment in the mice. Knockdown of nuclear HK2 decreased engraftment of cells into the mice. As TEX cells do not circulate in the peripheral blood after engraftment into mice, we could not measure peripheral blood levels of leukemia. In addition, as outlined in point 2 below, there is not a strong correlation

between the engraftment of TEX cells and level of distress of the mouse. In other words, mice can display distress with variable amounts of leukemia in the marrow – similar to patients. As such, we elected to measure leukemia marrow engraftment after selective knockdown of nuclear HK2, rather than survival.

As an additional experimental approach, we also selectively knocked down nuclear HK2 in OCI-AML2 cells and injected the cells subcutaneously into the mice. Knock down of nuclear HK2 delayed tumor growth over time and decreased tumor volume and tumor mass 3 weeks post injection. While we agree that the subcutaneous model is not as physiologically relevant, it permits the continuous direct visualization of tumor growth over time

Finally, Reviewer 1's original comment stated "The authors demonstrate that nuclear HK2 over-expression in AML cells results in increased engraftment (Figure 1F). Does this translate into an acceleration of leukemia development and decreased survival". We performed the requested experiment after injection of TEX cells over-expressing nuclear HK2. Survival was measured over time. Over-expression of nuclear HK2 decreased survival, compared to empty vector controls (Fig S2P,Q). Of note, we not specifically asked to perform in vivo experiments after the selective knockdown of nuclear HK2, but established these models and performed these experiments to provide additional data for the paper.

- 2. The authors have now added data showing that overexpression of nuclear HK2 in TEX leukemia decreases survival compared to empty vectors. These data are not statistically significant. Was leukemia confirmed as the reason for death or observed when terminal endpoints were reached? How were the cells injected?*

Response: In this experiment, we measured the survival of mice injected with TEX leukemia cells with and without the over-expression of nuclear HK2. Each mouse was injected with 2×10^5 cells intrafemorally. Mice were then followed overtime and sacrificed at the first sign of distress. The most likely cause of distress/death was AML, but necropsies were not performed on the mice at the time of death. We did not measure AML engraftment at the time of distress/sacrifice as we had already shown in previous experiments that the over-expression of nuclear HK2 increased AML engraftment into mice at 8 weeks post infection. However, the last mouse from the control and nuclear HK2 over-expressing group were sacrificed on the same day and AML engraftment was measured in the mouse marrow. The mouse injected with TEX cells over-expressing nuclear HK2 had higher blast counts compared to the mouse injected with wild type TEX cells (35.6% vs 12.4%), yet both mice died on the same day. While the higher AML engraftment in the mouse injected with TEX cells over-expressing nuclear HK2 is supportive of our conclusions, it also highlights limitations of the survival study. Despite the differences in levels of marrow engraftment, the mouse injected with control cells died on the same day as the mouse injected with TEX cells over-expressing HK2. Thus, there is not a perfect correlation between distress and AML engraftment – similar to the clinical situation in patients. Finally, we agree that the difference in survival approached, but did not quite reach statistical significance at $p = 0.054$. We chose to present the full p value and not round to $p = 0.05$

so the reader could judge the importance of the difference in survival.

3. The authors indicate in their rebuttal that they cannot associate nuclear HK2 with any leukemia subtypes as the FAB subtype classification is not available for the samples used for RPPA analysis. Would it be possible to use the 25 patient samples they analyzed to evaluate the correlation between nuclear HK2 and survival (answer to point 5 of reviewer 3) to investigate if nuclear HK2 is associated with specific types of leukemia subtypes?

Response: In response to a question by Reviewer 3, we demonstrated that increased nuclear HK2 protein levels were associated with a trend towards decreased survival. In response to the new question from Reviewer 1, we obtained cytogenetic information on 23/25 patients in this dataset. Unfortunately, molecular mutations were not available on these samples.

In our dataset, 45.5% (5/11) of patients with high nuclear HK2 levels had unfavorable cytogenetics, while 33% (4/12) of patients with low nuclear HK2 levels had unfavorable cytogenetics ($p = 0.67$). As with the survival data, increased patient sample numbers would help clarify the relationship between nuclear HK2 and cytogenetic and molecular abnormalities further.

Final Decision Letter:

Dear Aaron,

I am pleased to inform you that your manuscript, "The metabolic enzyme Hexokinase 2 localizes to the nucleus in AML and normal hematopoietic stem and progenitor cells to maintain stemness", has now been accepted for publication in Nature Cell Biology. Congratulations to you and your team!

Due to the importance of these deadlines, we ask that you please let us know now whether you will be difficult to contact over the next month. If this is the case, we ask you provide us with the contact

information (email, phone and fax) of someone who will be able to check the proofs on your behalf, and who will be available to address any last-minute problems.

Please note that *Nature Cell Biology* is a Transformative Journal (TJ). Authors may publish their research with us through the traditional subscription access route or make their paper immediately open access through payment of an article-processing charge (APC). Authors will not be required to make a final decision about access to their article until it has been accepted. Find out more about Transformative Journals

If you have not already done so, we strongly recommend that you upload the step-by-step protocols used in this manuscript to the Protocol Exchange (www.nature.com/protocolexchange), an open online resource established by Nature Protocols that allows researchers to share their detailed experimental know-how. All uploaded protocols are made freely available, assigned DOIs for ease of citation and are fully searchable through nature.com. Protocols and Nature Portfolio journal papers in which they are used can be linked to one another, and this link is clearly and prominently visible in the online versions of both papers. Authors who performed the specific experiments can act as primary authors for the Protocol as they will be best placed to share the methodology details, but the Corresponding Author of the present research paper should be included as one of the authors. By uploading your Protocols to Protocol Exchange, you are enabling researchers to more readily reproduce or adapt the methodology you use, as well as increasing the visibility of your protocols and papers. You can also establish a dedicated page to collect your lab Protocols. Further information can be found at www.nature.com/protocolexchange/about

nature portfolio

With kind regards,
Stelios

Stylios Lefkopoulos, PhD
He/him/his
Associate Editor
Nature Cell Biology
Springer Nature
Heidelberger Platz 3, 14197 Berlin, Germany

E-mail: stylios.lefkopoulos@springernature.com
Twitter: @s_lefkopoulos

** Visit the Springer Nature Editorial and Publishing website at www.springernature.com/editorial-and-publishing-jobs for more information about our career opportunities. If you have any questions please click here.**